# A novel numerical implementation for the surface energy budget of melting snowpacks and glaciers

Kévin Fourteau[1], Julien Brondex[1], Fanny Brun[2], and Marie Dumont[1]

[1]Univ. Grenoble Alpes, Université de Toulouse, Météo-France, CNRS, CNRM, Centre d'Études de la Neige, Grenoble, France
[2]Univ. Grenoble Alpes, CNRS, IRD, Grenoble INP, IGE, Grenoble, France

**Correspondence:** Marie Dumont (marie.dumont@meteo.fr)

**Abstract.** The surface energy budget drives the melt of the snow cover and glacier ice and its computation is thus of crucial importance in numerical models. This surface energy budget is the result of various surface energy fluxes, which depend on the input meteorological variables and surface temperature, of heat conduction towards the interior of the snow/ice, and potentially of surface melting if the melt temperature is reached. The surface temperature and melt rate of a snowpack or ice are thus driven by coupled processes. In addition, these energy fluxes are non-linear with respect to the surface temperature, making their numerical treatment challenging. To handle this complexity, some of the current numerical models tend to rely on a sequential treatment of the involved physical processes, in which surface fluxes, heat conduction, and melting are treated with some degree of decoupling. Similarly, some models do not explicitly define a surface temperature and rather use the temperature of the internal point closest to the surface instead. While these kinds of approaches simplify the implementation and increase the modularity of models, it can also introduce several problems, such as instabilities and mesh sensitivity. Here, we present a numerical methodology to treat the surface and internal energy budgets of snowpacks and glaciers in a tightly-coupled manner, including potential surface melting when the melt temperature is reached. Specific care is provided to ensure that the proposed numerical scheme is as fast and robust as classical numerical treatment of the surface energy budget. Comparisons based on simple test cases show that the proposed methodology yields smaller errors for almost all time steps and mesh sizes considered and does not suffer from numerical instabilities, contrary to some classical treatments.

## 1 Introduction

Snowpacks and glaciers are crucial parts of the Earth system that have a profound impact, among others, on the water cycle (e.g. Barnett et al., 2005) and on the radiative budget of continental surfaces (e.g. Flanner et al., 2011). A key tool to understand the interaction between snowpacks/glaciers and the other components of the Earth system are numerical models that aim to quantitatively represent the evolution of snowpacks and glaciers under various atmospheric forcings. To reach this goal, the representation and evolution of the thermodynamical state (that is to say temperature profiles and phase changes) of snowpacks and glaciers are implemented in most numerical snowpack/glacier models (e.g. Anderson, 1976; Brun et al., 1989; Jordan, 1991; Bartelt and Lehning, 2002; Liston and Elder, 2006; Vionnet et al., 2012; Sauter et al., 2020).

Among the various processes driving the thermodynamical state of snowpacks and glaciers, the surface energy budget (SEB)

has received detailed attention in the past, notably because of its central role (e.g. Etchevers et al., 2004; Miller et al., 2017; Schmidt et al., 2017, among many others). Indeed, the SEB governs most of the net energy input and output within the snowpack/glacier and thus has a fundamental role for its warming/cooling and for its melting. This SEB is the net result of various energy fluxes, including turbulent fluxes and long-wave radiative flux that directly depend on the surface temperature of the snowpack/glacier. Mathematically, the SEB thus appears as a non-linear top boundary condition for snowpacks and glaciers. This non-linearity is even reinforced by the existence of a regime change between a melting and non-melting surface, with different thermodynamical behaviors below and at the melting point. Indeed, once the melting point is reached at the surface, the SEB becomes more akin to a Stefan-problem with a discontinuity in the energy fluxes and can no longer be simply described in terms of surface temperature. This leads to numerical challenges when solving the governing equations.

As a consequence, there are currently no uniquely employed strategies to treat this problem, and various numerical schemes have been proposed and implemented for solving the SEB and its link with the thermodynamical state of a snowpack/glacier (Bartelt and Lehning, 2002; Vionnet et al., 2012; van Pelt et al., 2012; Sauter et al., 2020). Among the different published implementations, one can notably cite the so-called "skin-layer" formulation, usually employed in combination with a finite volume method (FVM) for the internal heat equation, in which the surface and internal temperatures are solved sequentially over a given time step (Oerlemans et al., 2009; Kuipers Munneke et al., 2012; van Pelt et al., 2012; Covi et al., 2023). While this approach naturally offers modularity and simplifies the treatment of the SEB (and of the associated surface temperature), a sequential treatment of tightly-coupled processes or variables is also known to display some instability (e.g. Ubbiali et al., 2021; Brondex et al., 2023) and large time step sensitivity (e.g. Barrett et al., 2019). On the other hand, some FVM implementations do not define a specific temperature associated with the surface, but rather use the temperature of the top-most numerical layer of the domain (i.e. the top layer of the simulated snowpack/glacier) for solving the SEB (Anderson, 1976; Brun et al., 1989; Jordan, 1991; Vionnet et al., 2012; van Kampenhout et al., 2017). While this enables to easily solve the SEB and the internal heat budget in a tightly-coupled way, this method requires to refine the numerical grid near the surface, in order to properly simulate the SEB. Thus, currently-employed FVM strategies in snowpack/glacier models present some limitations that can be detrimental for the obtained numerical solutions.

Here, we propose a FVM numerical scheme meant to combine the advantages of the previously published numerical strategies. Precisely, our goal is to offer a tightly-coupled treatment (as opposed to a sequential treatment) of the internal and surface temperatures of a snowpack or glacier. For this, the proposed implementation explicitly defines a temperature right at the surface (viewed as an infinitely thin horizontal layer), which improves the simulated results in terms of accuracy and stability. As the snowpack and glacier models are sometimes used in distributed or long-time spanning simulations, specific care is taken to ensure that the proposed numerical scheme has a similar numerical cost as the already published ones.

The article is organized as follows: Section 2 presents the physical equations governing the energy budget of snowpacks and glaciers, Section 3 briefly recalls some of the existing numerical schemes to solve these governing equations, and Section 4 presents the proposed numerical scheme overcoming some of the limitations of existing strategies, while keeping their strong points. Finally, some simple examples are presented in Section 5, and a discussion comparing the different numerical schemes

is provided in Section 6.

## 2 Governing equations

The goal of this Section is to briefly recall the general equations governing the thermal regime of snowpacks and glaciers, before presenting their numerical discretization in the next Section. As snowpack and glaciers share many similarities and processes, such as heat conduction or the presence of a phase transition when the melt temperature is reached, they can be represented by the same type of equations. These similarities enable simulations mixing snow and glacier ice within a single framework (e.g. Sauter et al., 2020). Hence, for the sake of generality, the equations discussed in the following sections apply to both snow and glacier ice. That being said, snow and glacier ice present some differences, notably concerning liquid water percolation. As addressed later, this might require a differential treatment of glacier ice and snow when implementing the liquid water percolation scheme.

### 2.1 Internal energy budget

The thermal regime of the inner part of a snowpack or glacier is governed by the principle of energy conservation. Assuming that Fourier's law of heat conduction applies in snow/ice with a well-defined macroscopic thermal conductivity (e.g. Calonne et al., 2011), this energy conservation writes:

$$\partial_t h - \nabla \cdot (\lambda \nabla T) = Q \tag{1}$$

where $h$ is the internal energy content of snow/ice (expressed in $\mathrm{J\,m^{-3}}$), $\lambda$ the thermal conductivity, $T$ the temperature, and $Q$ volumetric energy sources (such as the distributed absorption of shortwave radiations). Here, $h$ is understood as the energy content, including latent heat associated with the presence of liquid water (Tubini et al., 2021). The volumetric energy sources $Q$ (expressed in $\mathrm{W\,m^{-3}}$) therefore do not include the absorption or release of latent heat during solid/liquid water phase changes. In this article, we assume that the snowpack/glacier can be represented as 1D column, and therefore Eq. (1) should be understood as 1D equation.

Assuming thermodynamical equilibrium between the ice and liquid water, the temperature $T$ and the energy content $h$ are related through:

$$h = c_\mathrm{p}(T - T_0) + \rho_\mathrm{w} L_\mathrm{fus} \theta \tag{2}$$

where $c_\mathrm{p}$ is the volumetric heat capacity of snow/ice (expressed in $\mathrm{J\,K^{-1}\,m^{-3}}$), $T_0$ an arbitrary reference temperature taken as the melt temperature, $\rho_\mathrm{w}$ the density of liquid water, $L_\mathrm{fus}$ the specific enthalpy of fusion of water (expressed in $\mathrm{J\,kg^{-1}}$), and $\theta$ the liquid water content (expressed in $\mathrm{m^3}$ of liquid water per $\mathrm{m^3}$ of snow/ice) (Tubini et al., 2021).

Note that in Eq. (1) the time derivative of the internal energy content $h$ cannot in principle be replaced by $c_\mathrm{p}\partial_t T$, but should also include the term $\rho_\mathrm{w} L_\mathrm{fus}\partial_t \theta$. Indeed, once the temperature has reached the melting point, a further increase in energy translates into an increase in the liquid water content ($\partial_t \theta \neq 0$) and of the associated latent heat content, rather than a further increase in the temperature. Yet, as discussed below, snowpack and glacier models nonetheless usually consider that the temperature can increase past the melting point when integrating Eq. (1) in time (Vionnet et al., 2012; Sauter et al., 2020). This is equivalent to neglecting the effects of first-order phase changes (melting and refreezing) on the temperature field, and thus setting $\rho_\mathrm{w} L_\mathrm{fus}\partial_t \theta$ to zero while solving the heat equation. This results in temperature overshoots that are then corrected in a second step by creating melt and setting back the temperature to the melt value (e.g., Vionnet et al., 2012; Sauter et al., 2020). In this article, we follow this simple scheme as it is commonly employed in snowpack and glacier models. That being said, other, more complex, strategies have been proposed in the literature. This notably includes the use of a finite temperature-range over which melt/freezing occurs (e.g. Albert, 1983; Dutra et al., 2010), including melt/refreeze as an additional energy source term (e.g. Bartelt and Lehning, 2002; Wever et al., 2020), or the use of enthalpy as the prognostic variable (e.g. Meyer and Hewitt, 2017; Tubini et al., 2021). Finally, in this article we consider the thermal conductivity $\lambda$ and capacity $c_\mathrm{p}$ not to depend on temperature. The motivation for this is twofold as it (i) corresponds to a simplifying assumption regularly made by snowpack and glacier surface models (e.g. van Pelt et al., 2012; Vionnet et al., 2012; Sauter et al., 2020; Covi et al., 2023) and (ii) it allows keeping the internal heat equation linear.

## 2.2 Surface energy balance

To model an actual snowpack/glacier subjected to atmospheric forcings, it is necessary to complement the internal energy budget with an appropriate boundary condition. At the top of the snowpack/glacier, this boundary condition is given by the SEB. This SEB states that the net sum of energy fluxes between the top of the snowpack/glacier and the atmosphere equals the energy thermally conducted from the surface to the interior of the snowpack plus a potential surface melting term if the melt temperature is reached (Oerlemans et al., 2009; Sauter et al., 2020; Covi et al., 2023). We thus have:

$$SW_\mathrm{net}^\mathrm{surf} + LW_\mathrm{in} + LW_\mathrm{out} + H + L + R = G + \dot{m}L_\mathrm{fus} \tag{3}$$

where $SW_\mathrm{net}^\mathrm{surf}$ is the net shortwave radiation absorbed right at the surface (that is thus distinguished from the portion of shortwave radiation penetrating within the snow/ice), $LW_\mathrm{in}$ is the incoming longwave radiation flux, $LW_\mathrm{out}$ is the outgoing longwave radiation flux, $H$ is the turbulent sensible heat flux, $L$ is the turbulent latent heat flux, $R$ the surface energy brought by precipitating rain, $G$ is the conductive heat flux penetrating within the snowpack/glacier, and $\dot{m}$ is the rate of surface melting (expressed in $\mathrm{kg\,m^{-2}\,s^{-1}}$). Fluxes are orientated towards the bottom, and thus towards the surface for $SW_\mathrm{net}^\mathrm{surf}$, $LW_\mathrm{in}$, $LW_\mathrm{out}$, $H$, $L$, and $R$ and away from the surface for $G$. The surface melting rate $\dot{m}$ vanishes when the surface temperature $T_\mathrm{s}$ is below the melt temperature, and can take non-zero values when the surface temperature equals the melt temperature.

Among the various terms of the SEB of Eq. (3), $LW_{out}$, $H$, $L$, and $G$ depend non-linearly on the surface temperature $T_s$. Notably, the outgoing longwave radiation is given by Stefan-Boltzmann law, i.e. $LW_{out} = -\sigma T_s^4$ (with $\sigma$ the Stefan-Boltzmann constant) and the turbulent heat fluxes $H$ and $L$ can be estimated through the use of a bulk approach (e.g. Foken, 2017). These three terms are therefore non-linear functions of the surface temperature. In addition, the conductive heat flux is given by

$$G = -\left(\lambda \partial_z T\right)|_{z=\text{surf}} \tag{4}$$

and is therefore proportional to the temperature gradient within snow/ice right below the surface. This conductive flux depends on both the surface temperature $T_s$ and the temperature within the snow/ice. This flux is responsible for the thermal coupling between the surface and the interior of the snowpack/glacier.

## 3 Numerical strategy of existing models

Since the computation of the heat budget with a SEB as a top boundary condition is at the core of all snow/glacier models, several numerical implementations have been proposed for solving the resulting system of equations. In order to provide a general overview of the numerical frameworks and strategies, we propose to separate them into two broad classes, to which most existing models can somehow be related. While classifying existing strategies into only two groups (and not more) remains arbitrary, we believe it is helpful to highlight differences in handling the numerical solving of the energy budget. Moreover, we focus on numerical schemes based on FVM, as it is the method employed by most models (e.g. Anderson, 1976; Sauter et al., 2020; Westermann et al., 2023). We note that, contrary to the FVM, the use of the finite element method (FEM) naturally incorporates the presence of a surface temperature, which can be used for a fully-coupled treatment of the SEB, as done in SNOWPACK for instance (Bartelt and Lehning, 2002).

### 3.1 Class 1: Finite volumes without explicit surface

A first class of models relies on FVM for discretization of the internal heat budget, without the inclusion of an extra degree of freedom to model the surface temperature (schematically depicted as Class 1 in Fig. 1). To this end, the domain to be modeled (snowpack or glacier) is first decomposed into a finite number of cells with non-zero thicknesses (that are also sometimes referred to as layers, but should not be confused with the strata forming a snowpack). Then, the equations governing the temporal evolution of the average heat content of each cell is determined by integrating Eq. (1) over each cell. The energy fluxes between cells are finally estimated based on cell-to-cell temperature differences and on the thermal conductivities of the cells. As discussed above, the effects of the first-order phase transition during melting/refreezing are usually not taken into account when solving the internal heat budget. Rather, it is considered that snow/ice temperature can exceed the melt temperature without modification of its physical behavior (i.e., of its heat capacity). When integrating the equations in time, this can result in temperature overshooting the melt point. These overshoots are later used to determine where the melting point has been

crossed, and the excess energy is then used to estimate melting (e.g. Vionnet et al., 2012).

This FVM framework thus amounts to determining the average temperature in each cell, which is usually considered to cor-
respond to the temperature at the center of the cell. Without further modification, the surface temperature, which corresponds
to the temperature on the upper edge of the top cell, is not present in the system of equations. In order to apply the SEB as a
boundary condition, this first class of models considers the surface temperature to be equal to the temperature of the top-most
cell. The energy fluxes between the surface and the atmosphere are then directly integrated into the heat budget of the top cell.
The internal heat budget and the integrated surface fluxes can then be solved at the same time, i.e. in a tightly-coupled fashion.
The advantage of this approach is that it naturally allows one to take into account the SEB within a standard FVM framework,
without the necessity to handle extra degrees of freedom. This numerical strategy roughly corresponds to the one adopted in
SNTHERM (Jordan, 1991), Crocus (Vionnet et al., 2012), CLM (van Kampenhout et al., 2017), or CryoGrid (Westermann
et al., 2023).

## 3.2 Class 2: Finite volumes with an explicit but decoupled surface

The second class of models also relies on FVM for the spatial discretization of the internal heat budget. Similarly to the mod-
els of class 1, the first-order phase transition of snow/ice is usually neglected for the resolution of the equations, resulting in
temperature overshoots that are later corrected by creating melting.

However, this class of models explicitly takes into account the presence of a surface temperature that differs from the temper-
ature of the cell just below (schematically depicted as Class 2 in Fig. 1). This surface temperature is computed by searching
for the temperature that equilibrates the SEB of Eq. (3), assuming no melting. If the equilibrium temperature is larger than the
melting point, it is then capped to the melt temperature and the excess surface energy converted into surface melting.

Because of the numerical complexity of this task, it is usually performed separately from the computation of the internal heat
budget. Typically, the surface temperature is first resolved, using the internal temperatures of the previous time-step for the
heat conduction term of the SEB, and then the internal temperatures are solved using the newly computed surface temperature
and SEB.

This class of models encompasses the models using a so-called skin-layer formulation for the SEB. Its advantage is that it
allows to explicitly define a surface temperature without complexifying the solving of the internal heat budget and keeping a
low numerical cost. It roughly corresponds to the models SnowModel (Liston and Elder, 2006), EBFM (van Pelt et al., 2012),
or COSIPY (Sauter et al., 2020).

Finally, we want to stress that the actual implementations of the aforementioned models (e.g. Crocus, SNTHERM, COSIPY,
EBFM, etc) cannot be perfectly captured by our simple classification. Particular choices regarding the spatial and temporal
discretizations, the treatment of melting and refreezing, and the coupling between individual processes make each model
unique and more complex than the above presentation. Also, models can in principle display the characteristics of both classes
(i.e. no explicit surface and a SEB solved with a decoupling from the rest of the domain), although we did not find any concrete

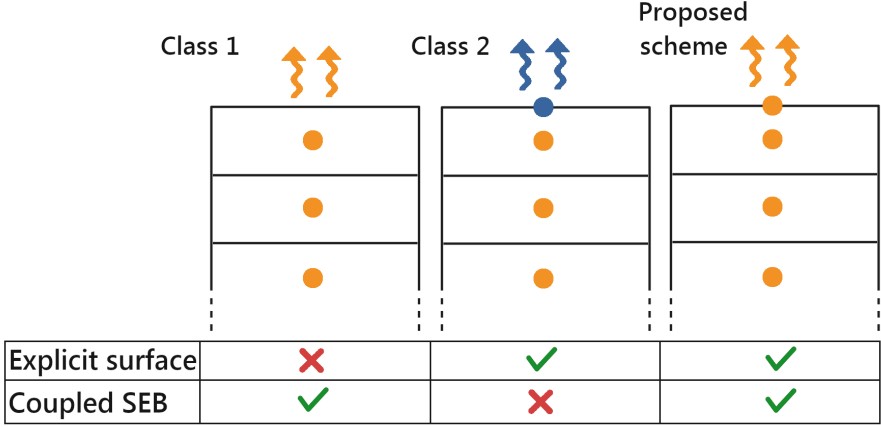

**Figure 1.** Classification of FVM models with respect to their treatment of the SEB. Class 1: The surface energy and the internal temperatures are solved in a tightly-coupled manner, but there is no explicit surface. Class 2: An explicit surface temperature (and surface melting) exists, but it is solved in sequential manner with respect to the internal temperatures. Proposed scheme in this article: An explicit surface temperature is considered and is solved in a tightly-coupled manner with the internal temperatures. In the schematic, dots represent the prognostic variables of the schemes (with or without temperature at the surface) while the colors indicate which variables are solved simultaneously.

example. This diversity of models offers an actual illustration of how the numerical implementation of the same processes (internal heat budget with a complex SEB) has been handled by different authors.

## 4  A tightly-coupled solution for the surface and internal heat budget

As seen above, each class of models comes with advantages but also limitations. While class 1 models solve the internal and SEB in a tightly-coupled manner, they do not take into account the fact that the surface temperature is in general different from the temperature in the cell below. On the contrary, while class 2 models explicitly consider a surface temperature, the internal and SEB are treated in a sequential, and therefore loosely-coupled fashion, which can be detrimental to stability (Ubbiali et al., 2021).

Based on these observations, the goal of this section is to present a FVM methodology that allows one (i) to explicitly work with a surface temperature, (ii) to treat the surface and internal heat budgets in a tightly-coupled fashion. Moreover, as the goal of this paper is to focus on the treatment of the SEB and its coupling with the internal thermal state, we also follow the standard approach to handle melting in the interior of the domain. Namely, first-order phase transition effects are neglected while solving for the internal energy budgets. This means that interior temperatures will overshoot in case of melting, and this excess temperatures will be used to generate melt afterward.

## 4.1 Governing system of discretized equations

In this section, we derive the discretized equations governing the coupled surface and internal heat budgets, based on the FVM. For this, we consider a domain divided into $N$ cells. The temporal evolution of the average heat content of each cell is given by integrating Eq. (1) over the cell and making use of the fundamental theorem of calculus. Neglecting phase change during the resolution of the internal heat budget, the time derivative of the (average) temperature $T_\mathrm{k}$ of the $k^\mathrm{th}$ cell is given by:

$$\Delta z_\mathrm{k} c_{\mathrm{p}_\mathrm{k}} \partial_t T_\mathrm{k} + F_{\mathrm{k}+\frac{1}{2}} - F_{\mathrm{k}-\frac{1}{2}} - \Delta z_\mathrm{k} Q_\mathrm{k} = 0 \tag{5}$$

where $\Delta z_\mathrm{k}$ is the thickness of the $k^\mathrm{th}$ cell, $c_{\mathrm{p}_\mathrm{k}}$ its volumetric heat capacity, $Q_\mathrm{k}$ the average volumetric energy source in the cell, and $F_{\mathrm{k}+\frac{1}{2}}$ and $F_{\mathrm{k}-\frac{1}{2}}$ are the heat conduction fluxes at the top and bottom interfaces of the cell. For internal cells, $F_{\mathrm{k}+\frac{1}{2}}$ and $F_{\mathrm{k}-\frac{1}{2}}$ correspond to the fluxes between the $k^\mathrm{th}$ and the $k+1^\mathrm{th}$ cells and the $k-1^\mathrm{th}$ and $k^\mathrm{th}$ cells, respectively. For the top cell $F_{\mathrm{k}+\frac{1}{2}}$ corresponds the heat flux leaving towards the surface (i.e. $-G$) and for the bottom cell $F_{\mathrm{k}-\frac{1}{2}}$ corresponds to the flux from the ground. By convention, we take $F_{\mathrm{k}+\frac{1}{2}}$ as positive if the heat flux is oriented from the $k^\mathrm{th}$ cell to the $k+1^\mathrm{th}$. Note that in this paper we consider the $1^{st}$ cell to be at the bottom of the snowpack, and the cells to be counted positively upwards. Other numbering choices could be made and would lead to the same end-result.

The heat conduction fluxes between cells need to be estimated from the temperatures and thermal conductivities of adjacent cells. The flux $F_{\mathrm{k}+\frac{1}{2}}$ between cells $k$ and $k+1$ is computed as:

$$F_{\mathrm{k}+\frac{1}{2}} = \lambda^\mathrm{harm}_{\mathrm{k}+\frac{1}{2}} \frac{T_\mathrm{k} - T_{\mathrm{k}+1}}{\frac{\Delta z_\mathrm{k}}{2} + \frac{\Delta z_{\mathrm{k}+1}}{2}} \tag{6}$$

where $\lambda^\mathrm{harm}_{\mathrm{k}+\frac{1}{2}}$ is the weighted harmonic average of the thermal conductivity of the two adjacent cells. The use of a harmonic average provides better results in the case of layered media such as snow (Kadioglu et al., 2008) and ensures that no heat conduction occurs in case one of the cells is a perfect thermal insulator.

Note that Eq. 6 only applies to fluxes between cells and must be replaced for the two boundary cells, at the top and bottom of the domain. For the bottom cell, a flux between the domain and the ground below must be used as a bottom boundary condition. For the top cell, the heat flux coming from the surface must be used. This flux is given by the discretized version of the term $G$ in the SEB, provided in Eq. (10) below.

This FVM discretization results in $N$ equations governing the evolution of the $N$ internal temperatures. The surface temperature can be added to this system of equations by introducing an additional degree of freedom, localized at the top of the domain. This surface temperature can be deduced from the SEB of Eq. (3) and its coupling to the interior of the domain through the subsurface heat flux $G$ of Eq. (4). However, the SEB cannot be fully characterized using the surface temperature only. Indeed, in case of melting, the surface temperature is blocked at the melt temperature $T_0$ and can no longer be used as

a prognostic variable to characterize the surface. In this case, it is necessary to introduce a non-zero melting rate $\dot{m}$ to close the energy budget. We thus have two regimes for the surface: below the melting point the surface is fully characterized by its temperature and the melting rate term vanishes; at the melting point, the surface temperature becomes constant and the melting rate term $\dot{m}$ becomes the quantity that characterizes the state of the surface. At any time, the surface is fully characterized by only one independent variable, but neither the temperature nor the melt rate can be used in the general case.

To circumvent this problem, we rely on a variable switching technique (Bassetto et al., 2020). Concretely, we introduce a fictitious variable, denoted $\tau$, whose goal is to behave as $T_s$ below the melting point and as $\dot{m}$ during melting. In other words, we parametrize the $\{T_s(\tau), \dot{m}(\tau)\}$ graph, such that every possible state of the surface can be appropriately described by a well-defined $\tau$ value. A possibility is to take $\tau$ such as:

$$T_s = \begin{cases} \tau & \text{if} \quad \tau < T_0 \\ T_0 & \text{otherwise} \end{cases} \tag{7}$$

and

$$\dot{m} = \begin{cases} 0 & \text{if} \quad \tau < T_0 \\ \frac{\tau - T_0}{\beta} & \text{otherwise} \end{cases} \tag{8}$$

where $\beta$ is an arbitrary constant, necessary to ensure dimensional homogeneity (concretely taken as $1 \, \text{kg} \, \text{m}^{-2} \, \text{s}^{-1} \, \text{K}^{-1}$ in our implementation).

Then, the SEB can be expressed as:

$$SW_{\text{net}}^{\text{surf}} + LW_{\text{in}} + LW_{\text{out}}(\tau) + H(\tau) + L(\tau) + R(\tau) - G(\tau) - \dot{m}(\tau)L_{\text{fus}} = 0 \tag{9}$$

where the dependence of $LW_{\text{out}}$, $H$, $L$, $R$, and $G$ to $\tau$ through $T_s$ has been made explicit. The subsurface conduction heat flux can thus be approximated by spatially discretizing Eq. (4):

$$G = \lambda_{\text{k}} \frac{T_s(\tau) - T_k}{\Delta z_{\text{k}}/2} \tag{10}$$

where the index $k$ is taken to correspond to the top-most cell. As explained above, this flux must also be taken into account in the equation governing the heat content of the top-most cell.

We thus have a system of $N+1$ equations (one for each cell plus the SEB), which governs the evolution of $N+1$ prognostic variables (the temperature of each cell plus the surface temperature/melt-rate encapsulated into $\tau$). To be numerically solved, this system also requires a temporal discretization. In this article, we choose an implicit backward Euler's method for its

simplicity and stability (Fazio, 2001; Butcher, 2008). Nonetheless, the method proposed here could also be applied with other temporal integration schemes (e.g. Crank-Nicolson).

This system of equations presents several non-linearities, coming from the non-linearity of some terms in the SEB with respect to the surface temperature ($LW_{\text{out}}$, $H$, and $L$) and from the regime change of the surface (between melting and non-melting conditions). In order to deal with these non-linearities, we rely on the use of a specific Newton's method, described below. We also note that some models made the choice of performing only a single iteration to solve this linear system of equations (with sometimes an extra iteration to handle specific cases, such as surface melting). However, we chose here to perform multiple iterations, in order to obtain the actual Backward Euler solution.

### 4.1.1   A dedicated Newton's method

One of the main benefits of the skin-layer formulation used by models of class 2 is its low numerical cost. Indeed, all the non-linearity of the problem only appears in the SEB, i.e. in a single scalar equation that can be solved iteratively. While iterations are costly in numerical models, this cost is here tempered by the fact that this only needs to be performed on a scalar equation, with a limited number of terms to be re-estimated at each iteration. Once the surface temperature has been determined, the internal temperatures can be solved through a N×N linear system of equations that does not require multiple iterations. On the contrary, solving the (N+1)×(N+1) non-linear system of equations derived in Section 4 can be much more numerically expensive if the whole system is to be re-assembled and re-inverted at each iteration.

Keeping this issue of numerical cost in mind, we propose a numerical strategy to solve the system of equations describing the coupled internal and surface energy budgets. It is based on a modified Newton scheme, with two modifications proposed to make the iteration process both more robust and faster.

Truncation method for regime changes:

A first modification made to this standard Newton's method is the use of the truncation method when crossing discontinuities during the iteration process (Wang and Tchelepi, 2013; Bassetto et al., 2020). The idea behind truncation is that the Jacobian (i.e. the derivative of the equations with respect to the unknowns to be solved for) computed on one side of a derivative discontinuity does not apply on the other side, and can therefore perturb the convergence towards the solution, typically leading to an endless iteration loop. In our model, this problem notably arises from the SEB that shows discontinuity with respect to $\tau$ when crossing the melting point. A similar problem can also appear in the turbulence terms of the SEB. For instance, some formulations of the turbulent fluxes can include derivative discontinuities for the stability correction of the latent and sensible fluxes with respect to the bulk Richardson number (as in e.g. Martin and Lejeune, 1998; Sauter et al., 2020). Thus, during the iteration process each time the surface changes regime (between non-melting/melting or stable/unstable conditions), the value of $\tau$ is brought back in the vicinity of the regime change by setting $\tau = \tau^* \pm \epsilon$, where $\tau^*$ is the value for which a derivative discontinuity occurs. This truncation procedure is schematized in Fig. 2, depicting a switch between a non-melting and melting surface. The numerical parameter $\epsilon$ is made to ensure that the next iteration starts from the good regime and needs to be taken

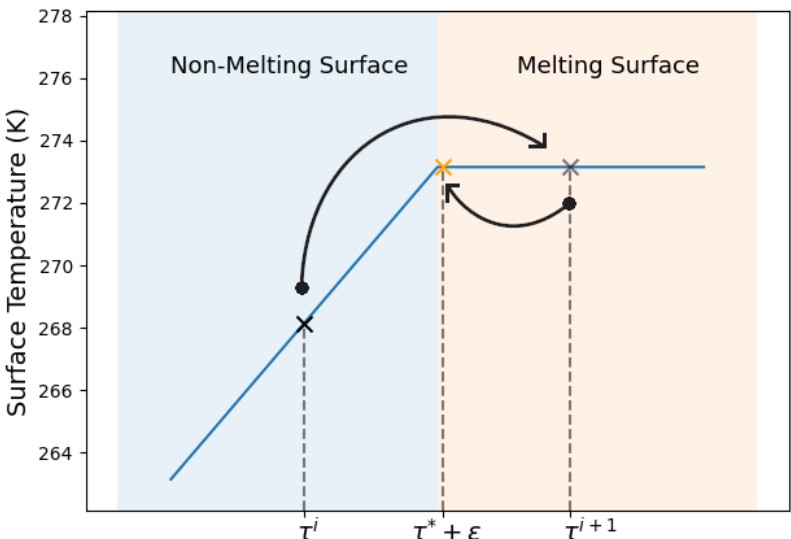

**Figure 2.** Example of the truncation method made to handle derivative discontinuities during Newton's iterations (schematic inspired by Fig. 2.3 of Bassetto, 2021). Starting from an estimate $\tau^i$, a new estimate $\tau^{i+1}$ is computed based on the Jacobian estimated at $\tau^i$. As a derivative discontinuity is crossed, the fictitious variable $\tau$ is set back near the discontinuity $\tau^*$ but in the "melting surface" regime.

small (typically $10^{-5}$).

Variable elimination to reduce the size of the non-linear problem:

A second improvement can be made by realizing that most of the equations governing the internal heat budgets are actually
linear equations, and thus only need to be assembled and inverted once per time step. Indeed, the (N-1) first equations, corresponding to the time evolution of the temperature of the internal cells not in contact with the surface, express simple linear relationships between the N internal cell temperatures. This can be used to reduce the size of the non-linear system to be iteratively solved.

For this, we eliminate the $N-1$ linearly-dependent variables using a Schur complement technique (Zhang, 2006). Concretely,
writing the system of Eqs. (5) and (9) in block-matrix form, one has:

$$\left(\begin{array}{c|c} A_{\text{diag}} & A_{\text{up}} \\ \hline A_{\text{low}} & A_{\text{s}} \end{array}\right) \left(\begin{array}{c} T_{\text{int}} \\ U_{\text{s}} \end{array}\right) = \left(\begin{array}{c} B_{\text{int}} \\ B_{\text{s}} \end{array}\right) \tag{11}$$

where $A_{\text{diag}}$, $A_{\text{up}}$, $A_{\text{low}}$, and $A_{\text{s}}$ are $(N-1) \times (N-1)$, $(N-1) \times 2$, $2 \times (N-1)$, and $2 \times 2$ matrices, respectively. Note that we refer to the vector composed of the two last unknowns, thus composed of $[T_N, \tau]$, as $U_{\text{s}}$ in order not to have it mistaken

with the surface temperature. The expressions of the matrices forming the block system are given in Appendix A, including the derivatives necessary for Newton's method.

Under this form, the matrices $A_{\text{diag}}$, $A_{\text{up}}$, $A_{\text{low}}$ and the vector $B_{\text{int}}$ are constant during the non-linear iterations and do not need to be re-estimated at each non-linear iteration. Thus, the $(N-1)$ internal temperatures can be expressed as:

$$T_{\text{int}} = A_{\text{diag}}^{-1}(B_{\text{int}} - A_{\text{up}}U_{\text{s}}) \tag{12}$$

and thus

$$(A_{\text{s}} - A_{\text{low}}A_{\text{diag}}^{-1}A_{\text{up}})U_{\text{s}} = B_{\text{s}} - A_{\text{low}}A_{\text{diag}}^{-1}B_{\text{int}} \tag{13}$$

where $A_{\text{s}} - A_{\text{low}}A_{\text{diag}}^{-1}A_{\text{up}}$ corresponds to the Schur complement of $A_{\text{diag}}$ in the system of Eqs. (11) (Zhang, 2006).

The system of Eqs. (13) is a $2 \times 2$ non-linear system where only $A_{\text{s}}$ and $B_{\text{s}}$ need to be re-assembled at each non-linear iteration and whose solution for $U_{\text{s}}$ is the same as the large system of Eqs.( 11). Therefore, an efficient numerical scheme to solve the whole system of Eqs. (11) is to (i) first assemble $A_{\text{low}}$, $A_{\text{diag}}$, $A_{\text{up}}$, and $B_{\text{int}}$, (ii) compute the products $A_{\text{diag}}^{-1}B_{\text{int}}$ and $A_{\text{diag}}^{-1}A_{\text{up}}$ (which is cheaper than directly inverting $A_{\text{diag}}$), (iii) iteratively solve the $2 \times 2$ non-linear system of Eqs. (13) yielding $U_{\text{s}}$ (only reassembling $A_{\text{s}}$ and $B_{\text{s}}$ at each iteration), and (iv) retrieve the remaining internal temperatures by applying Eq. (12).

This technique, namely eliminating linearly-dependent variables using a Schur complement to reduce the size of non-linear systems to be solved for, can also be applied to speed up the solving of class 1 models. This is presented in Appendix B. We also note that to apply this technique, the assumption of temperature-independent heat capacity and conductivity is important, as otherwise the internal heat equation system would not be linear and thus the matrices $A_{\text{diag}}$, $A_{\text{up}}$, and $A_{\text{low}}$ not constant. Finally, a translation of this numerical strategy (including the fictitious variable and the Schur-complement technique) in a FEM framework is presented in Appendix C.

An analysis of the numerical cost (in terms of number of basic operations) of this numerical scheme is given in Appendix A, alongside analyses of the numerical cost of Class 1 and 2 models. It shows that the proposed scheme and the Class 1 models have similar numerical costs, which a bit less than 1.7 times larger than the standard-skin layer.

## 5 Simulation setup

The system of equations 11 and its resolution scheme presented in Section 4 enable the computation of the tightly-coupled evolution of the surface and of the internal energy budget. The goal of this section is to compare this approach to more classical implementations, falling either in class 1 (all temperatures solved at once but without an explicit surface) or class 2 (presence of an explicit surface, but sequential treatment for the computation of the surface and internal temperatures).

For this purpose, we thus implemented a class 1 and a class 2 model alongside the scheme presented in Section 4. For the implementation of a class 1 model, a specific treatment of the first cell is adopted. Indeed, in order to have results comparable with the other model implementations, the temperature of the first cell is computed taking into account the effect of first-order phase transition in order to cap the surface temperature at $T_0$. The resulting non-linear system is solved with the modified Newton method presented in Section 4.1.1, including the truncation and Schur-complement techniques. Not taking into account first-order phase transitions in the first cell would result in surface temperature overshoots (not present in the other implementations), which would be detrimental to the SEB. We stress that our specific implementation has differences with already published models (for instance the Crocus model does not perform non-linear iterations and treats surface melting differently; Vionnet et al., 2012), and thus that the results obtained with our implementation might deviate from that of the aforementioned models (Crocus, SNTHERM, Cryogrid, or CLM).

For the implementation of a class 2 model, we adopt the following sequential treatment for each time step: (i) first the surface temperature that equilibrates the SEB is computed using the internal temperatures of the previous time step and ignoring potential melting, (ii) if the surface temperature exceeds melt it is capped at $T_0$ and the excess energy used for surface melting, (iii) the internal temperatures are then computed using the value of the sub-surface heat flux $G$ computed from the SEB as the top boundary condition. Again, our specific implementation of a class 2 model might differ from some of the already existing "skin-layer" models (COSIPY, EBFM, or SnowModel).

In order to obtain physically sound results, note that we have included a treatment of water percolation through a simple bucket scheme (Bartelt and Lehning, 2002; Vionnet et al., 2012; Sauter et al., 2020) as well as the representation of the motion of the surface in response to surface melting and vapor sublimation/deposition. In our bucket-scheme, cells whose density is close to that of ice are considered as impermeable and water cannot percolate through them. Instead, excess water present in cells above an impermeable horizon is sent to runoff. This choice is meant to avoid liquid water percolation through an entire glacier. Our models also include a remeshing algorithm that merges adjacent cells when they become smaller than a given threshold (defined here as $75\%$ of the smallest cell size at the start of a simulation). This remeshing step is also used to ensure that the melt of a layer cannot exceed its ice content. If such a case is encountered, the layer is merged with one of its neighbors before attempting melting. If the total melt exceeds the total mass, the simulations should be stopped. However, this last case did not arise in the simulations presented here. These processes (melting, percolation, and remeshing) are treated after the resolution of the heat budget and are handled in a sequential (and thus partially decoupled) fashion, as usually done

in current snowpack/glacier modeling (e.g. Bartelt and Lehning, 2002; Vionnet et al., 2012; Sauter et al., 2020). To ease comparison between the various implementations, the melting, percolation, and remeshing routines are common to all of them. The temporal integration scheme is also the same for all models in order to facilitate the comparison between them, namely an implicit backward Euler method. Also, as some of the current snowpack and glacier models include the effect of internal phase-change while solving the internal heat equation (e.g. Bartelt and Lehning, 2002; Meyer and Hewitt, 2017), we quantified the sensitivity of our results to this specific treatment of melt/freeze. For that, we have also implemented versions of our three models that include such internal phase-changes in the heat equation.

Finally, note that we do not include the FEM in this comparison. As detailed in Appendix C, a specificity of FEM models is to rely on a temperature field that can be defined element-wise or node-wise. It is thus required to convert back and forth between these two representations. However, the relation between the two is not bijective. This prevents an unambiguous transformation from element-wise to node-wise temperatures, which affects the end-result of our simulations. Because of this problem, the FEM is not further explored in this article, as a direct comparison to the FVM models is not possible.

Two simple examples, showcasing the differences between numerical treatments, are presented below. We note that these simulations cannot be considered as fully realistic simulations of a snowpack or glacier surface, as many processes, such as the deposition of atmospheric precipitation (rain or snow) or mechanical settling, are lacking. The goal is rather to provide a simplified setting in which the impact of the numerical implementation of the SEB can be analyzed. In the same idea, we do not attempt to compare the simulation results to field observations. Indeed, it would not be possible to decipher errors due to the numerical discretization (the focus of this paper) from errors due to the assumed physics, parametrizations and atmospheric forcings. Nonetheless, in order for the results to still be informative of how a given numerical implementation might behave in a more realistic setting, we use realistic atmospheric forcing, initial conditions, and physical parametrizations. The first simulation is meant to highlight the behavior of the numerical models when simulating the SEB on a snow-free glacier. The second one focuses on the impact of the model implementations on the simulation of the energy budget of a seasonal snowpack, during the melting period.

## 5.1 Test case 1: Snow-free glacier

We start by considering the case of a snow-free and firn-free glacier, neglecting the accumulation of mass through precipitation. This test case is motivated by the recent studies of Potocki et al. (2022) and Brun et al. (2022), which discuss current models capability of modeling the surface mass balance of such a snow and firn-free glacier in a cold environment.

As such, our simulations are forced by the weather data provided by Potocki et al. (2022) that include all necessary information to take into account the shortwave, longwave and turbulent energy fluxes at the top of our domain. To compute the shortwave absorption, we assume that the surface has a constant broadband albedo of $0.4$ and that $80\%$ of the flux is absorbed right at the surface (Bintanja and Broeke, 1995; Sauter et al., 2020), without penetrating deeper. The remaining shortwave

radiation penetrates in the ice following an exponential decay profile with a $0.4\,\mathrm{m}$ e-folding depth (Bintanja and Broeke, 1995; Sauter et al., 2020). The longwave emissivity of the ice is assumed to be unity. Finally, the turbulent fluxes are computed based on a slightly modified version of Eqs. (17-21) of Sauter et al. (2020) and are described in the Appendix D. The roughness length over the ice surface is taken constant and set to $z_0 = 1.7\,\mathrm{mm}$ (Sauter et al., 2020). For the bottom boundary condition, we apply a simple no-heat-flux condition. As the simulated domain is large (about $189\,\mathrm{m}$) and the simulation only run for a single year, this choice of bottom boundary condition has little effect on the simulated surface temperature and energy budget. For instance, we performed a simulation in which a $64.7\,\mathrm{mW\,m^{-2}}$ geothermal heat flux is applied instead (Davies, 2013). The impact on the surface temperature remains below $0.4\,\mathrm{mK}$.

For the internal material properties, we assumed the ice heat capacity $c_{\mathrm{p}}$ to equal $2000\,\mathrm{J\,K^{-1}\,kg^{-1}}$ and not to depend on temperature (Lide, 2006). Similarly, the ice thermal conductivity $\lambda$ is set to $2.24\,\mathrm{W\,K^{-1}\,m^{-1}}$, independently of temperature (Lide, 2006; Sauter et al., 2020). Finally, we want to stress that in such a case of a snow and firn-free glacier, the numerical implementation of our bucket-scheme results in the runoff of all melted water, without percolation into the glacier and thus without warming the ice below it.

For the initial conditions, we used a spin-up simulation presented in Brun et al. (2022) and generated with the COSIPY model (Sauter et al., 2020). It corresponds to an initially $189\,\mathrm{m}$ thick glacier. The output of the spin-up notably includes a non-uniform mesh for the glacier, from which we build the meshes for our simulations. In order to study the influence of spatial resolution on the simulation, the original spin-up mesh was refined/downgraded by increasing/decreasing the number of cells. This was done by keeping the same relative cell sizes in the domain, such that the smallest cells remained near the surface and the largest ones deep in the glacier, as in the original spin-up mesh.

Finally, we want to stress that the aforementioned simplifying assumptions (such as constant albedo, constant surface roughness length, absence of precipitation, simplistic treatment of percolation, etc) imply that the results of our simulations should not be quantitatively interpreted. Rather, the choice of simplified physics is meant to ease the comparison of the numerical treatments of the SEB.

For each numerical scheme, we perform simulations with initial numbers of cells varying between 22 and 450 and with time steps ranging from 30 to 7200 s. This range includes the time steps typically used in models (e.g. 900 s in Crocus or 3600 s in COSIPY). In the absence of an analytical solution, the simulations performed at a high spatial and temporal resolution (i.e. 30 s and 450 cells) are meant to provide a reference to study the convergence of the other simulations with the gradual increase of the spatial and temporal resolutions. These high-resolution simulations reveal that the class 1 model implementation (no explicit surface) remains different from the two other implementations even for this level of time step and mesh refinement. Therefore, as the reference solution for the glacier test-case, we take the average of the two implementations with an explicit surface, as they both converged to similar solutions (and similar results will thus be obtained if only the solution of the proposed tightly-coupled surface scheme were taken). Specifically, to quantify the difference between a given simulation and the reference, we focus on the surface temperature and on the phase change rate (understood in this article as the net melt and refreeze

over the entire domain after solving the heat equation). For this purpose, we compute the time series of absolute differences between the simulations and the reference, as well as the corresponding Root-Mean-Square-Deviation (RMSD). Note that in this specific test case, no refreezing was observed (as melt occurs at the surface and is sent to runoff), meaning that the phase change rate directly corresponds to the melt rate.

## 5.2 Test case 2: Melting snowpack

Our second test case corresponds to the case of a melting snowpack. For simplicity, we assume that the snowpack surface has a constant broadband albedo of $0.7$ and that all shortwave radiation penetrates in the snow following an exponential decay profile with a $0.058\,\text{m}$ e-folding depth (Bintanja and Broeke, 1995; Sauter et al., 2020). Similarly to that of ice, the longwave emissivity of snow is assumed to be unity. The turbulent fluxes are computed with the same law as in the glacier test case but with a constant roughness length of $z_0 = 0.24\,\text{mm}$ (Sauter et al., 2020). As in the glacier case, the bottom boundary condition for the heat equation is taken as no-flux condition. The use of a more realistic boundary condition could be achieved by coupling the snowpack model to a soil model (e.g. Decharme et al., 2011). It however remains beyond the scope of this article, which is focused on the impact of the implementation of the SEB on simulations.

Regarding internal material properties, we assume snow to have the specific heat capacity of ice, i.e. $2000\,\text{J}\,\text{K}^{-1}\,\text{kg}^{-1}$, independent of temperature (Lide, 2006; Morin et al., 2010). The thermal conductivity of snow is taken as a function of density, following the Calonne et al. (2011) parametrization. For the percolation scheme, we assume that a snow cell is able to retain up to $5\%$ of its porosity as liquid water (Vionnet et al., 2012). Liquid water percolating from the last cell of the snowpack is simply sent to runoff. The initial conditions of the simulation are taken from a Crocus simulation of the snowpack at Col de Porte (Lejeune et al., 2019) during the 2010/2011 season. As we are interested in the case of melting, we start our simulation from the 14/03/2011, corresponding to the peak of snow height in the Crocus simulation ($1.49\,\text{m}$), run it for 63 days, and stop it before reaching the total disappearance of the snowpack in our simulations. The original Crocus mesh is refined/downgraded by increasing/decreasing the number of cells in order to study the impact of mesh resolution of the numerical solutions. The atmospheric forcings, for both the spin-up and the simulation, are based on the reanalysis of Vernay et al. (2022). Finally, as in the glacier case, the results of the simulations should not be quantitatively interpreted (for instance in terms of days for snowpack disappearance) but are only meant to provide an easy way of comparison between numerical treatments of the internal and surface energy budgets.

The simulations are performed with initial cell numbers varying between $22$ and $440$ and with time steps ranging from $30$ to $7200\,\text{s}$. As in the glacier test case, the high-resolution simulations ($30\,\text{s}$ time step and $440$ cells) are meant to provide a reference solution. In this case, all three models converge to similar solutions with the considered levels of mesh and time step refinement. Thus, the reference solution was taken as the average of the three implementations. The comparison between a given simulation and the reference was done focusing on the surface temperature and the phase change rate, as in the glacier test-case.

## 6 Results and Discussion

### 6.1 General behavior of the models

An example of simulated surface temperature, phase change rate, and temperature profiles obtained in the glacier test case for a time step of $3600\,\text{s}$ and an initial cell number of $44$ (corresponding to a minimum cell size of $10\,\text{mm}$ at the top) is displayed in Fig. 3. Similarly, for the snowpack test case, simulated surface temperatures, phase change rates, and temperature profiles obtained for a time step of $3600\,\text{s}$ and a starting cell number of $44$ (corresponding to minimum of cell size of $9.1\,\text{mm}$ at the top) are visible in Fig. 4.

While the three models tend to generally agree in terms of simulated surface temperatures and phase change rates, they nonetheless present some notable differences. Concerning the glacier test-case, Fig. 3 shows that the class 1 model (no explicit surface) is systematically different compared to the two other models, with a slower decrease of the surface temperature at night, resulting in a surface temperature that is usually warmer of a couple of degrees for the represented period. For comparison, Sauter et al. (2020) report root-mean-square-errors around $3\,\text{K}$ when comparing COSIPY simulations with observations of the Zhadang glacier surface temperature. Besides the surface temperature, the class 1 model also displays internal temperatures (starting from about $10\,\text{cm}$ below the surface) that are colder (by about $0.50\,\text{K}$) than the two other implementations. This internal temperature difference is consistent with the fact that the surface temperature in the class 1 model is on average warmer than the two others, favoring the loss of energy through turbulent and radiative fluxes.

As in the glacier test case, models tend to generally agree in the snowpack case, with nonetheless some differences as displayed in Fig. 4. In particular, all predict that most of the melt occurs internally and without the surface temperature necessarily reaching the melting point. As previously, the class 2 model and the new tightly-coupled approach exhibit the best agreement (even though the agreement is not as clear as with the glacier case), while the class 1 model displays surface temperatures that reach higher peaks during the day. As with the glacier test case, the models exhibit surface temperature differences of about a couple of degrees. This is of the same order as the biases observed in the snow model inter-comparison exercise ESM-SnowMIP (Menard et al., 2021). Despite their relative agreement, the class 2 model appears to "lag" by about one time step behind the tightly-coupled implementation. This lag can be explained by the fact that, in this case, shortwave radiations are not directly affected to the surface (as they penetrate). A large variation in shortwave radiations is therefore not directly visible by the surface, which only reacts to it at the next time step, once the shortwave radiations have impacted the cell below the surface. The impact of this lagging problem can be mitigated by the use of small time steps, but with the drawback of numerical cost. Beside surface temperature, the class 1 model also shows differences compared to the two other models in terms of internal temperatures, being colder in the deepest part of the snowpack. This effect is due to the smaller melting predicted by the class 1 model. There is therefore less melt water percolating down the snowpack, which carries latent heat to warm the snowpack. Finally, we note that the class 2 model exhibits some time step to time step oscillations, characteristic of numerical instability. Such oscillations are visible both in the surface temperature and the phase change rate that display over and undershoots com-

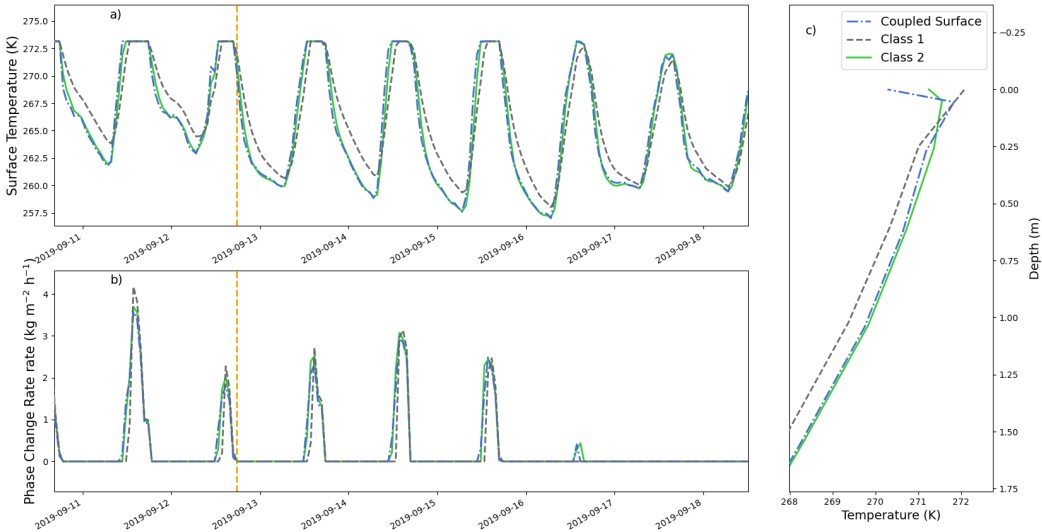

**Figure 3.** Overview of the simulation of a snow and firn-free glacier using three different numerical schemes. The simulations were performed with a time step of $3600\,\text{s}$ and an initial number of cells of $44$ (minimum cell size of $10\,\text{mm}$). a) and b): Surface temperature and total phase change rate (including surface and subsurface melt/refreeze) around mid-September. c): Upper part of the temperature profiles on the 12/09/2019 at 15:45 local time. The dashed orange line in panels a) and b) corresponds to the selected date of panel c).

pared to the other models.

Finally, using the versions of the models including phase-changes in the heat equation, we quantified the sensitivity of these observations to the treatment of the melt/refreeze. While the simulated temperature sometimes differ from our basic implementations (especially in the snowpack test case where melt occurs internally), the general behavior of the models, including the potential presence of instabilities in the Class 2 models, remain unchanged.

## 6.2 Convergence with time step and mesh refinement

As they solve the same physical equations, all numerical implementations of the heat budget are expected to converge to the same results when the time step size and mesh size tend to zero. However, in general different numerical implementations do not show the same levels of error and convergence rates toward this solution, as the time step and mesh size are progressively reduced. The goal of this section is to analyze the convergence of the three SEB implementations discussed in this article with time step and mesh size refinement. In other words, we quantify their respective time step and mesh size sensitivities.

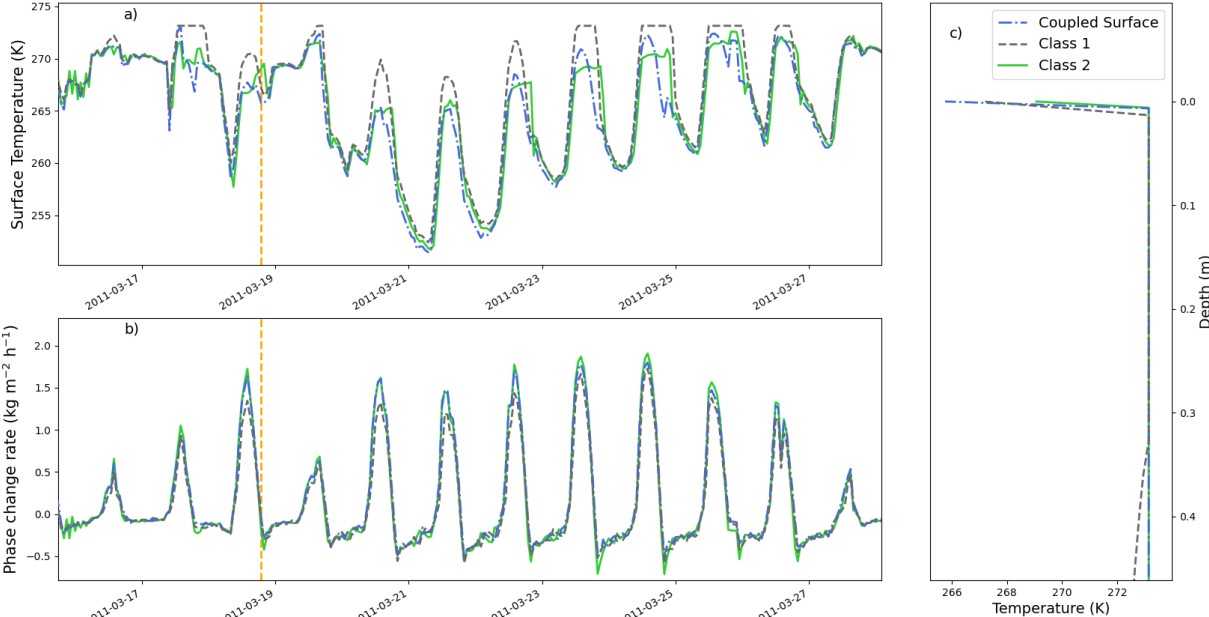

**Figure 4.** Overview of the simulation of a snowpack using three different numerical schemes. The simulations were performed with a time step of 3600 s and an initial number of cells of 44 (minimum cell size of 9.1 mm). a) and b): Surface temperature and total phase change rate (including surface and subsurface melt/refreeze) near the end of March. Note that negative phase change rate values imply refreezing within the snowpack. c): Upper part of the temperature profiles on the 18/03/2011 at 19:00 local time. The dashed orange line in panels a) and b) corresponds to the selected date of panel c).

We start here by analyzing the sensitivity of the three numerical implementations to the time step. For this purpose, we analyze the differences between the reference solutions and the three implementations using about 220 cells (i.e. about 5 times the usual number of cells used in detailed models) and time steps between 112 and 7200 s. Figures 5 to 8 compare the simulations performed with various time steps to the reference (time step of 30 s) for the glacier and snowpack test cases, respectively. The largest time step of 7200 s corresponds to twice the default value used for instance in COSIPY (Sauter et al., 2020) and

is meant to represent the case of models used at quite large time steps for numerical cost considerations. Note that for the left panels showing time series of absolute differences, a 10 days running average was used to remove daily and weekly variability from the data. Also, while the right panels display RMSDs over the entire simulation, we also computed biases. These were in general about an order of magnitude smaller than the RMSD values, except for the surface temperature of the snowpack test case, where the bias was about half of the RMSD.

As seen in the four Figures, all models show a general decrease in errors with smaller time steps. For almost all investigated time steps and in both test cases, the newly proposed scheme displays the lowest level of errors. Sometimes, the class 2 model yields the smallest error, but does so only by a small margin. Figure 5 reveals that for the glacier test case and at large time steps

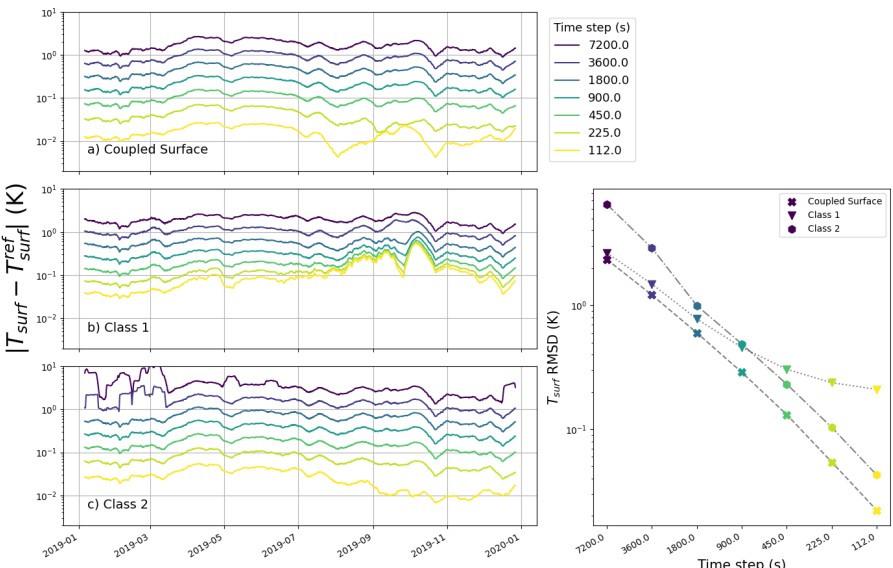

**Figure 5.** Impact of time step size on the simulated surface temperature for the glacier test case and for the three numerical schemes. Left panels a), b), and c): Errors in surface temperature for the different implementations (panels) and for different time step sizes (colors) during the simulated period. Right panel: RMSD of the surface temperature over the whole simulated period for each implementation (marker) and time step (color). The same time step color scheme applies to all panels.

(between 30 min and 2 h), the decoupled skin-layer formulation (class 2 model) shows the largest errors in terms of surface
temperature, with a marked increase of the error with increasing time steps. However, we do not observe such a sharp increase
at large time steps for the phase change rate errors with the class 2 model, even though Fig. 6 highlights that for such large
time steps, the class 2 model wrongly predicts melting early in the season (notably during the month of February). Figures 5
and 7 show that for smaller time steps and in both test cases, it is on the contrary the class 1 model that yields the largest errors
in terms of surface temperature, with a limited decrease in the error level with decreasing time steps compared to the two other
implementations. Concerning the phase change rate errors for small time steps, it depends on the investigated test case: for the
glacier it is the class 2 model that shows the largest errors (Fig. 6), while it is the class 1 model for the snowpack test case
(Fig. 8). The results of the glacier test case displayed in Figs. 5 and 6 thus highlight that depending on the considered metric
(surface temperature or phase change rate), the ranking of models might differ.

Similarly, while the numerical results are expected to converge to the same solution when the grid is refined, they do not
show the same errors and convergence rates with decreasing mesh size. Notably, integrating the top boundary conditions di-
rectly in the first cell (as in class 1 models) instead of adding an extra independent variable at the surface is known to slow the
convergence of FVM with mesh refinement, as it requires a very small top-cell to properly approximate the surface temperature.
As with time step sensitivity, we quantify the impact of mesh refinement by comparing simulations performed with different

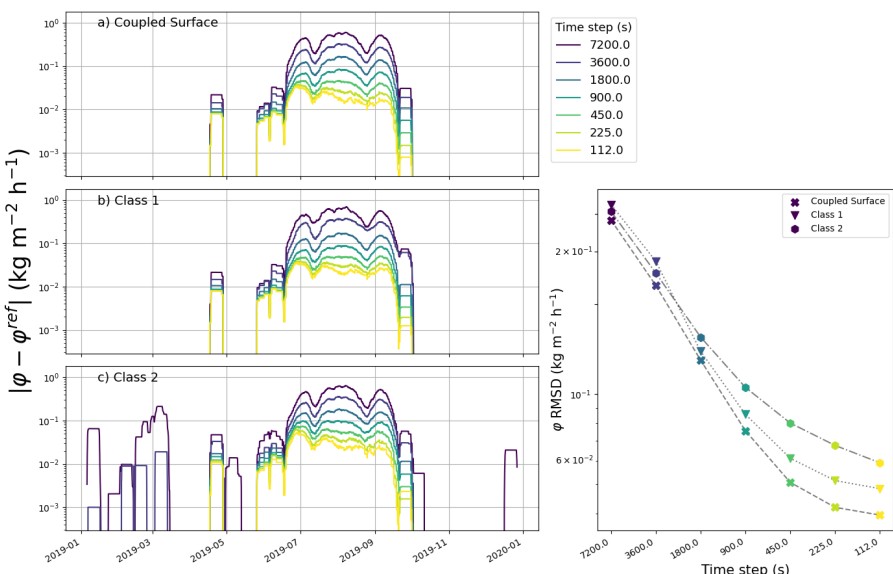

**Figure 6.** Impact of time step size on the simulated phase change rate (here denoted $\varphi$ to lighten the plot) for the glacier test case and for the three numerical schemes. Left panels a), b), and c): Errors in phase change rate for the different implementations (panels) and for different time step sizes (colors) during the simulated period. Right panel: RMSD of the phase change rate over the whole simulated period for each implementation (marker) and time step (color). The same time step color scheme applies to all panels.

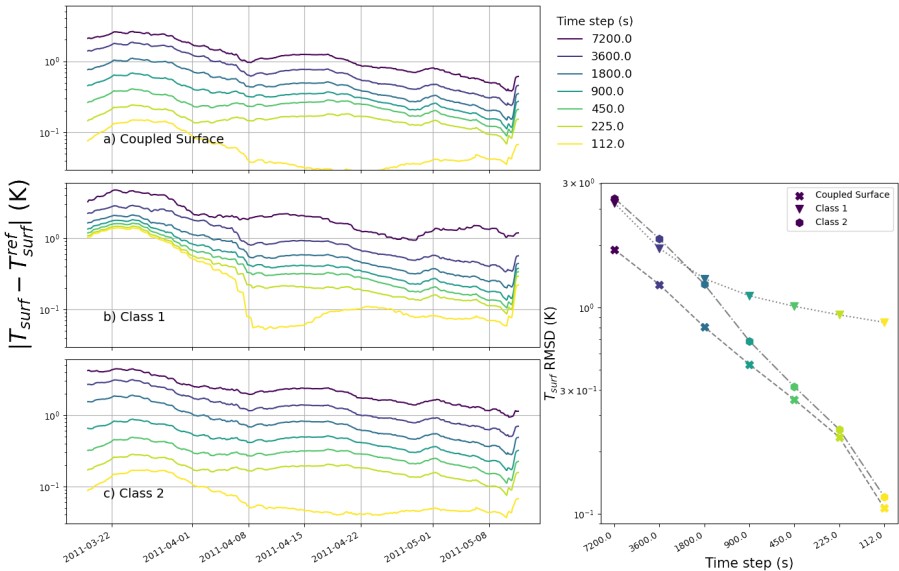

**Figure 7.** Same as Fig. 5, but for the snowpack test case.

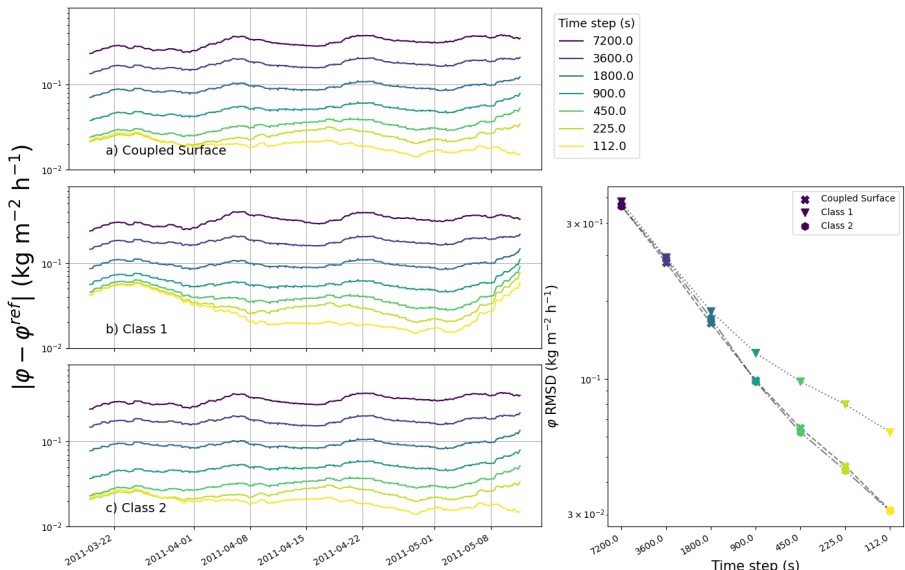

**Figure 8.** Same as Fig. 6, but for the snowpack test case.

spatial resolutions to reference simulations. We used the same reference simulations as with the time step analysis. The results are displayed in Figs. 9 to 12 and show the errors in terms of surface temperature and phase change rate for both investigated test cases. As with the time step convergence, bias values over the simulations were found to be an order of magnitude smaller than the RMSD values.

As with time step refinement, all models display a general decrease of errors with finer meshes. Again, among the three implementations the tightly-coupled surface model yields the smaller errors for almost all investigated mesh refinements (as in the glacier test case, the class 2 model is however sometimes marginally better). On the other hand, the class 1 model displays comparatively large errors for almost all mesh refinements and for both test cases. As seen in Fig. 11, this is particularly marked in the snowpack simulation, where the class 1 simulation with the finest mesh refinement (about 220 initial cells) has the same

level of surface temperature error as the two other models with a coarser mesh (44 initial cells). In other words, in this case, the class 1 model needs about five times more cells (and thus five times thinner cells) to achieve the same precision as the two other implementations. The addition of an extra degree of freedom to represent the surface is thus highly beneficial and offers the possibility to use coarser (and thus computationally cheaper) meshes. Finally, Fig. 10 reveals that in the glacier test case, the phase change rate errors of the class 2 tend to deteriorate with further mesh refinement past a certain point (here for an initial

cell number above 90). We interpret this deterioration as a result of the appearance of numerical instabilities that develop with small mesh sizes. Due to this effect, the class 2 model exhibits the largest phase change rate errors for an initial number of cells of 225. Finally, using the versions of the models including phase-changes in the heat equation, we verified that the conclusions of this convergence analysis remain valid in the case of a different treatment of the internal phase-changes.

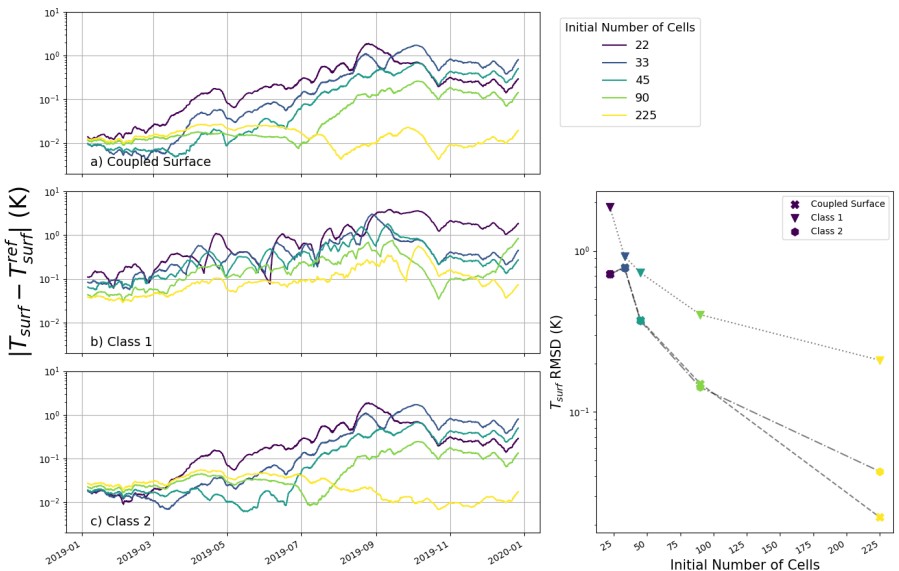

**Figure 9.** Impact of mesh size on the simulated surface temperature for the glacier test case and for the three numerical schemes. Left panels a), b), and c): Errors in surface temperature for the different implementations (panels) and for different mesh sizes (colors) during the simulated period. Right panel: RMSD of the surface temperature over the whole simulated period for each implementation (marker) and mesh size (color). The same mesh size color scheme applies to all panels.

## 6.3 Tight-coupling as a way to reduce instabilities

As discussed above, the decoupled nature of the standard skin-layer formulation (class 2 models) leads to greater errors for large time steps compared to the two coupled formulations, with or without an explicit surface. Moreover, the class 2 model can show some deterioration in the case of highly-refined meshes (Fig. 10). Both these phenomena can be explained by the fact that the skin-layer formulation displays instabilities. We observe especially large instabilities for time steps of 2 hours, visible as oscillations in the temperatures of the surface and of the cell below, with peak-to-peak amplitudes sometimes reaching $100\,\mathrm{K}$

and with a daily running standard deviation up-to about $50\,\mathrm{K}$. Such oscillations then lead to an abnormally cold and warm surface and a deteriorated SEB. As displayed in Fig. 13, these instabilities are even worsened in the case of mesh refinement. On the contrary, no such instabilities have been observed for the tightly-coupled schemes (with or without an explicit surface). The unstable nature of class 2 models can be shown with a linear stability analysis, provided in Appendix E. Such analysis shows that class 2 models are only conditionally stable, and confirm that instabilities are favored in the case of large time steps

and small mesh sizes. We stress that these oscillations can appear even if the time integration of the internal energy budget relies on the Backward Euler method, known for its robustness against instabilities (Fazio, 2001; Butcher, 2008). Our understanding is that the sequential treatment of the standard skin-layer formulation breaks the implicit nature of the time integration by using "lagged" (in other words, explicit rather than implicit) terms. This, combined with the fact that the surface layer does not

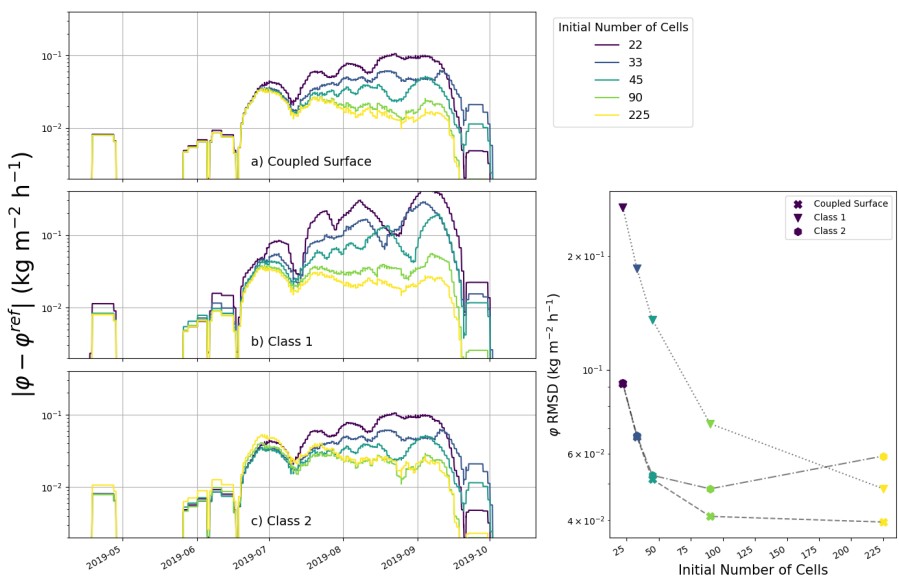

**Figure 10.** Impact of mesh size on the simulated phase change rate (denoted here $\varphi$ to lighten the plot) for the glacier test case and for the three numerical schemes. Left panels a), b), and c): Errors in the phase change rate for the different implementations (panels) and for different mesh sizes (colors) during the simulated period. Right panel: RMSD of the phase change rate over the whole simulated period for each implementation (marker) and mesh size (color). The same mesh size color scheme applies to all panels.

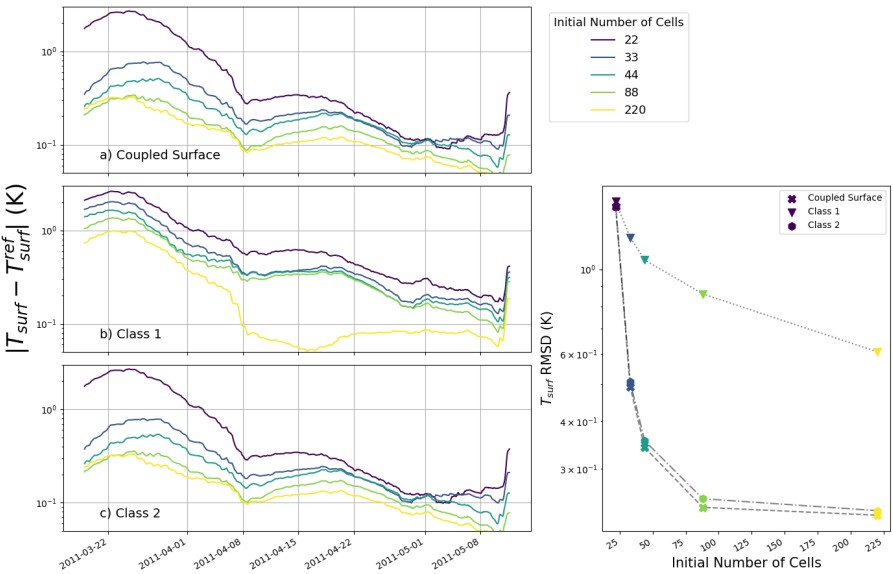

**Figure 11.** Same as Fig. 9, but for the snowpack test case.

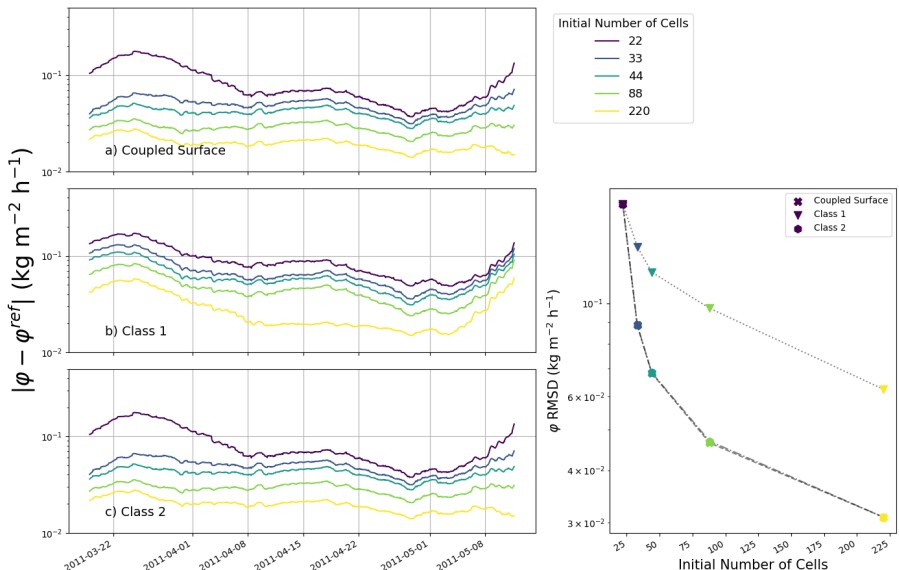

**Figure 12.** Same as Fig. 10, but for the snowpack test case.

possess any thermal inertia and that its temperature can thus vary rapidly in time, permits large temperature swings if the time
step is too large or the mesh size too small. On the other hand, it can be shown that the two schemes with a tightly-coupled
SEB are unconditionally stable (Appendix E), in agreement with the absence of oscillations in their simulations. Notably, the
unconditional stability of the coupled-surface scheme proposed in this article entails that the model does not need an adaptive
time step size strategy depending on the mesh size. This ensures that it remains robust, regardless of the time step and mesh
size.

## 6.4 Energy conservation in the standard skin-layer formulation

As explained in Section 2.2, the heat conduction flux from the surface to the interior of the domain (i.e. $G$ in Equation 3) needs
to have the same value in the computation of the SEB and in the computation of the energy budget of the first interior cell.
Inconsistencies in $G$ between these two budgets lead to the violation of energy conservation and create an artificial energy
source/sink near the surface. Such inconsistencies could be created when implementing the standard skin-layer formulation
(class 2 models) due to the sequential treatment of the surface and internal energy budgets. Indeed, after solving the SEB, one
can either use the surface temperature or the subsurface heat flux $G$ as a boundary condition for the computation of the internal
temperatures. We note that the use of the computed surface temperature as a boundary condition leads to an unconditionally
stable numerical scheme (Appendix E). However, using such Dirichlet condition in order to stabilize the standard-skin layer
formulation comes at the expense of energy conservation and deteriorates of the simulated results.

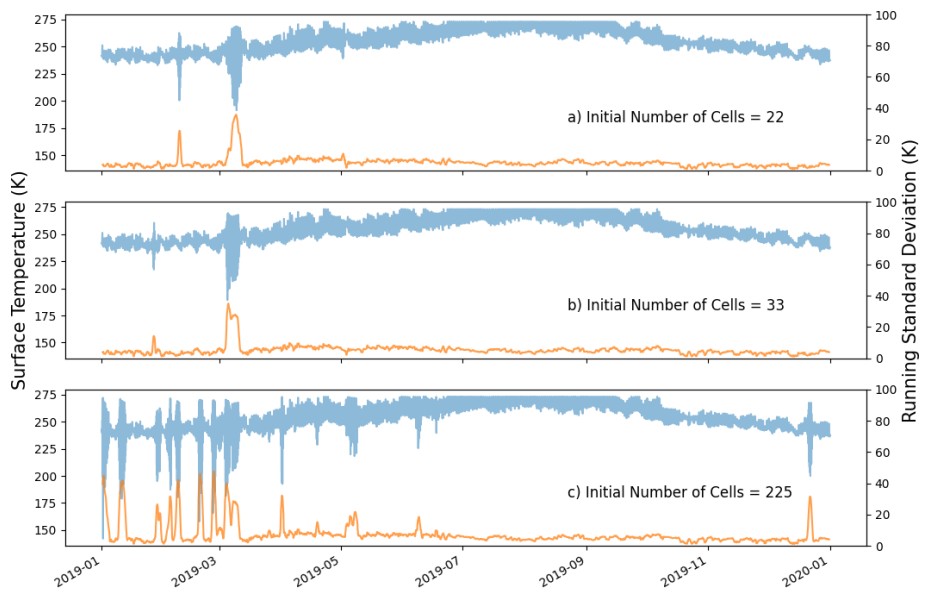

**Figure 13.** Time series of surface temperatures (in blue, left y-axis) and of their 24hr-running standard deviations (in orange, right y-axis) highlighting the presence of numerical instabilities with the standard skin-layer scheme. The simulations correspond to the glacier test case with a time step of 2 hr. Each panel corresponds to a level of mesh refinement. The lowest mesh refinement is at the top and displays the smallest level of instabilities, while the highest mesh refinement is at the bottom and displays numerous large instabilities in the first half of the simulation.

As an illustration, we have also run skin-layer simulations (class 2) using the surface temperature as the boundary condition, rather than directly used as a flux boundary condition. A comparison of the energy-conserving and non-energy-conserving simulations is shown in Fig. 14. The surface temperatures show RMSDs of $3.96$ and $2.16$ K and the phase change rates RMSDs of $3.6 \times 10^{-1}$ and $3.0 \times 10^{-1} \, \mathrm{kg \, m^{-2} \, h^{-1}}$ for the glacier and snowpack test cases, respectively. In general, the non-conservative scheme displays smaller daily variations of the surface temperature, with a less pronounced warming during the day (sometimes impending surface melt) and a less pronounced cooling at night.

For the non-conservative implementation, the inconsistency in $G$ can be expressed as an equivalent, and artificial, surface energy sink/source. For the glacier test case, this non-conservation of energy is equivalent to an additional energy flux with an average of $-14.5 \, \mathrm{W \, m^{-2}}$ (thus cooling the domain) and a standard-deviation of $123.5 \, \mathrm{W \, m^{-2}}$. In the snowpack test case, this corresponds to an additional energy flux with an average of $0.34 \, \mathrm{W \, m^{-2}}$ (warming the domain) and a standard deviation of $39 \, \mathrm{W \, m^{-2}}$. In both cases, the large value of the standard deviation compared to the average indicates that this "artificial" energy term displays large fluctuations, strongly affecting the simulations. Notably, in both cases the ablation of the glacier and the snowpack is reduced, with a decrease of respectively $40$ and $8 \, \%$ compared to the energy-conserving implementation.

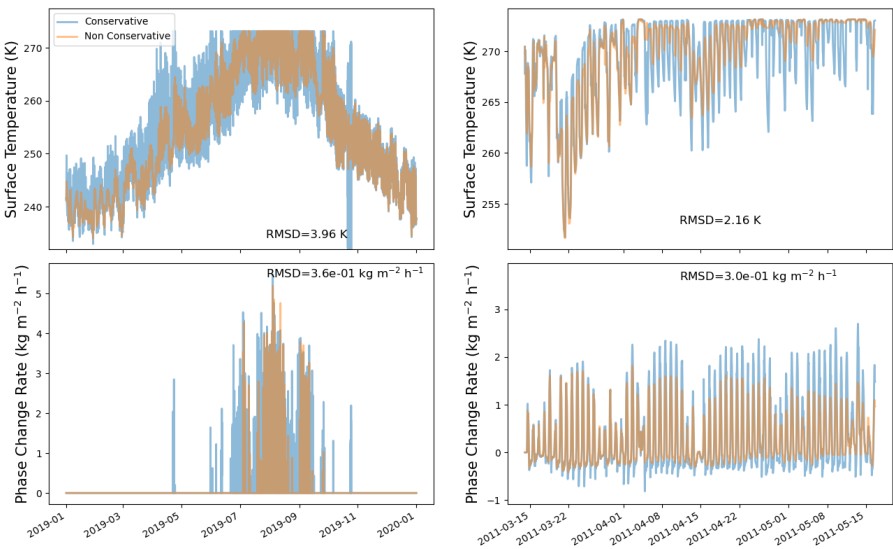

**Figure 14.** Comparison between the energy conservative and non-energy conservative skin layer numerical schemes. The left column corresponds to the glacier test case and the right column to the snowpack test case. Upper panels display the surface temperatures and the lower panels display the phase change rates.

## 7 Conclusions

Current implementations of the SEB in a finite volume framework can present one of the two limitations: (i) with the standard skin-layer formulation the SEB is solved sequentially with the internal heat budget, therefore creating a form of decoupling between the surface and the interior of the domain, or (ii) the SEB is integrated in the first cell, and there is no difference between this first cell temperature and the surface temperature. To circumvent these limitations, we derive a mathematical framework that includes both (i) an explicit surface, with a temperature different from that of the first cell below, and (ii) the tightly-coupled resolution of the surface and internal heat budgets including a potential surface melting. Notably, a unified treatment of melting and non-melting surface is proposed via the use of a fictitious variable playing the role of a switch between melting and non-melting conditions.

A specific Newton's method is also presented to robustly and efficiently solve the resulting non-linear system of equations. The robustness of the standard Newton's method is increased by using a truncation method, made to handle discontinuities in the equations. Furthermore, a reduction technique, based on the computation of a Schur complement, is presented so that the numerical cost of the proposed framework remains of the same order as that of the standard implementations for the same mesh. In particular, for a given mesh, the numerical cost is similar to that of models not explicitly having a surface and about 1.7 larger than that of the standard-skin layer formulation. It can therefore be implemented in existing snowpack and glacier models, while preserving their current numerical efficiency. Moreover, the reduction technique presented in this article can also be employed for other non-linear systems of equations (besides the energy budget treated here), by eliminating linearly-

dependent variables and reducing the size of the non-linear system to be iteratively solved, providing substantial gain when only a small portion of the discretized equations contains non-linearities.

Numerical test cases, corresponding to a snow-free glacier and a snowpack, have been performed in order to compare the results obtained with the different numerical treatments of the SEB. Mesh and time step convergence analyses show that combining a coupled treatment of the SEB with the explicit introduction of a surface results in an overall better accuracy when compared to the classical implementations. Notably, defining an explicit surface temperature enables the use of about $5$ times coarser meshes, compared to models using the temperature of the first cell as the surface temperature, for the same level of accuracy on temperature and phase change. Moreover, a tightly-coupled treatment of the SEB allows unconditional stability, while the standard skin-layer formulation can be unstable and displays large spurious oscillations with large time steps and small mesh sizes. Thus, while a bit more numerically costly, the formulation presented in this article can be used to overall reduce the numerical cost of a snowpack/glacier model through the use of larger time steps. Finally, we show that the conservation of energy could easily be broken when implementing a standard (loosely-coupled) skin-layer model. While this could be used as a technique to numerically stabilize the model, it leads to greatly deteriorated simulations.

## Appendix A: Matrix expressions and numerical cost of the coupled-surface scheme

### A1 Matrix expressions

Combing Eqs. (5), (6), and (10), the Newton scheme of the coupled-surface model proposed in this article can be written under block matrix form

$$
\left(\begin{array}{c|c} A_{\text{diag}} & A_{\text{up}} \\ \hline A_{\text{low}} & A_{\text{s}} \end{array}\right) \left(\begin{array}{c} T_{\text{int}} \\ U_{\text{s}} \end{array}\right) = \left(\begin{array}{c} B_{\text{int}} \\ B_{\text{s}} \end{array}\right) \tag{A1}
$$

with non-zero terms being

$$
A_{\text{diag}}(k,k) = \Delta z_{\text{k}} c_{\text{p}_{\text{k}}} + \Delta t \left( \frac{\lambda_{\text{k}+\frac{1}{2}}^{\text{harm}}}{\frac{\Delta z_{\text{k}}}{2} + \frac{\Delta z_{\text{k}+1}}{2}} + \frac{\lambda_{\text{k}-\frac{1}{2}}^{\text{harm}}}{\frac{\Delta z_{\text{k}}}{2} + \frac{\Delta z_{\text{k}-1}}{2}} \right) \tag{A2}
$$

$$
A_{\text{diag}}(k,k-1) = -\Delta t \frac{\lambda_{\text{k}-\frac{1}{2}}^{\text{harm}}}{\frac{\Delta z_{\text{k}}}{2} + \frac{\Delta z_{\text{k}-1}}{2}} \tag{A3}
$$

$$
A_{\text{diag}}(k,k+1) = -\Delta t \frac{\lambda_{\text{k}+\frac{1}{2}}^{\text{harm}}}{\frac{\Delta z_{\text{k}}}{2} + \frac{\Delta z_{\text{k}+1}}{2}} \tag{A4}
$$

$$
A_{\text{up}}(N-1,1) = A_{\text{low}}(1,N-1) = -\Delta t \frac{\lambda_{N-\frac{1}{2}}^{\text{harm}}}{\frac{\Delta z_{\text{N}-1}}{2} + \frac{\Delta z_{\text{N}}}{2}} \tag{A5}
$$

$$A_{\mathrm{s}}(1,1) = \Delta z_{\mathrm{N}} c_{\mathrm{pN}} + \Delta t \left( \frac{\lambda_{\mathrm{N}-\frac{1}{2}}^{\mathrm{harm}}}{\frac{\Delta z_{\mathrm{N}}}{2} + \frac{\Delta z_{\mathrm{N}-1}}{2}} + \frac{\lambda_{\mathrm{N}}}{\frac{\Delta z_{\mathrm{N}}}{2}} \right) \tag{A6}$$

$$A_{\mathrm{s}}(2,2) = \Delta t \left( \frac{\lambda_{\mathrm{N}}}{\frac{\Delta z_{\mathrm{N}}}{2}} \mathrm{d}_{\tau} T_{\mathrm{surf}} + L_{\mathrm{fus}} \mathrm{d}_{\tau} \dot{m} - \mathrm{d}_{\tau} H - \mathrm{d}_{\tau} L - \mathrm{d}_{\tau} LW_{\mathrm{out}} - -\mathrm{d}_{\tau} R \right) \tag{A7}$$

$$A_{\mathrm{s}}(1,2) = -\Delta t \frac{\lambda_{\mathrm{N}}}{\frac{\Delta z_{\mathrm{N}}}{2}} \mathrm{d}_{\tau} T_{\mathrm{surf}} \tag{A8}$$

$$A_{\mathrm{s}}(2,1) = -\Delta t \frac{\lambda_{\mathrm{N}}}{\frac{\Delta z_{\mathrm{N}}}{2}} \tag{A9}$$

$$B_{\mathrm{int}}(k) = \Delta z_{\mathrm{k}} c_{\mathrm{pk}} T_{\mathrm{k}}^{n-1} + \Delta t \mathrm{SW}_{\mathrm{int,k}} \tag{A10}$$

$$B_{\mathrm{s}}(1) = \Delta z_{\mathrm{N}} c_{\mathrm{pN}} T_{\mathrm{N}}^{n-1} + \Delta t \left( \mathrm{SW}_{\mathrm{int,N}} - \frac{\lambda_{\mathrm{N}}}{\frac{\Delta z_{\mathrm{N}}}{2}} \left( \mathrm{d}_{\tau} T_{\mathrm{surf}} \tau^{\mathrm{i}} - T_{\mathrm{s}}(\tau^{\mathrm{i}}) \right) \right) \tag{A11}$$

$$\begin{aligned}
B_{\mathrm{s}}(2) = \Delta t \Big( & SW_{\mathrm{net}}^{\mathrm{surf}} + LW_{\mathrm{in}} - \frac{\lambda_{\mathrm{N}}}{\frac{\Delta z_{\mathrm{N}}}{2}} \left( T_{\mathrm{s}}(\tau^{\mathrm{i}}) - \mathrm{d}_{\tau} T_{\mathrm{surf}} \tau^{\mathrm{i}} \right) - L_{\mathrm{fus}} \left( m(\tau^{\mathrm{i}}) - \mathrm{d}_{\tau} m \tau^{\mathrm{i}} \right) \\
& + \left( H(\tau^{\mathrm{i}}) - \mathrm{d}_{\tau} H \tau^{\mathrm{i}} \right) + \left( L(\tau^{\mathrm{i}}) - \mathrm{d}_{\tau} L \tau^{\mathrm{i}} \right) + \left( R(\tau^{\mathrm{i}}) - \mathrm{d}_{\tau} R \tau^{\mathrm{i}} \right) + \left( LW_{\mathrm{out}}(\tau^{\mathrm{i}}) - \mathrm{d}_{\tau} LW_{\mathrm{out}} \tau^{\mathrm{i}} \right) \Big)
\end{aligned} \tag{A12}$$

In the above expressions, $T_{\mathrm{k}}^{n-1}$ is the temperature of cell $k$ at the previous time step, $\mathrm{SW}_{\mathrm{int,k}}$ is the quantity of shortwave radiation absorbed in cell $k$, and $\tau^{\mathrm{i}}$ is the value of the fictitious variable $\tau$ at the start of the current non-linear iteration. The terms $T_{\mathrm{s}}(\tau^{\mathrm{i}})$, $H(\tau^{\mathrm{i}})$, etc, and $\mathrm{d}_{\tau} T_{\mathrm{surf}}$, $\mathrm{d}_{\tau} H$, etc, are the values of the surface temperature, sensible heat flux, etc, and their derivatives at the current $\tau^{\mathrm{i}}$ estimation.

Among the different partial derivatives, $\mathrm{d}_{\tau} H$ and $\mathrm{d}_{\tau} L$ can be difficult to analytically derive. For that, we first note that the chain rule yields $\mathrm{d}_{\tau} H = \mathrm{d}_{T_{\mathrm{s}}} H \mathrm{d}_{\tau} T_{\mathrm{s}}$, and $\mathrm{d}_{\tau} L = \mathrm{d}_{T_{\mathrm{s}}} L \mathrm{d}_{\tau} T_{\mathrm{s}}$. Then, for the expression of $H$ given in Appendix D we have:

$$\mathrm{d}_{T_{\mathrm{s}}} H = \rho_{\mathrm{a}} c_{\mathrm{p,a}} u \left( \mathrm{d}_{T_{\mathrm{s}}} C_{\mathrm{H}} (T_{\mathrm{a}} - T_{\mathrm{s}}) - C_{\mathrm{H}} \right) \tag{A13}$$

Moreover, the chain rule yields $\mathrm{d}_{T_{\mathrm{s}}} C_{\mathrm{H}} = \mathrm{d}_{\mathrm{Ri_b}} C_{\mathrm{H}} \mathrm{d}_{T_{\mathrm{s}}} \mathrm{Ri_b}$. In our case:

$$\mathrm{d}_{\mathrm{Ri_b}}C_{\mathrm{H}} = \frac{\kappa^2}{\ln\left(\frac{z}{z_0}\right)\left(\frac{z}{z_{0t}}\right)} \begin{cases} 0 & \text{if } \mathrm{Ri_b} < 0 \\ 50\mathrm{Ri_b} - 10 & \text{if } 0 \le \mathrm{Ri_b} < 0.2 \\ 0 & \text{if } 0.2 \le \mathrm{Ri_b} \end{cases} \tag{A14}$$

and

$$\mathrm{d}_{T_{\mathrm{s}}}\mathrm{Ri_b} = -\frac{g z_{\mathrm{a}}}{T_{\mathrm{a}} u^2} \tag{A15}$$

Similarly, for $L$, we have:

$$\mathrm{d}_{T_{\mathrm{s}}}L = \rho_{\mathrm{a}} L_{\mathrm{s}} u \left(\mathrm{d}_{T_{\mathrm{s}}}C_{\mathrm{E}}(q_{\mathrm{a}} - q_{\mathrm{s}}) - C_{\mathrm{E}}\mathrm{d}_{T_{\mathrm{s}}}q_{\mathrm{s}}\right) \tag{A16}$$

The derivative $\mathrm{d}_{T_{\mathrm{s}}}C_{\mathrm{E}}$ can be computed as the one of $C_{\mathrm{H}}$ through the chain rule and its dependence to $\mathrm{Ri_b}$. The derivative of $q_{\mathrm{s}}$ with respect to $T_{\mathrm{s}}$ can be easily obtained using the derivative of the saturated water vapor pressure, which is given by the Clausius-Clapeyron relation.

**A2   Numerical cost**

We see that the whole system of Eqs. (A1) is a tri-diagonal system of dimension $(N+1)\times(N+1)$, with $N$ the number of cells. Without a Schur-complement, the computation of $A^{-1}B$ can thus be solved with Thomas algorithm (Versteeg and Malalasekera, 2007) in $10N-1$ base operations (addition, subtraction, multiplication, and division) per non-linear iteration (neglecting the time spent assembling the matrices). We also note that $A_{\mathrm{diag}}$ is a tri-diagonal matrix, and thus Thomas algorithm also
applies. Moreover, we see that $A_{\mathrm{up}}$ and $A_{\mathrm{low}}$ are almost empty matrices, which simplifies the number of operations necessary to compute $A_{\mathrm{diag}}^{-1}A_{\mathrm{up}}$ and $A_{\mathrm{low}}A_{\mathrm{diag}}^{-1}A_{\mathrm{up}}$. Specifically, the Schur-complement technique used in this paper can be employed with $7N-9$ ($A_{\mathrm{diag}}^{-1}A_{\mathrm{up}}$, once per time step) + $10N-21$ ($A_{\mathrm{diag}}^{-1}B_{\mathrm{int}}$, once per time step) + 15 (assembly and solving of Schur-complement, once per iteration) + $2N$ (re-injection to compute $T_{\mathrm{int}}$, once per time step) steps, i.e. a total of $17N-6+15n_{\mathrm{it}}$ steps, with $n_{\mathrm{it}}$ the number of non-linear iterations. We see, that the advantage of the Schur-complement technique is that the
cost of performing non-linear iterations do not increase with the mesh resolution, yielding a smaller numerically cost than inverting the whole system for each non-linear iteration.

One may then wonder how the numerical cost of the scheme proposed in the article compares to the Class 1 and 2 models discussed in the paper. The Class 1 model (once a Schur-complement technique has been employed) as a similar numerical cost
as the proposed coupled-surface scheme approach, namely $17N-23+15n_{\mathrm{it}}$ steps. For a given mesh, it has one less degree of freedom as the coupled-surface scheme and is thus only marginally less costly. The Class 2 model is the least costly of all schemes discussed in the paper. Indeed, once the SEB and the surface temperature have been solved through scalar non-linear

iterations, it relies on a single tri-diagonal inversion of dimension $N \times N$, which can be done in $10N - 11$ steps. The ratio of the numerical cost of the scheme proposed in the article over that of the standard skin-layer is of about $1.7$.

## Appendix B: System size reduction for class 1 models

The size-reduction technique presented in Section 4.1.1 can also be employed for class 1 models, i.e. models where the SEB is integrated directly within the first cell and where the temperature of this first cell plays the role of the surface temperature. Such an implementation is used for our comparison in Section 5 as a way to speed up our implementation of a class 1 model.

As explained in Section 5, we made sure that for our resolution of class 1 model, the top-most cell does not overshoot the melt temperature, as it would bias the SEB. This is done by including the effect of first-order phase change in the top-most cell. For that, we use the energy content $h$ of the top cell as the prognostic variable, instead of its temperature. The discrete energy budget of the top cell thus writes:

$$\Delta_z h^{n+1} + \Delta t F_{\mathrm{SEB}} + \Delta t F = \Delta t Q + \Delta_z h^n \qquad \text{(B1)}$$

where $h^{n+1}$ and $h^n$ are the energy content at the end and start of the time step, $F_{\mathrm{SEB}}$ the net energy sum of the surface energy fluxes (taken positive if oriented towards the domain), $F$ the heat conduction flux exchanged with the cell below, $Q$ the volumetric internal heat source, and $\Delta t$ the time step size. The conduction flux $F$ is computed as the other conduction fluxes (Eq. (6)), simply noting that the temperature of the top cell is a non-linear function of its energy content $h$.

Combining all budget equations over the domain leads to a matrix system of the type:

$$\left( \begin{array}{c|c} A_{\mathrm{diag}} & A_{\mathrm{up}} \\ \hline A_{\mathrm{low}} & A_{\mathrm{s}} \end{array} \right) \left( \begin{array}{c} T_{\mathrm{int}} \\ \hline U_{\mathrm{s}} \end{array} \right) = \left( \begin{array}{c} B_{\mathrm{int}} \\ \hline B_{\mathrm{s}} \end{array} \right) \qquad \text{(B2)}$$

where $U_s = [T_{N-1}, h]$, and $A_{\mathrm{diag}}$, $A_{\mathrm{up}}$, $A_{\mathrm{low}}$ and the vector $B_{\mathrm{int}}$ are constant during the non-linear iterations. Therefore, the reduction technique presented in Section 4.1.1 applies and the unknown $U_s$ can be solved through the $2 \times 2$ non-linear system:

$$(A_{\mathrm{s}} - A_{\mathrm{low}} A_{\mathrm{diag}}^{-1} A_{\mathrm{up}}) U_{\mathrm{s}} = B_{\mathrm{s}} - A_{\mathrm{low}} A_{\mathrm{diag}}^{-1} B_{\mathrm{int}} \qquad \text{(B3)}$$

with only $A_{\mathrm{s}}$ and $B_{\mathrm{s}}$ to be re-assembled at each iteration.

## Appendix C: Finite Element Method scheme

In this paper, we focus on the FVM for spatial discretization. However, the heat budget equation could also be spatially discretized with the FEM. Indeed, the FEM naturally includes a node at the surface, and thus possesses a surface temperature,

which helps to tightly couple the SEB to the interior of the snowpack/glacier. This strategy is for instance employed in the SNOWPACK model (Bartelt and Lehning, 2002; Wever et al., 2020). Specifically, in SNOWPACK, the coupled SEB is intro-
715 duced as a top Robin boundary condition.

The goal of this appendix is to briefly present how the techniques presented in the main part of the manuscript (namely the use of fictitious variable and of a Schur-complement) can be used to implement a tightly-coupled FEM model.

## C1 Expression of the heat equation in FEM

We consider the mesh of the domain to be discretized into $N$ 1D elements (the direct equivalent of the cells in FVM) and thus of $N + 1$ nodes (the end-points of the elements). As classically done with FEM (Pepper and Heinrich, 2005), we assume the temperature field to be a linear combination of basis functions $\varphi_j$, i.e. $T(z,t) = \sum_{k=1}^{N} T_j(t)\varphi_j(z)$. Here, we use basic linear elements. In this framework, $T_j(t)$ corresponds to the nodal value of the temperature field (which evolves over time) and the basis functions $\varphi_j(z)$ are piece-wise linear functions, valued 1 at node $j$ and 0 at all other nodes. The standard Galerkin form
(Pepper and Heinrich, 2005) of the internal heat budget (Eq. (1)) is:

$$\forall i \quad \sum_j \mathrm{d}_t T_j \int_\Omega c_\mathrm{p} \varphi_j \varphi_i \mathrm{dL} + \sum_j T_j \int_\Omega \lambda \nabla\varphi_j \cdot \nabla\varphi_i \mathrm{dL} = \int_\Omega Q\varphi_i \mathrm{dL} + F_\mathrm{s}\varphi_i(\mathrm{s}) \tag{C1}$$

where $\Omega$ represents the domain of simulation, $F_\mathrm{s}$ is the energy fluxes entering at the top of the domain (i.e. $G$), and $\varphi_i(\mathrm{s})$ is the basis function $\varphi_i$ evaluated at top of the domain. We note that similarly to the FVM case, the temperature at the top of the domain presents a regime change whether the surface is melting or not. To handle this, we rely on the fictitious variable $\tau$,
i.e. $T_\mathrm{s} = T_\mathrm{s}(\tau)$. The vector of unknowns, denoted $U$, is thus composed of the internal temperatures and of the surface fictitious variable. Finally, we have not included any bottom energy flux to lighten the notation, but it could be included easily. Once temporally discretized with a Backward Euler scheme and linearized, the problem can be expressed in matrix form $AU^n = B$, with $A = (M + \Delta t K + \Delta t L)J_T$ and $B = MT^{n-1} + \Delta t Q + \Delta t F$ ($T^{n-1}$ being the vector of temperature from the previous time step), and

$$735 \quad M(i,j) = \int_\Omega c_\mathrm{p}\varphi_j\varphi_i \mathrm{dL} \tag{C2}$$

$$K(i,j) = \int_\Omega \lambda\nabla\varphi_j \cdot \nabla\varphi_i \mathrm{dL} \tag{C3}$$

$$L(N+1, N+1) = -\mathrm{d}_\tau SEB + L_\mathrm{fus}\mathrm{d}_\tau\dot{m} \tag{C4}$$

$$J_T(i,i) = \begin{cases} 1 & \text{if } i \leq N \\ d_\tau T_\text{s} & \text{else} \end{cases} \tag{C5}$$

$$Q(i) = \int_\Omega Q\varphi_i \text{dL} \tag{C6}$$

and

$$F(N+1) = SEB(\tau^i) - d_\tau SEB\tau^i - \dot{m} + L_\text{fus}\left(d_\tau \dot{m}\tau^i\right) \tag{C7}$$

where $SEB$ and $d_\tau SEB$ corresponds to the atmospheric fluxes in the SEB and their derivatives with respect to $\tau$ at the current iteration, and $\dot{m}$ and $d_\tau \dot{m}$ are the melting rate and its derivative at the current iteration. In the equations above, only the non-zero terms have been given.

As in the FVM case, this system is composed of a linear-part (the interior, corresponding to the first $N-1$ equations) and a non-linear part (the surface, corresponding to the last two equations). Its solving can thus be accelerated using a Schur-complement technique (Section 4.1.1) by breaking the matrix $A$ into four blocks: a constant $(N-1) \times (N-1)$ diagonal $A_\text{diag}$ block, a constant $(N-1) \times 2$ vertical $A_\text{up}$ block, a constant $2 \times (N-1)$ horizontal $A_\text{low}$ block, and a $2 \times 2$ diagonal block $A_\text{s}$ to be re-computed at each non-linear iteration.

## C2    The rest of the model

After solving the coupled heat budgets with FEM, we obtain a nodal temperature field. Since conserved quantities, such as energy or mass, are defined element-wise in snowpack/glacier FEM models (Bartelt and Lehning, 2002), the nodal temperature field needs to be converted into an element-wise energy field. We note that this also defines an element-wise temperature field, where the temperature of an element is simply the average of the nodal temperatures at its end. This element-wise energy field can then be used to simulate melt/refreeze, liquid water percolation, and to remesh the domain using the same routines as in FVM models.

Once all routines for a given time step have been performed, we are left with an element-wise temperature field that needs to be converted back to a nodal temperature field, as required for the FEM. However, this conversion is not straightforward. First, as we have $N$ element-wise temperatures to transform into $N+1$ nodal temperatures, the problem is not properly closed and an extra (arbitrary) constraint needs to be added. This could, for instance, be setting the surface temperature to the value

computed in the SEB. Furthermore, even after choosing an extra constraint to close the problem, the element-wise to node-wise transformation can produce spurious oscillations in the nodal field even if the element-wise field is monotonous (in other words, the transformation does not respect a form of discrete maximum principle; Ciarlet and Raviart, 1973). It is therefore not possible to derive an optimal scheme for this transformation that would (i) not modify the element-wise temperature field and (ii) not create spurious oscillations in the node-wise temperature field.

As spurious oscillations in the temperature field would affect the estimation of the temperature gradients that are used in snow-pack models to estimate metamorphism (e.g. Bartelt and Lehning, 2002; Vionnet et al., 2012), it seems preferable to rather allow the modification of the element-wise temperature field. That being said, such a strategy implies a spatial re-distribution of energy between elements that is not motivated by any underlying physical mechanism. We note that the SNOWPACK model handles this element to node transformation during a phase change step after the liquid percolation scheme, and does so without creating large spurious temperature oscillations.

Unfortunately, it is not possible to directly implement the SNOWPACK scheme in our toy-model, as the sequential treatment is not the same. Moreover, we did not manage to derive a scheme that performs this element to node transformation without affecting the surface temperature. Thus, in our numerical simulations, the FVM and FEM models yield different results. In the absence of an analytical solution, a direct comparison of the FEM and FVM implementations remains impossible.

## Appendix D: Expression of turbulent fluxes used in this work

The computations of the turbulent fluxes used in this work are based on those provided by Sauter et al. (2020), with slight modifications. The sensible and latent heat fluxes, $H$ and $L$, are taken as:

$$H = \rho_a c_{p,a} C_H u (T_a - T_s) \tag{D1}$$

and

$$L = \rho_a L_s C_E u (q_a - q_s) \tag{D2}$$

with $\rho_a$ the density of air, $c_{p,a}$ the heat capacity of air at constant pressure, $u$ the wind velocity (at a given height), $L_v$ the latent heat of sublimation of water, $T_a$ and $q_a$ the temperature and specific humidity of the air, $T_s$ and $q_s$ the temperature and specific humidity of the surface, assuming the saturation of vapor, and $C_H$ and $C_E$ two coefficients given by:

$$C_H = \frac{\kappa^2}{\ln\left(\frac{z}{z_0}\right)\left(\frac{z}{z_{0t}}\right)} \psi(\mathrm{Ri_b}) \tag{D3}$$

and

$$C_{\mathrm{E}} = \frac{\kappa^2}{\ln\left(\frac{z}{z_0}\right)\left(\frac{z}{z_{0q}}\right)}\psi(\mathrm{Ri_b}) \tag{D4}$$

with $\kappa = 0.41$ the von Kármán constant, $z_0$ the aerodynamic roughness length, $z_{0q}$ and $z_{0t}$ taken 1 and 2 orders of magnitude smaller than $z_0$, respectively (Sauter et al., 2020), and $\psi$ a stability correction factor. Specifically, we take $\psi$ as:

$$\psi(\mathrm{Ri_b}) = \begin{cases} 1 & \text{if } \mathrm{Ri_b} < 0 \\ (1 - 5\mathrm{Ri_b})^2 & \text{if } 0 \leq \mathrm{Ri_b} < 0.2 \\ 0 & \text{if } 0.2 \leq \mathrm{Ri_b} \end{cases} \tag{D5}$$

with $\mathrm{Ri_b}$ the bulk Richardson number:

$$\mathrm{Ri_b} = \frac{g}{T_{\mathrm{a}}}\frac{(T_{\mathrm{a}} - T_{\mathrm{s}})\,z_{\mathrm{a}}}{u^2} \tag{D6}$$

with $z_{\mathrm{a}}$ the height at which the air temperature measurement is performed.

There are two main differences compared to the expression of the turbulent fluxes given in (Sauter et al., 2020). First, in
Sauter et al. (2020), the transition between the unstable and stable correction factor $\psi$ is taken for $\mathrm{Ri_b} = 0.01$, while we take it for $\mathrm{Ri_b} = 0$. This choice is made to ensure the continuity of the stability factor, and thus of the turbulent fluxes, as a function of $T_{\mathrm{s}}$. In the presence of a discontinuity, it can indeed happen that the SEB does not have a solution in terms of $T_{\mathrm{s}}$, and the surface temperature is no longer defined in this case. Secondly, for the expression of the latent heat flux, we simply keep the latent heat of sublimation $L_{\mathrm{s}}$ and do not replace it with the latent heat of vaporization $L_v$. Again, the goal is to avoid discontinuities in
the SEB as a function of $T_{\mathrm{s}}$ so that the problem remains mathematically well-posed. This approach is, for instance, used in the Crocus model (personal communication; M. Lafaysse). Another strategy could be to fix the latent heat to either its sublimation or vaporization value, depending on the initial state of the surface.

**Appendix E: Stability Analysis**

Here, we present the derivation of the criteria for the numerical stability of the different numerical schemes presented in the
paper. We follow the proof classically used to show the (un)conditional stability of the Forward and Backward Euler method (Butcher, 2008). Notably, the proof relies on a linearized version of the system of equations. As the system needs to be linearized, we cannot account for the potential melting of the surface. Under this consideration, the atmospheric fluxes in the SEB (long-wave radiations, turbulent fluxes, etc) are simply expressed as a linear function of the surface temperature $T_{\mathrm{s}}$, i.e. as $fT_{\mathrm{s}} + b$, where $f$ and $b$ are constant scalars expressed in $\mathrm{J\,s^{-1}\,m^{-2}\,K^{-1}}$ and in $\mathrm{J\,s^{-1}\,m^{-2}}$, respectively.
Also, for simplicity, we consider a system composed of only one cell and its surface. The problem could be generalized to more

cells, but it would make the computation more cumbersome and is not crucial as we are considering numerical instabilities that develop in the vicinity of the surface.

### E1   Standard skin-layer formulation (Class 2)

To compute the surface temperature $T_{\mathrm{s}}^{n+1}$ at time step $n+1$, we use the discretized Surface Energy Balance (SEB):

$$f T_{\mathrm{s}}^{n+1} + b + \frac{2\lambda}{\Delta z}\left(T_{\mathrm{s}}^{n+1} - T_{\mathrm{i}}^{n}\right) = 0 \tag{E1}$$

where the first two terms corresponds to the sum of outgoing/incoming atmospheric fluxes, and the last term to the subsurface heat conduction flux. Here, $\lambda$ is the thermal conductivity of the internal cell and $\Delta z$ its thickness. Note that the internal

temperature $T_{\mathrm{i}}^{n}$ is taken from the previous time step. To compute the internal temperature at time step $n+1$, we use the heat budget of the internal cell:

$$\Delta z c_p T_{\mathrm{i}}^{n+1} + \Delta t \frac{2\lambda}{\Delta z}\left(T_{\mathrm{i}}^{n} - T_{\mathrm{s}}^{n+1}\right) = \Delta z c_p T_{\mathrm{i}}^{n} \tag{E2}$$

where the second term of the LHS is the opposite of the subsurface conduction flux appearing in the SEB (for energy conservation), and $c_p$ is the heat capacity of the internal cell. The two above equations can be expressed in matrix form

$M U_{n+1} = N U_n + B$, with $U_n$ the solution vector $[T_{\mathrm{s}}, T_{\mathrm{i}}]^T$ at the $n^{\text{th}}$ time step and

$$M = \begin{bmatrix} 1 & 0 \\ -\frac{2\Delta t \lambda}{c_p \Delta z^2} & 1 \end{bmatrix} \tag{E3}$$

$$N = \begin{bmatrix} 0 & \frac{2\lambda}{2\lambda + \Delta z f} \\ 0 & 1 - \frac{2\Delta t \lambda}{c_p \Delta z^2} \end{bmatrix} \tag{E4}$$

and $B = [-\frac{\Delta z b}{\Delta z f + 2\lambda}, 0]^T$. We thus have, $U_{n+1} = Q U_n + M^{-1} B$, with

$$Q = M^{-1} N = \begin{bmatrix} 0 & \frac{2\lambda}{2\lambda + \Delta z f} \\ 0 & 1 - \Delta t \frac{2\lambda}{c_p \Delta z^2} \frac{\Delta z f}{2\lambda + \Delta z f} \end{bmatrix} \tag{E5}$$

By recursion, it follows that $U_n = Q^n U_0 + M^{-n} B$. The numerical scheme is deemed stable if $\lim_{n \to \infty} Q^n = 0$. This is achieved if:

$$\left| 1 - \Delta t \frac{2\lambda}{c_p \Delta z^2} \frac{\Delta z f}{2\lambda + \Delta z f} \right| < 1 \tag{E6}$$

which after some computation yields a criterion of the time step $\Delta t$:

$$\Delta t < \Delta t_{\text{crit}} = \frac{c_p \Delta z}{\lambda} \frac{2\lambda + \Delta z f}{f} \tag{E7}$$

The (linearized) standard skin-layer is thus only conditionally stable. The stability criterion is relaxed with increasing heat capacity ($c_p$) and increasing cell size ($\Delta z$), and is made more restrictive with increasing thermal conductivity ($\lambda$) or if the SEB is more sensitive to changes in the surface temperature ($f$ term).

## E2  Coupled-surface formulation

Similarly, for a one cell system, the coupled-surface equations, after linearization, write:

$$f T_{\text{s}}^{n+1} + b + \frac{2\lambda}{\Delta z} \left( T_{\text{s}}^{n+1} - T_{\text{i}}^{n+1} \right) = 0 \tag{E8}$$

for the SEB, and

$$\Delta z c_p T_{\text{i}}^{n+1} + \Delta t \frac{2\lambda}{\Delta z} \left( T_{\text{i}}^{n+1} - T_{\text{s}}^{n+1} \right) = \Delta z c_p T_{\text{i}}^{n} \tag{E9}$$

for the cell's heat budget. These two equations can be cast into the matrix form $MU_{n+1} = NU_n + B$, with $B = [-\frac{\Delta z b}{\Delta z f + 2\lambda}, 0]^T$,

$$M = \begin{bmatrix} 1 & \frac{-2\lambda}{2\lambda + \Delta z f} \\ -\frac{2\Delta t \lambda}{c_p \Delta z^2 + 2\lambda \Delta t} & 1 \end{bmatrix} \tag{E10}$$

and

$$N = \begin{bmatrix} 0 & 0 \\ 0 & \frac{c_p \Delta z^2}{c_p \Delta z^2 + 2\lambda \Delta t} \end{bmatrix} \tag{E11}$$

We thus have $U_n = Q^n U_0 + M^{-n} B$, with:

$$Q = \begin{bmatrix} 0 & \frac{2\lambda}{2\lambda + \Delta z f} \frac{c_p \Delta z^2}{c_p \Delta z^2 + 2\lambda \Delta t} \\ 0 & \frac{c_p \Delta z^2}{c_p \Delta z^2 + 2\lambda \Delta t} \end{bmatrix} \tag{E12}$$

The numerical scheme is deemed stable if $\lim_{n \to \infty} Q^n = 0$. This is always achieved, as $\frac{c_p \Delta z^2}{c_p \Delta z^2 + 2\lambda \Delta t} < 1$. Thus, the surface-coupled scheme is unconditionally stable.

### E3 Non-conservative skin-layer formulation

For the non-conservative skin-layer formulation (see Section 6.4), we start with the linearized discrete SEB:

$$fT_s^{n+1} + b + \frac{2\lambda}{\Delta z}\left(T_s^{n+1} - T_i^n\right) = 0 \tag{E13}$$

Using the surface temperature $T_s^{n+1}$ as a Dirichlet condition for the internal energy budget, we thus have

$$\Delta z c_p T_i^{n+1} + \Delta t \frac{2\lambda}{\Delta z}\left(T_i^{n+1} - T_s^{n+1}\right) = \Delta z c_p T_i^n \tag{E14}$$

These two equations can be cast into the matrix form $MU_{n+1} = NU_n + B$, with $B = [-\frac{\Delta z b}{\Delta z f + 2\lambda}, 0]^T$,

$$M = \begin{bmatrix} 1 & 0 \\ -\frac{2\Delta t \lambda}{c_p \Delta z^2 + 2\lambda \Delta t} & 1 \end{bmatrix} \tag{E15}$$

and

$$N = \begin{bmatrix} 0 & \frac{2\lambda}{2\lambda + \Delta z f} \\ 0 & \frac{c_p \Delta z^2}{c_p \Delta z^2 + 2\lambda \Delta t} \end{bmatrix} \tag{E16}$$

We thus have $U_n = Q^n U_0 + M^{-n} B$, with:

$$Q = \begin{bmatrix} 0 & \frac{2\lambda}{2\lambda + \Delta z f} \\ 0 & X \end{bmatrix} \tag{E17}$$

where $X = \frac{2\lambda \Delta t \frac{2\lambda}{2\lambda + \Delta z f} + c_p \Delta z^2}{2\Delta t \lambda + c_p \Delta z^2}$. The scheme is deemed stable if $|X| < 1$.

As $\frac{2\lambda}{2\lambda + \Delta z f} < 1$, we always have that $2\lambda \Delta t \frac{2\lambda}{2\lambda + \Delta z f} + c_p \Delta z^2 < 2\Delta t \lambda + c_p \Delta z^2$, and thus that the scheme is unconditionally
stable. That being said, we recall that this scheme is not energy conservative and can lead to large errors.

### E4 No-surface formulation (Class 1)

Finally, we note that the linearized No-surface formulation corresponds to a classic heat equation with a Backward Euler time integration. As demonstrated elsewhere in the literature (e.g. Butcher, 2008), it is unconditionally stable.

*Code and data availability.* The source files of the code and the forcing data are provided at https://doi.org/10.5281/zenodo.10426228.

*Author contributions.* The research was designed by KF, JB and MD. MD acquired funding. The numerical models were developed by KF with the help of JB. The numerical simulations were performed by KF with the help of FB. FB and MD provided data for the simulations. The manuscript was written by KF with the help of all co-authors.

*Competing interests.* The authors declare having no competing interests.

*Acknowledgements.* We are thankful to the authors of Potocki et al., (2022) for making the meteorological forcings used in their study available. We acknowledge Marion Réveillet, Matthieu Lafaysse, Isabelle Gouttevin for the fruitful discussions on surface energy balance modeling and we acknowledge Clément Cancès for those on FVM modeling and fictitious variables. KF, JB and MD have received funding from the European Research Council (ERC) under the European Union's Horizon 2020 research and innovation program (IVORI, grant no. 949516). CNRM/CEN is part of Labex OSUG (ANR10 LABX56). We thank Richard Essery, Michael Lehning, and the two anonymous referees for their helpful reviews of the article, as well as Danilo Mello for editing it.

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
