# Peer review of "A novel numerical implementation for the surface energy budget of melting snowpacks and glaciers"

_EGUsphere, 2023_

## Referee Comment (RC4)

Review "A novel numerical implementation for the surface energy budget of melting snowpacks and glaciers" by K. Fourteau, J. Brondex, F. Brun and M. Dumont

Review by M. Lehning

General:

The paper presents a review on how to numerically implement the surface energy budget into a certain class of snow and ice models. The paper is very well written and in general presents the material in a clear manner. It is overall considered to be a useful contribution to the scientific community dealing with snow and ice modelling despite its rather theoretical setting, in which conclusions on existing snow and ice models are only possible in a limited way.

In this context, it is mandatory that existing snow and ice models that have schemes that come close to the solution presented here are discussed in sufficient detail. In particular, since for example SNOWPACK uses a finite element method (FEM), for which the nodal temperature is explicitly solved at the surface, it already achieves both aspects of the paper, an explicit surface and a tight coupling with internal heat transfer merely by construction of the FEM. This is true for the original version of SNOWPACK, which is now more than 20 years old. Moreover, the statement in l.81 is not a fair representation of the current state of snow models, since also efforts have been made to implement a coupled solver in SNOWPACK that does not generate temperature overshoots. This was crucial for sea ice simulations, where an additional complexity is created by the fact that the melting point of the snow and ice is a function of salinity, and that salinity in turn is impacted by the phase changes. This means that a simple approach of allowing overshoots to occur and then setting back the temperature to fusion value is not suitable any longer. This has been presented in Wever et al. (2020) and should be discussed in the current paper. The proper acknowledgment of the state of art is necessary and as a consequence limits the novelty of the proposed approach here. It is not acceptable to say "we don't discuss FEM models" as the authors do. This neglect is even more surprising since an overlapping group of authors proposes in another paper to use the FEM method for snow modelling (Brondex et al., 2023).

A second major point to address is the inconsistency and incompleteness with respect to the phase change (fusion) implementation as suggested. If I understand the set-up correctly, you explicitly implement the fusion process at the surface and keep the temperature solution at the phase change temperature with your variable switching formulation supported by the truncation method. But you don't do so below the surface, which generates an inconsistency for the sub-surface heat flux. For example, for the case of shortwave penetration into snow and ice, you would generate temperatures above the melt temperature below the surface, which would lead to an upwards heat flux towards the surface, which is at the melt temperature. But heat would flow downwards in reality. This inconsistency is not even mentioned in section 6.4 and probably has consequences for energy conservation. While the tight coupling and explicit surface are sufficiently investigated with sensitivity cases in the paper, the same needs to be done for this fusion treatment. The effect needs to be quantified and compared to the more classical "overshoot" solution.

Minor comments:

1) At least I am more used to the terms "melt" temperature and "heat" capacity instead of "fusion" and "thermal".
2) Eq. (3) does not contain heat advection by precipitation.
3) l. 108: Not true, SNOWPACK does not do a separate SEB, see above.
4) l. 126: "result" not results.
5) l. 284: "equation" not equations.
6) I don't understand the argument here: "Note that the method used to downscale the data does not guarantee physical consistency of the variables. This allows us to take into account shortwave, longwave and turbulent energy fluxes at the top of our domain".
7) Figures 3,4: These uncertainties should be discussed in light of typical snow/ice model errors.
8) l. 438: why "model 2" now, not clear?
9) l. 450 ff. should the reference not be a hundreds (900) of seconds consistent with typical time steps used?
10) l. 460: should it be "worse" instead of better?
11) l. 491: can you explain the deterioration?

References:

Brondex, J., Fourteau, K., Dumont, M., Hagenmuller, P., Calonne, N., Tuzet, F., and Löwe, H.: A finite-element framework to explore the numerical solution of the coupled problem of heat conduction, water vapor diffusion and settlement in dry snow (IvoriFEM v0.1.0), Geosci. Model Dev. Discuss., 2023, 1–50, https://doi.org/10.5194/gmd-2023-97, 2023.

Wever, N., Rossmann, L., Maaß, N., Leonard, K. C., Kaleschke, L., Nicolaus, M., and Lehning, M.: Version 1 of a sea ice module for the physics-based, detailed, multi-layer SNOWPACK model, Geosci. Model Dev., 13, 99–119, https://doi.org/10.5194/gmd-13-99-2020, 2020).

---

## Author Comment (AC1)

We are thankfull Richard Essery for his constructive review. Please find below our point by point response to the review. The comment of the referee are shown in blue and our response in black below. Proposed modifications of the manuscript are shown in green with page and line numbering corresponding to the preprint version of the article.

I greatly enjoyed reading this paper. The method for efficiently coupling the nonlinear surface energy balance to the linear subsurface heat conduction is a clever piece of matrix algebra, but it is not just that; it directly relates to a point of contention in the lively interactive discussions of Brun et al. (2022) and Potocki et al. (2022) concerning the mass balance of the Everest South Col Glacier.

There are limitations, however. Many processes are generally handled sequentially in snow models (Clark et al. 2015), but this paper only couples two of them. Only idealized test cases are shown and not full model performance in real applications.

We agree with the reviewer that two of the limitations of this work (which were also pointed out by other reviewers and the editor) are that: (i) we only tightly-couple two processes (the SEB and the internal heat equation) and leave others (such phase changes or liquid water percolation) sequentially treated and (ii) that we only treat idealized test cases. We were also aware of this potential limitation when doing this study, and wondered if more realistic cases should be analyzed. We eventually decided to leave them out.

Our motivation behind this choice is to allow to focus on a single topic, namely the efficient numerical coupling of the SEB to the internal heat budget in a FVM framework. We worry that introducing other tight-couplings (such as phase changes while solving the heat equation) might make the role of coupling the surface and internal energy budgets less clear, and thus renders this point less-readily available for current FVM models, such as Crocus or COSIPY. Likewise, we decided to focus on test cases without comparisons to direct observations, as it is not be possible to decipher errors due to the numerical implementations (which is the focus of our paper) from errors due to the assumed physics, parametrizations, and forcings (which we do not and cannot not address in this study). Therefore, we think that to meaningfully analyze numerical implementations in terms of cost, accuracy and robustness, the use of simplified test cases is appropriate. We however agree that the test cases should not be too unrealistic if we want their results to be informative of how a numerical scheme might behave in a realistic settings. That it is why we have used realistic forcings and initial conditions.

We now specify our intention more clearly in the text, and clearly explain that our simplifying assumptions are meant to ease the comparison of the numerical implementations of the surface-internal energy budgets, but that our toy-model should not be viewed as proper a snowpack/glacier model as many important components are lacking.

**P12 - L330**
*"Two simple examples, showcasing the differences between numerical treatments, are presented below. We note that these simulations cannot be considered as fully realistic simulations of a snowpack or glacier surface, as many processes, such as the deposition of atmospheric precipitation (rain or snow) or mechanical settling, are lacking. The goal is rather to provide a simplified setting in which the impact of the numerical implementation of the SEB can be analyzed. In the same idea, we do not attempt to compare the simulation results to field observations. Indeed, it would not be possible to decipher errors due to the numerical discretization (the focus of this paper) from errors due to the assumed physics, parametrizations and atmospheric forcings. Nonetheless, in order for*

*the results to still be informative of how a given numerical implementation might behave in a more realistic setting, we use realistic atmospheric forcing, initial conditions, and physical parametrizations. The first simulation is meant to highlight the behavior of the numerical models when simulating the SEB on a snow-free glacier. The second one focuses on the impact of the model implementations on the simulation of the energy budget of a seasonal snowpack, during the melting period."*

From the test case results, I could take contrary conclusions that the added complexity of coupling is not needed and the standard skin-layer formulation is fine as long the time step is not made too large, which is well known ("not too large" could still be prohibitively small for thin layers, though).

Indeed, very reasonable results can be obtained with the standard skin-layer formulation as long as the time step is kept short enough to avoid instabilities. The same conclusion could be made for the Class 1 model (no surface), as long as the top cell is kept thin enough. As all models solve the same equations, they converge to the "true" solution when the spatial and temporal resolution are refined, and the tightly-coupled approach is not expected to yield a different solution.

However, we believe the property of the tightly-coupled approach to accept both large time steps and mesh sizes, while keeping a similar numerical cost, motivates it use over the standard approaches. A numerical stability analysis is provided at this end of this response that shows that the coupled-surface scheme is unconditionally stable, contrary to the standard-skin layer formulation, and thus does not to require the implementation of an adaptative time step strategy. This is now discussed more clearly in the revised manuscript and the numerical stability analysis provided in an appendix.

**P21 - L503**
*"The unstable nature of class 2 models can be shown with a linear stability analysis, provided in Appendix E. Such analysis shows that class 2 models are only conditionally stable, and confirm that instabilities are favored in the case of large time steps and small mesh sizes. We stress that these oscillations can appear even if the time integration of the internal energy budget relies on the Backward Euler method, known for its robustness against instabilities (Fazio, 2001, Butcher, 2008). Our understanding is that the sequential treatment of the standard skin-layer formulation breaks the implicit nature of the time integration by using "lagged" (in other words, explicit rather than implicit) terms. This, combined with the fact that the surface layer does not possess any thermal inertia and that its temperature can thus vary rapidly in time, permits large temperature swings if the time step is too large or the mesh size too small. On the other hand, it can be shown that the two schemes with a tightly-coupled SEB are unconditionally stable (Appendix E), in agreement with the absence of oscillations in their simulations. Notably, the unconditional stability of the coupled-surface scheme proposed in this article entails that the model does not need an adaptive time step size strategy depending on the mesh size. This ensures that it remains robust, regardless of the time step and mesh size.*

**P26 - L563**
*"Moreover, a tightly-coupled treatment of the SEB allows unconditional stability, while the standard skin-layer formulation can be unstable and displays large spurious oscillations with large time steps and small mesh sizes."*

Specific comments:

Author list
"Brun Fanny" might like to have her name turned around.
We put Fanny's name in the good order.

Introduction

I don't recommend writing a comprehensive review of SEB formulations, but only giving recent examples of applications of a skin layer and no examples using a finite surface layer in the introduction, rather than original model development papers, gives a distorted view. An uncoupled skin layer has been in use for snow models at least as far back as Yamazaki and Kondo (1990). There is a snow surface layer temperature in Anderson (1968).
We added older model development papers in the revised manuscript. Notably we now provide references when discussing finite-top-layer models.

**P1 - L20**
*"To reach this goal, the representation and evolution of the thermodynamical state (that is to say temperature profiles and phase changes) of snowpacks and glaciers are implemented in most numerical snowpack/glacier models (e.g. Anderson, 1976, Brun et al. 1989, Jordan, 1991, Bartelt and Lehning 2002, Liston and Elder, 2006, Vionnet et al. 2012, Sauter et al., 2020)."*

**P2 - L40**
*"On the other hand, some FVM implementations do not define a specific temperature associated with the surface, but rather use the temperature of the top-most numerical layer of the domain (i.e. the top layer of the simulated snowpack/glacier) for solving the SEB (Anderson, 1976, Brun et al., 1989, Jordan, 1991, Vionnet et al., 2012, van Kampenhout et al., 2017)."*

22- There are many "numerical models" that are not snowpack/glacier models.

We reformulated the sentence to clearly state that by "most numerical models", we want to refer to snowpack/glacier models.

**P1 - L20**
*"To reach this goal, the representation and evolution of the thermodynamical state (that is to say temperature profiles and phase changes) of snowpacks and glaciers are implemented in most numerical snowpack/glacier models "*

32- The surface energy balance is described as "profoundly non-linear". Actually, this is a pretty benign nonlinearity in the field of nonlinear equations; it does not have multiple or chaotic solutions.
We removed the word "profoundly" to only state the problem is non-linear, and hence might requires some iterations for the proper solution to be computed.

49- The "infinitely small horizontal layer" would be better described as infinitely thin.
We replaced "small" with "thin".

While Eq. (1) is more generally applicable, it could already be emphasized that this is invariably implemented as a 1D model with T a function of z.

Consistently, with the remark of Reviewer 3, we now state that while the equation remains valid in 3D, we use it in a 1D set-up only as transitionally done in snowpack/surface-glaciers models.

**P3 - L70**

*"In this article, we assume that the snowpack/glacier can be represented as 1D column, and therefore Eq. (1) should be understood as 1D equation."*

I think that there will be very few exceptions to this "usually" of allowing snow temperature to exceed the fusion point before calculating melt, but there are examples of models with phase changes over a temperature range in Albert (1983) and Dutra et al. (2010).

As also pointed out by the review of Michael Lehning, several strategies have been proposed to handle phase change in snowpack/glaciers models. We modified the revised manuscript to clearly state that we rely on the method of exceeding the fusion point and then restoring thermodynamic equilibrium as it employed in the majority of snowpack/glacier models, but that alternatives exist. We now also stress that this method of "overshooting" is a form a sequential treatment, to which better treatments have been proposed in the recent literature. Building on this idea, we are currently working on the efficient tightly-coupled resolution of all internal thermodynamic processes, and will address it in a future work.

**P3 - L82**

*"In this article, we follow this simple scheme as it is commonly employed in snowpack and glacier models. That being said, other, more complex, strategies have been proposed in the literature. This notably includes the use of a finite temperature-range over which melt/freezing occurs (e.g. Albert, 1983, Dutra et al., 2010), including melt/refreeze as an additional energy source term (e.g. Bartelt and Lehning, 2002, Wever et al., 2020), or the use of enthalpy as the prognostic variable (e.g. Meyer and Hewitt, 2017, Tubini et al., 2021)."*

We have also estimated the sensitivity of our results to the treatment of these phase changes. We found that the conclusions of the article concerning the accuracy and stability of the different SEB schemes hold with a different treatment of phase changes (graphs provided in the response to the review of Michael Lehning). We now address this point in the revised manuscript:

**P12 - L329**

*"Also, as some of the current snowpack and glacier models include the effect of internal phase-change while solving the internal heat equation (e.g. Bartelt and Lehning, 2002, Meyer and Hewitt, 2017), we quantified the sensitivity of our results to this specific treatment of melt/freeze. For that, we have also implemented versions of our three models that include such internal phase-changes in the heat equation."*

**P16 - L441**

*"Finally, using the versions of the models including phase-changes in the heat equation, we quantified the sensitivity of these observations to the treatment of the melt/refreeze. While the simulated temperature sometimes differ from our basic implementations (especially in the snowpack test case where melt occurs internally), the general behavior of the models, including the potential presence of instabilities in the Class 2 models, remain unchanged."*

*P20 - L493*
*"Finally, using the versions of the models including phase-changes in the heat equation, we verified that the conclusions of this convergence analysis remain valid in the case of a different treatment of the internal phase-changes"*

136- "SNTHERM (Jordan, 1991), Crocus (Vionnet et al., 2012)"
The typo is corrected.

146- Another step is required if the calculated melt exceeds the available snow mass.
It is indeed important that the local calculated melt does not exceed the available snow mass. In our implementation, if the local melt exceeds the snow mass, layers are locally merged until the melt falls behind the available snow mass. This is now specified in the manuscript.

**P12 - L323**
*"This remeshing step is also used to ensure that the melt of a layer cannot exceed its ice content. If such a case is encountered, the layer is merged with one of its neighbors before attempting melting. If the total melt exceeds the total mass, the simulations should be stopped. However, this last case did not arise in the simulations presented here."*

234- LWout and H are given as examples of fluxes that are nonlinear in the surface temperature; L should also be mentioned as intrinsically nonlinear. H as defined by Eq. (B1) is only nonlinear if $C_H$ is a function of surface temperature. It is, through $Ri_b$ here, but models often neglect this nonlinearity because of the complexity of the resulting derivatives; it is not clear if that is done here. A supplement giving the elements of the Jacobian might be a useful addition.
We added L in the list of SEB terms that are non-linear with respect to the temperature.

In our implementation, we take into account the dependence of $C_H$ to the surface temperature and include its impact on the Jacobian of the system (in order to have a true Newton method with quadratic convergence). Note that not taking this dependence in the Jacobian does not modify the solution of the non-linear system, but only the sequence of iterations and the convergence rate toward this solution.

To make our model readily-available we explicitly wrote the terms of the Jacobian in the new Appendix A.

**P11 - L275**
*"The expressions of the matrices forming the block system are given in Appendix A, including the derivatives necessary for Newton's method."*

261- I understand the problem, but I don't understand the benefit of returning the solution to the vicinity of the discontinuity.
The SEB should have a unique solution, but the Newton method is not guaranteed to find it. It can get trapped in a cycle of states around the solution. This situation can be diagnosed from the SEB, but I think that most models just give up and select the last iteration. Does the modified Newton method avoid this problem?
Yes, the goal of the truncation method is precisely to avoid the iterations to be stuck in a loop or to diverge and is quite adapted for the solving of the SEB with a fictitious variable. We've made a Figure as illustration below (the SEB non-linearity has been exaggerated for the illustration). In the case of the standard Newton method without truncation, the break in the slope can send the iterations far from the solution (or into loops depending of

the configuration). In the truncation case, the iteration is moved to the orange point after two truncations. At this point, the Newton scheme can converge normally to the solution.

**P9 - L251**
*"The idea behind truncation is that the Jacobian  (i.e. the derivative of the equations with respect to the unknowns to be solved for) computed on one side of a derivative discontinuity does not apply on the other side, and can therefore perturb the convergence towards the solution, typically leading to an endless iteration loop."*

[Figure]

**Figure –** *Solving of a non-linear SEB with and without a truncation in Newton's method. In the truncation case, the estimation is brought from the red point to the orange point after two successive truncations.*

Note that Newton's method can be made even more robust by applying a truncation at the inflection points. However, this was not done in our case, as the SEB does not displays such inflection point with respect to the surface temperature.

265- Another solution in use, with its own numerical errors, is to linearize the SEB and solve it in one step without iteration (e.g. Best et al. 2011). This is essentially the Penman-Monteith method.
Equation (11) and following
Indeed, some models only solve the linear system with one iteration (for instance Crocus). However in this case, the obtained solution is not the actual backward Euler solution and does not have all its properties. We mention this point in the article.

**P9 - L236**
*"We also note that some models made the choice of performing only a single iteration to solve this linear system of equations (with sometimes an extra iteration to handle specific cases, such as surface melting). However, we chose here to perform multiple iterations, in order to obtain the actual Backward Euler solution."*

Be consistent in making diag, up and low superscripts or subscripts.
We corrected the manuscript consistently, with all diag, up, and low being subscripts.

284- "The above equation" is Eq. (13).
We modified the text to state that the "above equation" refers to Eq. (13) and that it allows one to solve the first temperature, as if they were solved with the complete system of Eq. (11).

**P11 - L284**

*"The system of Eqs.(13) is a 2x2 non-linear system where only $A_s$ and $B_s$ need to be re-assembled at each non-linear iteration and whose solution for $U_s$ is the same as the large system of Eqs. (11)."*

286- "invert A_diag"

Following a comment of Reviewer 2 n the numerical efficiency of the method, we have proposed to partly rewrite the part of the article detailing the Schur-complement technique. This portion now reads:

**P11 - L286**

*"[…] (ii) compute the products $A_{diag}^{-1} B_{int}$ and $A_{diag}^{-1} A_{up}$ (which is cheaper than directly inverting $A_{diag}$, (iii) iteratively [...]"*

322- "cells which then become"

We wanted to write: "that merges adjacent cells when \*they\* become smaller than a given threshold". We modified the manuscript accordingly.

379- No refreeze in this test case.

Indeed, in the glacier test case, there is not refreeze as all water is sent to runoff. This will be mentioned in the text.
We still define the phase change rate in terms of melt and refreeze (general definition) and precise that in the glacier test case there is no refreezing.

**P14 - L381**

*"Note that in this specific test case, no refreezing was observed (as melt occurs at the surface and is sent to runoff), meaning that the phase change rate directly corresponds to the melt rate."*

421- "Concerning the glacier test-case, Fig. 3 shows"

We corrected the typo.

424- "by about 0.50 K"

We corrected the typo.

440- I have, indeed, seen time step oscillations like this in class 2 simulations. They are not the same as the well-known and catastrophic instability of the explicit Euler method with too large a timestep. Considering the wide use of class 2  models, a stability analysis to understand the origin of these oscillations (not necessarily for this paper) might be of interest.

To better explain the instabilities in Class 2 models we have performed a stability analysis, akin to the ones classically performed for the Forward/Backward Euler scheme. It is provided as the end of this response.
It shows that the standard skin-layer scheme is only conditionally stable, and that there is exist a maximum time step size. The presence of instabilities is favored in the case of large thermal conductivities or of a large derivative of the atmospheric fluxes with respect to the surface temperature in the SEB. On the contrary, these instabilities are hindered in the case of large cell sizes or large specific thermal capacity.
We believe this instability is of the same nature as the one observed with an explicit time-stepping, as it arises from the use of the first internal temperature from the previous time step in the computation of the subsurface heat flux. If the internal temperature from the

current time step is used instead (as in the scheme we propose), this instability is removed.

As mentioned above, the demonstration of the (un)conditional stability of the schemes is now presented in the new Appendix A and discussed in the text.

**P21 - L503**
*"The unstable nature of class 2 models can be shown with a linear stability analysis, provided in Appendix E. Such analysis shows that class 2 models are only conditionally stable, and confirm that instabilities are favored in the case of large time steps and small mesh sizes. We stress that these oscillations can appear even if the time integration of the internal energy budget relies on the Backward Euler method, known for its robustness against instabilities (Fazio, 2001, Butcher, 2008). Our understanding is that the sequential treatment of the standard skin-layer formulation breaks the implicit nature of the time integration by using "lagged" (in other words, explicit rather than implicit) terms. This, combined with the fact that the surface layer does not possess any thermal inertia and that its temperature can thus vary rapidly in time, permits large temperature swings if the time step is too large or the mesh size too small. On the other hand, it can be shown that the two schemes with a tightly-coupled SEB are unconditionally stable (Appendix E), in agreement with the absence of oscillations in their simulations. Notably, the unconditional stability of the coupled-surface scheme proposed in this article entails that the model does not need an adaptive time step size strategy depending on the mesh size. This ensures that it remains robust, regardless of the time step and mesh size.*

**P26 - L563**
*"Moreover, a tightly-coupled treatment of the SEB allows unconditional stability, while the standard skin-layer formulation can be unstable and displays large spurious oscillations with large time steps and small mesh sizes."*

460- "only marginally worse"
What we wanted to say here, is that sometimes the Class 2 model yield smaller error than the coupled-surface scheme, but when it do so it is only be small margin (which then justifies the use of a coupled-surface model in general). This was visibly not clearly phrased, as Micheal Lehning had the same remark. We rephrased the sentence to:

**P17 - L458**
*"For almost all investigated time steps and in both test cases, the newly proposed scheme displays the lowest level of errors. Sometimes, the class 2 model yields the smallest error, but does so only by a small margin."*

We have also re-formulated a similar sentence later in the manuscript.

**P20 - L481**
*"Again, among the three implementations the tightly-coupled surface model yields the smaller errors for almost all investigated mesh refinements (as in the glacier test case, the class 2 model is however sometimes marginally better)."*

490- Divergence of the class 2 model from the reference as the mesh is refined in the glacier test case (Fig. 10) is an odd result. I guess that this could happen if the time step in these mesh refinement tests is larger than in the reference. If so, this needs to be stated in the text.

We believe the increase of error with smaller mesh size st a result of numerical instabilities, that develop with small mesh sizes. This is now mentioned in the revised manuscript:

**P20 - L490**

*"Finally, Fig. (10) reveals that in the glacier test case, the phase change rate errors of the class 2 tend to deteriorate with further mesh refinement past a certain point (here for an initial cell number above 90). We interpret this deterioration as a result of the appearance of numerical instabilities that develop with small mesh sizes."*

Having said that, it is not apparent that the 225 cell simulation is worse than the one with 45 cells in Fig. 10c.

There are periods in Fig. 10c where the error in the 225 cells simulation is larger than the 45 cells. This is notably the case from mid-June to late-August.

504- "the backward Backward Euler method" sounds like it goes forward. Just one "backward" required.

Indeed. This is now corrected.

509- "mesh size too small"

We corrected the typo.

6.4- Having found G from the SEB, the obvious thing to do in a class 2 model is to use it as a flux boundary condition for the internal temperature calculations. Can any real class 2 model be found that uses the surface temperature as a Dirichlet boundary condition? If not, section 6.4, Fig. 14 and the last sentence of the conclusion should be deleted. A note that this would be the wrong thing to do will suffice.

The potentiality of using the surface temperature as a Dirichlet condition rather than the subsurface conduction flux was made aware to us from reading the publicly available COSIPY code (cosipy/modules/heatEquation.py files, last accessed 08/11/2023) and EBFM codes. However, we stress that these codes use a Forward Euler time stepping, and in this case the using the sub-surface conduction flux or a Dirichlet condition are equivalent.

We think it is important to mention and show that using a Dirichlet condition will lead to greatly deteriorated simulations, as the use of a Dirichlet condition actually numerically stabilizes the system (which can be seen with the absence of instabilities in the orange curve of Fig. 14 and can be demonstrated with a stability analysis, provided at the end of this document). However, this stabilization is at the detriment of accuracy and energy conservation.

We propose to better justify in the manuscript that the use of a Dirichlet condition might be tempting to obtain stability, but that it will produce large errors in response. We have also shorten the first part of the section:

**P23 - L511**

*"As explained in Section 2.2, the heat conduction flux from the surface to the interior of the domain (i.e. G in Equation 3) needs to have the same value in the computation of the SEB and in the computation of the energy budget of the first interior cell. Inconsistencies in G between these two budgets lead to the violation of energy conservation and create an artificial energy source/sink near the surface. Such inconsistencies could be created when implementing the standard skin-layer formulation (class 2 models) due to the sequential*

*treatment of the surface and internal energy budgets. Indeed, after solving the SEB, one can either use the surface temperature or the subsurface heat flux G as a boundary condition for the computation of the internal temperatures. We note that the use the computed surface temperature as a boundary condition leads to an unconditionally stable numerical scheme (Appendix E). However, using such Dirichlet condition in order to stabilize the standard-skin layer formulation comes at the expense of energy conservation and deteriorates of the simulated results."*

Search the text for ", that". In all but one case, it should be "that" or ", which".
This has been corrected.

Albert (1983): https://apps.dtic.mil/sti/pdfs/ADA134893.pdf

Anderson (1968):
https://agupubs.onlinelibrary.wiley.com/doi/abs/10.1029/WR004i001p00019

Best et al. (2011): https://gmd.copernicus.org/articles/4/677/2011/

Clark et al. (2015): https://agupubs.onlinelibrary.wiley.com/doi/full/10.1002/2015WR017200

Dutra et al. (2010): https://doi.org/10.1175/2010JHM1249.1

Yamazaki and Kondo (1990): https://doi.org/10.1175/1520-0450(1990)029<0375:APMFSS>2.0.CO;2

---

## Author Comment (AC2)

We are grateful to the referee for their constructive review. Please find below our point by point response to the review. The comment of the referee are shown in blue and our response in black below. Proposed modifications of the manuscript are shown in green with page and line numbering corresponding to the preprint version of the article.

Summary:

This work proposes a methodological improvement to surface energy balance modeling over frozen ice surfaces by merging the benefits of two diverging current approaches to coupling air temperature and ice temperature. The coupling approach appears effective and insightful and is an important contribution to the field. The paper presents two case studies, one over snow and one over a glacier (with highly idealized implementations) as demonstrations of the accuracy. There is a well motivated exploration of the implementation's dependence on time and spatial resolution, which are not only practically important for anyone wishing to implement this method, but also provide the opportunity to discuss numerical stability.

General comments:

It is exciting to see this paper address both snowpack and glacier surface energy balance. It would be good to discuss (briefly) the physical similarities and differences (structure, air content) between the two.

We have added to the manuscript that snowpacks and glacier surfaces can be modeled in a similar framework as they share (i) the same fundamental governing equation (i.e. the energy conservation equation with heat conduction and shortwave absorption as a processes), and (ii) a first order phase change transition, where melt/refreeze occurs with latent heat. These similarities have already been used in the literature to treat snow and glacier ice in a unified framework, for instance in the model COSIPY. However, snowpacks and glacier surfaces present some differences that might complexify this unified treatment, for instance the fact that water does not percolate similarly in snow and glacier ice or that vapor movement plays a role significant role in snow but not in glacier ice.

We revised the manuscript to:

**P3 - L60**

*"As snowpack and glaciers share many similarities and processes, such as heat conduction or the presence of a phase transition when the melt temperature is reached, they can be represented by the same type of equations. These similarities enable simulations mixing snow and glacier ice within a single framework (e.g. Sauter et al., 2020). Hence, for the sake of generality, the equations discussed in the following sections apply to both snow and glacier ice. That being said, snow and glacier ice present some differences, notably concerning liquid water percolation. As addressed later, this might require a differential treatment of glacier ice and snow when implementing the liquid water percolation scheme."*

There is a consistent overuse of commas in the setup ',that' (many of which should be 'which' with no comma)

This was also pointed out by the review Richard Essery. This is now corrected.

The manuscript is clearly structured in introducing a new method to approach temperature and melt numerical modeling and then applying that method to two test cases. However, the test cases are very specific and thus convey limited information about the broader

application of the method – these limitations should be discussed, especially as a future goal would likely be to apply this numerical routine to more complicated cases.

This point was also stressed in the review of Richard Essery. We were also aware of this potential limitation when doing this study, and wondered if more realistic cases should be analyzed. We however decided to limit this study to simple idealized cases. Our goal behind this choice was to provide simple cases from which the impact of the numerical implementation can be clearly analyzed.

We also decided not to include comparisons with direct observations. Indeed, it would not be possible to decipher errors due to the numerical implementations (which is the focus of our paper) from errors due to the assumed physics, parametrizations, and forcings (which we do not and cannot not address in this study). Therefore, we think that to meaningfully analyze numerical implementations in terms of cost, accuracy and robustness, the use of simplified test cases is appropriate. We however agree that the test cases should not be too unrealistic if we want their results to be informative of how a numerical scheme might behave in a realistic settings. That it is why we have used realistic forcings and initial conditions.

We revised the manuscript to specify our intention more clearly. We explain that our simple test cases are meant to ease the comparison of the numerical implementations of the surface-internal energy budgets, but that our toy-model should not be viewed as proper a snowpack/glacier model as many important components are lacking.

**P12 - L330**
*"Two simple examples, showcasing the differences between numerical treatments, are presented below. We note that these simulations cannot be considered as fully realistic simulations of a snowpack or glacier surface, as many processes, such as the deposition of atmospheric precipitation or mechanical settling, are lacking. The goal is rather to provide a simplified setting in which the impact of the numerical implementation of the SEB can be analyzed. In the same idea, we do not attempt to compare the simulation results to field observations. Indeed, it would not be possible to decipher errors due numerical discretization (the focus of this paper) from errors due to the assumed physics, parametrizations and atmospheric forcing. Nonetheless, in order for the results to still be informative of how a given numerical implementation might behave in a realistic setting, we use realistic atmospheric forcings, initial conditions, and physical parametrizations. The first simulation is meant to highlight the behavior of the numerical models when simulating the surface energy balance on a snow-free glacier. The second one focuses on the impact of the model implementations on the simulation of the energy budget of a seasonal snowpack, during the melting period."*

Lastly, the finding that a coupled surface model can outperform other models at coarser grid sizes is implied here to be more computationally efficient due to the change in mesh size. However, this is not generally true when you are also changing the numerical scheme, so the assertion of computational cost savings which maintaining accuracy (as claimed here) should be backed up by either reports of the time taken for the computations and/or a clear statement that the numerical implementations are computationally identical by construction. This, if true, should also be mentioned in the conclusion as it is an important outcome! This is somewhat related to the discussion of numerical reduction (back to the same order of the original models) that you get from the Schur complement, but they are not discussed together and the data are not shown.

To answer this question we have computed the number of basic operations (addition/substraction and multiplication/division) required to perform the linear algebra

problem solvings of the three presented models, including the use of Schur-complements. The exact number are now presented in the new Appendix A and discussed in the article. We found that in terms of operations the standard skin-layer scheme requires about 40% less operations than the coupled-surface and no-surface schemes (which both require very similar number of operations). The last two schemes are more computationally expensive as they require the extra computation of the Schur-complement that is a bit more costly than the standard inversion used in the standard-skin layer formulation. Therefore, and based on the Figures 5 to 9, the introduction of a coupled degree of freedom at the surface (to transform a Class 1 into the coupled-surface scheme) is an interesting numerical trade-off, as it only marginally increase the numerical cost of the method while allowing coarser meshes. Concerning the standard skin-layer models, the trade-off of transforming into a coupled-surface scheme is not as evident as the numerical cost is multiplied by a bit less than 1.70.  It allows the use of large time steps, without numerical instabilities, but at the expense of an increased number of steps.

We propose to discuss in more details the numerical cost of the methods in Section 4.1.1

**P11 - L294**
*"An analysis of the numerical cost (in terms of number of basic operations) of this numerical scheme is given in Appendix A, alongside analyses of the numerical cost of Class 1 and 2 models. It shows that the proposed scheme and the Class 1 models have similar numerical costs, which a bit less than 1.7 times larger than the standard-skin layer."*

in the conclusion:
**P25 - L551**
*"Furthermore, a reduction technique, based on the computation of a Schur complement, is presented so that the numerical cost of the proposed framework remains of the same order as that of the standard implementations for the same mesh. In particular, for a given mesh, the numerical cost is similar to that of models not explicitly having a surface and about 1.7 larger than that of the standard-skin layer formulation."*

**P26 - L563**
*"Moreover, a tightly-coupled treatment of the SEB allows unconditional stability, while the standard skin-layer formulation can be unstable and displays large spurious oscillations with large time steps and small mesh sizes. Thus, while a bit more numerically costly, the formulation presented in this article can be used to overall reduce the numerical cost of a snowpack/glacier model through the use of larger time steps."*

As well as in the new Appendix A:
*"We see that whole system of Eqs. (A1) is a tri-diagonal system of dimension $(N+1)$x$(N+1)$, with N the number of cells. Without a Schur-complement, the computation of $A^{-1}B$ can thus be solved with Thomas algorithm in $10N-1$ base operations (addition, subtraction, multiplication, and division) per non-linear iteration (neglecting the time spent assembling the matrices). We also note that $A_{diag}$ is a tri-diagonal matrix, and thus Thomas algorithm also applies. Moreover, we see that $A_{up}$ and $A_{low}$ are almost empty matrices, which simplifies the number of operations necessary to compute $A_{diag}^{-1} A_{up}$ and $A_{low} A_{diag}^{-1} A_{up}$. Specifically, the Schur-complement technique used in this paper can be employed with $7N-9$ ($A_{diag}^{-1} A_{up}$, once per time step) + $10N-21$ ($A_{diag}^{-1} B_{int}$, once per time step) +  15 (assembly and solving of Schur-complement, once per iteration) + $2N$ (re-injection to compute $T_{int}$, once per time step) steps, i.e. a total of $17N-6 + 15n_{it}$ steps, with $n_{it}$ the number of non-linear iterations. We see, that the advantage of the Schur-complement technique is that the cost of performing non-linear iterations do not increase with the mesh*

*resolution, yielding a smaller numerically cost than inverting the while system for each non-linear iteration.*

*One may then wonder how the numerical cost of the scheme proposed in the article compares to the Class 1 and 2 models discussed in the paper. The Class 1 model (once a Schur-complement technique has been employed) as a similar numerical cost as the proposed coupled-surface scheme approach, namely 17N-23 + 15$n_{it}$ steps. For a given mesh, it has one less degree of freedom as the coupled-surface scheme and is thus only marginally cheaper. The Class 2 model is the cheapest of all schemes discussed in the paper. Indeed, once the SEB and the surface temperature have been solved through scalar non-linear iterations, it relies on a single tri-diagonal inversion of dimension NxN, which can be done in 10N-11 steps. The ratio of the numerical cost of the scheme proposed in the article over that of the standard skin-layer is of about 1.7.”*

Finally, we note that we cannot analyze this numerical cost directly in terms of computation time in our implementations. Indeed, they were implemented using the (interpreted) python language with only some parts using pre-compiled (and thus much faster) libraries. Directly comparing computation time would unfairly favor the schemes using pre-compiled librairies.

Specific comments:

L3-4: “This surface energy budget is the sum of the various surface energy fluxes, that depend on the input meteorological variables and surface temperature, and to which heat conduction towards the interior of the snow/ice and potential melting need to be added.” the comma between 'fluxes' and 'that' is incorrect, as are similarly positioned commas throughout, and 'that' should be 'which.'
This is now corrected. The same error is also corrected elsewhere in the text.

L2-4: 'and to which heat conduction towards the interior…” this sentence is unclear to me
We wanted to highlights that the conduction of heat towards the interior of the snowpack/glacier is an important factor that affects the SEB and hence the surface temperature. We clarified this in the revised manuscript:

**P1 - L2**
*“This surface energy budget is the result of various surface energy fluxes, which depend on the input meteorological variables and surface temperature, of heat conduction towards the interior of the snow/ice, and potentially of surface melting if the melt temperature is reached.”*

L26: once the SEB acronym is introduced, it should be used consistently in the paper
We now systematically use SEB instead of “surface energy budget” once introduced.

L25-30:  There is a large focus on the nonlinearity of SEB processes, which is important but not hugely challenging in the modeling field, as many of the nonlinearities are easily solved. It would be good to mention this and discuss sources of nonlinearity in a more mechanistic sense. For example, the “regime change” mentioned is due to thermal energy being used for processes with different reaction coefficients in warming frozen ice vs. phase change. This will help build intuition to support the truncation method discussed later. Perhaps mention another example.

We have reformulated the paragraph to lighten the references to non-linearity and clarified that the regime change between a melting and non-melting surface occurs at the fusion point (and not above as previously stated).
We have also precised how the SEB of melting and non-melting surface differs.

**P2 - L29**
*"Mathematically, the SEB thus appears as a non-linear top boundary condition for snowpacks and glaciers. This non-linearity is even reinforced by the existence of a regime change between a melting and non-melting surface, with different thermodynamical behaviors below and at the melting point. Indeed, once the melting point is reached at the surface, the SEB becomes more akin to a Stefan-problem with a discontinuity in the energy fluxes and can no longer be simply described in terms of surface temperature. This leads to numerical challenges when solving the governing equations."*

L42: which domain? The ice domain?
By domain we mean the physical space over which the equations are solved, that is to say in our case the snowpack or the glacier. This is now clearer in the text.

**P2 - L40**
*"On the other hand, some FVM implementations do not define a specific temperature associated with the surface, but rather use the temperature of the top-most numerical layer of the domain (i.e. the top layer of the simulated snowpack/glacier) for solving the SEB (Anderson, 1976, Brun et al., 1989, Jordan, 1991, Vionnet et al., 2012, van Kampenhout et al., 2017)."*

L63: specify Fourier's law of heat conduction
This is now specified.

L90-95: specify the sign convention used for fluxes
We now specify the sign convention for the fluxes.

Figure :1: clarify the meaning of the blue/orange colors of dots in the figures. Additional labels within the diagram would improve the clarity of the figure. It is also somewhat redundant to label Class 1 as a), class 2 as b) etc. since they are all in essentially the same panel. Consider just labeling the columns as class 1, class2, this paper.
We now specify that the nodes corresponds to variables to be solved (i.e. the cell temperatures and the surface state) and their position in space. This was also added to the caption. The color are meant to group the variables that are solved simultaneously and will be explained in the caption. We also revised the Figure to change the panels labeling to "Class1", "Class2", and "proposed scheme".

We propose for the new caption:
**P6 - Fig 1**
*"Classification of FVM models with respect to their treatment of the SEB. Class 1: The surface energy and the internal temperatures are solved in a tightly-coupled manner but there is no explicit surface. Class 2: An explicit surface temperature (and surface melting) exists but it is solved in sequential manner with respect to the internal temperatures. Proposed scheme in this article: An explicit surface temperature is considered and is solved in a tightly-coupled manner with the internal temperatures. In the schematic, dots represent the prognostic variables of the schemes (with or without temperature at the surface) while the colors indicate which variables are solved simultaneously."*

Following the review of Michael Lehning, we now discuss the equivalent of our implementation in a FEM setting. We explain that by construction, the FEM posses a surface node which naturally allows one to computed a tightly-coupled SEB with the interior of the snowpack, but that the mix of node-wise (temperatures) and element-wise (energy content, liquid water content) variables in the FEM complexifies its use. The implementation of an equivalent FEM scheme is presented in the new Appendix C (attached at the end of this response) and discussed in the manuscript:

**P4 - L112**
*"Moreover, we focus on numerical schemes based on FVM, as it is the method employed by most models (e.g. Anderson, 1976, Sauter et al., 2020, Westermann et al., 2023). We note that, contrary to the FVM, the use of the finite element method (FEM) naturally incorporates the presence of a surface temperature, which can be used for a fully-coupled treatment of the SEB, as done in SNOWPACK for instance (Bartelt and Lehning, 2002)."*

**P11 - L292**
*"Finally, a translation of this numerical strategy (including the fictitious variable and the Schur-complement technique) in a FEM framework is presented in Appendix C."*

**P12 - L329**
*"Finally, note that we do not include the FEM in this comparison. As detailed in Appendix C, a specificity of FEM models is to rely on a temperature field that can be defined element-wise or node-wise. It is thus required to convert back and forth between these two representations. However, the relation between the two is not bijective. This prevents an unambiguous transformation from element-wise to node-wise temperatures, affecting the end-result of our simulations. Because of this problem, the FEM is not further explored in this article, as a direct comparison to the FVM models is not possible."*

We modified the manuscript accordingly.

We now clearly explain how Eqs (5) and (9) can be cast as the block-system of Equation (11). This is done in a new Appendix A, attached at the end of this response.

We think that in the glacier test case, the assumption of a no-flux boundary condition at the bottom is appropriate as the temperature is essentially isothermal in this region (as given by the initial conditions derived from a COSIPY run). Moreover, as this boundary condition is far away from the surface (~189m), it would take much more than a year for it to influence the surface where we perform our analysis. To quantify this point we have run a simulation of the glacier test case with a 64.7 mW/m2 geothermal heat flux (GHF; Davies, 2013, Talalay et al., 2020) instead of a no-flux conditions. This difference in surface temperature between the simulation with and without GHF is displayed in the Figure below. It barely exceeds 4mK over the simulation, with a standard deviation of 0.4mK.

[Figure]

We added this number in the text:

**P13 - L352**
*"For instance, we performed a simulation in which a 64.7 mW m⁻² geothermal heat flux is applied instead (Davies, 2013). The impact on the surface temperature remains below 0.4 mK."*

Finally, we also want to note that the goal of our simplified simulation set-up is to provide an easy framework for the comparison of numerical methods. While more realistic boundary conditions could be used, this will not change our conclusion that are confined to behavior of the numerical schemes.

L353-355: these constants are also introduced on L 66 and 72-75, use the same symbols here to connect them.
We now re-use the alredy introduced symbol to refer to the physical variables.

L354: thermal conductivity of ice is temperature sensitive! If making this assumption, please explain why it is warranted in this case (i.e., the temperature ranges reasonably experienced in this case are small enough that there is not meaningful variation?)
Indeed, the thermal conductivity of ice is expected to vary of about 10% over the range of temperature considered in this test case (from 2.5 W/K/m at 240K to about 2.22 W/K/m at 273K[1]).
We however chose not to introduce the temperature dependence of ice in our computation as (i) this is the assumption followed by the other models discussed in the paper (i.e. COSIPY or Crocus) and (ii) this added complexity would not influence the numerical benefit of tight-coupling the surface and internal energy budgets, and the properties we want to study (time step and mesh sensitivity, stability, etc). We also want to add that including a temperature-dependence for the thermal conductivity would render the system of equation globally non-linear (rather than just locally near the surface) and would thus obscure the advantage of variable elimination to speed up the resolution of system of equation where non-linearities only appears locally. We think this last point is important has it is relevant for simplified snowpack/glacier models, which do not necessarily include such temperature dependence, and that are part of larger climate and Earth system models and where speed up of the snowpack/glacier component would be beneficial.
* * *
1    https://www.engineeringtoolbox.com/ice-thermal-properties-d_576.html

Furthermore, we have run a simulation of the glacier test case with the temperature dependence on the thermal conductivity. The difference in surface temperature between the simulations with and without this temperature-dependence is visible in the Figure below. It shows that the difference remains below 0.06K, with a standard-deviation of 0.01K.

[Figure]

We now in the revised manuscript that considering the thermal conductivity (and specific thermal capacity) as temperature-independent is a simplifying assumption that is regularly made in models and that allows the internal heat budget equation to be linear (and hence more easily solvable). This assumption could be relaxed, but to the detriment of a more numerically costly system to be solved.

**P3 - L82**
*"Finally, in this article we consider the thermal conductivity $\lambda$ and capacity $c\_p$ not to depend on temperature. The motivation for this is twofold as it (i) corresponds to a simplifying assumption regularly made by snowpack and glacier surface models (e.g. van Pelt et al., 2012, Vionnet et al., 2012, Sauter et al., 2020, Covi et al., 2023) and (ii) it allows keeping the internal heat equation linear"*

**P11 - L294**
*"We also note that to apply this technique, the assumption of temperature-independent thermal capacity and conductivity is important, as otherwise the internal heat equation system would not be linear and thus the matrices $A_{diag}$, $A_{up}$, and $A_{low}$ not constant."*

L385: albedo over what wavelength range? In most of the visible spectrum, this would be a quite low value in clean snow. Further, the longwave emissivity of snow is more density dependent due to the presence of air. It seems reasonable to use 1 for this approach, as the emissivity is still usually quite high
The albedo used in this work refers to the broadband albedo (i.e. integrated over the while solar spectrum). This is now specified in the manuscript.

We chose a simple constant value of 0.6 as the simulation is meant to take place during the melting season, when the snow albedo is at its lowest. We agree that this value is on the lower-hand of snow albedo. Thus, we have changed this value to 0.7 in the article (Section 5.2), increased the duration of the snowpack simulations, re-ran them, and

updated the numbers in the manuscript (notably Figures 4, 7, 8, 11, 12, and 14). The conclusions of the article remain unchanged.

L435: the "lag" of one time step mentioned here is interesting and well explained. The impacts of this on interpreting a snow model output may be sensitive to the time step. If there is a long time step, this would be problematic as it may prevent modeling melt. A short time step may be more resilient to this lag.
We added to the text that this lag become less problematic with short time step.

**P16 - L435**
*"The impact of this lagging problem can be mitigated by the use of small time steps, but with the drawback of numerical cost."*

L440: the observation that numerical instability is leading to differences between class 2 models and other models is interesting and seen clearly in Figure 4. The fact that this is not happening in the glacier model is only vaguely referenced. A direct comparison of the reasons for this – if there is an inherent numerical instability in class 2 models, why is there not an instability in the glacier model? Is all of the oscillation occurring in the meltwater percolation?
We do not think that the overall difference between models visible in Figure 4 can be readily explained by the presence of oscillations in the Class 2 model. For instance, at the beginning of the plot the Coupled surface and Class 2 appear quite in agreement on average, despite the occasional instabilities of Class 2. The two models then diverges (even not considering the presence of instabilities) before re-agreeing later in the graph. It is therefore not straight forward to link the overall agreement/disagreement of the two models with the presence/absence of instabilities, as there are periods with a good agreement despite instabilities, and periods a divergence despite the absence of instabilities.

While less visible, instabilities in the glacier test case are still possible, as for instance seen in Figure 13. As far as we understand, the presence or absence of oscillations is linked to the stiffness of the equations that relates the SEB and the internal temperature, and will depend on the specific thermal conductivity, thermal capacity, cell sizes, and on the time step at play.

To illustrate this point we performed a simple stability analysis of the standard skin-layer scheme (keeping in mind that this kind of stability analysis requires to linearize the system of equations, which departs from the actual scheme). The derivation is available at the end of this response and in the new Appendix E. It shows that the standard skin-layer scheme is not unconditionally stable, and that there is exist a maximum time step size. The presence of instabilities is favored in the case of large thermal conductivities or of a large derivative of the atmospheric fluxes with respect to the surface temperature in the SEB. On the contrary, these instabilities are hindered in the case of large cell sizes or large specific thermal capacity.

This is now discussed in the manuscript:

**P21 - L503**
*"The unstable nature of class 2 models can be shown with a linear stability analysis, provided in Appendix E. Such analysis shows that class 2 models are only conditionally stable, and confirm that instabilities are favored in the case of large time steps and small mesh sizes. We stress that these oscillations can appear even if the time integration of the*

*internal energy budget relies on the Backward Euler method, known for its robustness against instabilities (Fazio, 2001, Butcher, 2008). Our understanding is that the sequential treatment of the standard skin-layer formulation breaks the implicit nature of the time integration by using "lagged" (in other words, explicit rather than implicit) terms. This, combined with the fact that the surface layer does not possess any thermal inertia and that its temperature can thus vary rapidly in time, permits large temperature swings if the time step is too large or the mesh size too small. On the other hand, it can be shown that the two schemes with a tightly-coupled SEB are unconditionally stable (Appendix E), in agreement with the absence of oscillations in their simulations. Notably, the unconditional stability of the coupled-surface scheme proposed in this article entails that the model does not need an adaptive time step size strategy depending on the mesh size. This ensures that it remains robust, regardless of the time step and mesh size.*

**P26 - L563**
*"Moreover, a tightly-coupled treatment of the SEB allows unconditional stability, while the standard skin-layer formulation can be unstable and displays large spurious oscillations with large time steps and small mesh sizes."*

Figure 4: it is impossible to see the coupled surface line in panel b – consider adding markers or some other formatting choice which would allow us to see it clearly. Layering the coupled surface model on the front may help if markers are not working favorably.
Indeed, for some reason panel b of the Figure was done using a lighter shade of blue for the coupled-surface line. This was fixed and the Figure should be more readable now.

Figure 6: right panel y axis would benefit from additional labels
We added additional labels in Figures 6.

L490-495: as worded, the phrase "the class 2 model exhibits the largest phase change rate errors for an initial number of cells of 225" is ambiguous – is 225 the worst number of cells for C2 models or is C2 the worst option when working with 225 cells? From the graph, it is the second option, which is potentially less important than discussing the fact that for the other two model options, a larger number of cells generally confers better performance (within the parameter space explored here), but that is not the case for class 2 when moving from 90 to 225. Why might this be?
This deterioration is due to the development of numerical instabilities with small mesh size in the Class 2 model. This is now specified in the text.

**P20 - L 490**
*"Finally, Fig. (10) reveals that in the glacier test case, the phase change rate errors of the class 2 tend to deteriorate with further mesh refinement past a certain point (here for an initial cell number above 90). We interpret this deterioration as a result of the appearance of numerical instabilities that develop with small mesh sizes."*

L504: implicit backward Euler method?
There is one backward too much. It is now corrected.

L506: "explicit" ?
We modified the text to:

**P21 - L506**
*"'(in other words, explicit rather than implicit)".*

We corrected the typo.

The same comment was brought-up by Richard Essery. The potentiality of using the surface temperature as a Dirichlet condition rather than the subsurface conduction flux was made aware to us from reading the publicly available COSIPY code (cosipy/modules/heatEquation.py files, last accessed 08/11/2023) and EBFM codes. However, we stress that these codes use a Forward Euler time stepping, and in this case the using the sub-surface conduction flux or a Dirichlet condition are equivalent.

As mentioned in our response to the Richard Essery's review, we think it is important to mention and show that using a Dirichlet condition will lead to greatly deteriorated simulations, since the use of a Dirichlet condition actually numerically stabilizes the system (which can be seen with the absence of instabilities in the orange curve of Fig. 14 and can be demonstrated with a stability analysis, provided at the end of this document and in the new Appendix E) and might be used in this attempt. However, this stabilization is at the detriment of accuracy and energy conservation.

We propose to better justify in the manuscript that the use of a Dirichlet condition might be tempting to obtain stability, but that it will produce large errors in response. We also propose to shorten the first part of the Section:

**P23 - L511**
*"As explained in Section 2.2, the heat conduction flux from the surface to the interior of the domain (i.e. G in Equation 3) needs to have the same value in the computation of the SEB and in the computation of the energy budget of the first interior cell. Inconsistencies in G between these two budgets lead to the violation of energy conservation and create an artificial energy source/sink near the surface. Such inconsistencies could be created when implementing the standard skin-layer formulation (class 2 models) due to the sequential treatment of the surface and internal energy budgets. Indeed, after solving the SEB, one can either use the surface temperature or the subsurface heat flux G as a boundary condition for the computation of the internal temperatures. We note that the use the computed surface temperature as a boundary condition leads to an unconditionally stable numerical scheme (Appenddix E). However, using such Dirichlet condition in order to stabilize the standard-skin layer formulation comes at the expense of energy conservation and deteriorates of the simulated results."*

We modified the caption to refer to the Figure as a temperature time series, and we will add the computation of a rolling standard deviation to quantify the instabilities and their presence.

**P24 - Fig. 13**
*"Time series of surface temperatures (in blue, left y-axis) and of their 24hr-running standard deviations (in orange, right y-axis) highlighting the presence of numerical instabilities with the standard skin-layer scheme. The simulations correspond to the glacier*

*test case with a time step of 2 hr. Each panel corresponds to a level of mesh refinement. The lowest mesh refinement is at the top and displays the smallest level of instabilities, while the highest mesh refinement is at the bottom and displays numerous large instabilities in the first half of the simulation."*

L560: the level of accuracy is similar but not identical

We are not sure to fully understand the comment of the referee. We have have modified the manuscript to explain that the tightly-coupled scheme results *overall* in a better accuracy, but not always.

**P26 - L559**

*"Mesh and time step convergence analyses show that combining a coupled treatment of the SEB with the explicit introduction of a surface results in an overall better accuracy when compared to the classical implementations."*

L613 "This approach is, for instance, used in the Crocus model" add commas

We added the commas.

---

## Author Comment (AC3)

We thank the referee for their constructive review. Please find below our point by point response to the review. The comment of the referee are shown in blue and our response in black below. Proposed modifications of the manuscript are shown in green with page and line numbering corresponding to the preprint version of the article.

The authors present an approach to numerical modeling of snowpack or glacier interface with atmosphere using a finite volume method discretization of thermodynamic relations. The novelty of the approach lies in coupled computation of heat transfer through the ice/snow and the thermodynamic balance at the surface. The authors provide sufficient numerical experiments to support the agreement of their implementation with previously published results.

The only critical comment I would like to make is the relatively vague mathematical description of their approach, or the problem at hand. The authors discus the Fourier's law for the heat transfer in ice (Equation 1) and the balance of energy fluxes at the ice surface (Equation 3). Then, they immediately follow on to numerical discretization, leaving the reader curious as to what assumptions and specific method choices they made. I would outline below a few of my concerns.

We revised the manuscript trying to be more precise on the mathematical framework and on the notations. We hope the following modifications clarified the text.

The authors start with the heat equation:

$\partial_t h - \mathrm{div}\,(\lambda\,\mathrm{grad}(T)) = Q$
where
$h = c_p(T-T_0) + \rho_w L\theta$

which leads to

$$c_p\partial_t T + \rho_w L\partial_t\theta - \mathrm{div}\,(\lambda\,\mathrm{grad}(T)) = Q. \qquad (1)$$

In the subsequent paragraph they discuss issues with representing the effects of phase changes on the temperature, but I believe they mean that they neglect the $\rho_w L\partial_t\theta$ term in their model. Please state that clearly.

Yes, we meant that while solving the processes of heat conduction and shortwave absorption, we neglect the $\rho_w L\partial_t\theta$ term, and all accumulated energy is used to modify the temperature, even if the fusion point has been crossed. Note that this term is then used in a second step to re-establish thermal equilibrium between the ice and liquid water. In case of melt/refreeze, the sensible heat (cp∂tT term) and liquid water latent heat (ρwL∂tθ term) are both used to create/remove water while maintaining the energy conservation.

This is now rephrased more clearly in the revised manuscript:

**P3 - L75**
*"Note that in Eq. (1) the time derivative of the internal energy content h cannot in principle be replaced by $c_p\,d_t T$, but should also include the term $\rho_w\,L_{fus}\,d_t\theta$. Indeed, once the temperature has reached the fusion point, a further increase in energy translates into an increase in the liquid water content ($d_t\theta$ != 0) and of the associated latent heat content, rather than a further increase in the temperature. Yet, as discussed below, snowpack and glacier models nonetheless usually consider that the temperature can increase past the fusion point when integrating Eq. (1) in time (Vionnet et al., 2012, Sauter et al., 2020). This*

*is equivalent to neglecting the effects of first-order phase changes (melting and refreezing) on the temperature field, and thus setting $\rho_w\, L_{fus}\, d_t\theta$ to zero while solving the heat equation."*

Moving on, the authors jump to Equation 5, where they present the discretized version of (1) using finite volumes. It would be useful to state the implicit assumptions here, that the three dimensional equation (1) is now considered as one-dimensional equation

$$c_p \partial_t T - \partial_z (\lambda\, \partial_z T) = Q,$$

which is then integrated over each "volume", which in this case is segment of length $\Delta z_k$. This integration, along with replacing the point variables with their volume averages (with abuse of notation: $T_k = 1/\Delta z_k \int T\,dz$), and using fundamental theorem of calculus (we are in one dimension now, no need for divergence theorem) gives $\Delta z_k c_p \partial_t T - (\lambda\, \partial_z T)_{k+1/2} + (\lambda\, \partial_z T)_{k-1/2} = \Delta z_k Q$, where subscripts $k+1/2$ and $k-1/2$ refer to the (top and bottom) endpoints of the cell $\Delta z_k$

We now state directly from Eq. (1) that we are working in a 1D setting.

**P3 - L70**
*"In this article, we assume that the snowpack/glacier can be represented as 1D column, and therefore Eq. (1) should be understood as 1D equation."*

For the introduction of Equation (5) we specify that the $T_k$ in represent the average temperature of the $k^{th}$ cell. Moreover, reading the reviewer comment we realized that a subscript k is missing for the temperature in Equation (5). This is now corrected.

At this point the authors introduce the approximation of the $(\lambda\, \partial_z T)_{k+1/2}$ term with Equation 6. I am curious, however, whether it is not better to leave the term $(\lambda\, \partial_z T)_{z=surf}$ at the top of the first layer as is, and replace it with the term G from the surface energy balance equation (3)? I am not sure whether this is the way the authors achieve coupling, or whether they still discretize the temperature gradient at the ice surface using the surface temperature and half of the top layer size?

We were indeed sloppy in the description of the fluxes at the cell boundaries. We believe that the issue arises from the fact the top (and bottom) cell is a special case, which was not reflected in our article. For the top cell, the top flux is not computed using Eq. (6), but rather with the subsurface conduction flux G.
We propose to modify the text to clearly state that Eqs (5) and (6) only applies to interior cells, and that cells touching the top and bottom boundaries needs to include the boundary fluxes (which is G for the top most-cell).

**P7 - L183**
*"where $\Delta z_k$ is the thickness of the $k^{th}$ cell, $c_{pk}$ its volumetric thermal capacity, $Q_k$ the average volumetric energy source in the cell, and $F_{k+1/2}$ and $F_{k-1/2}$ are the heat conduction fluxes at the top and bottom interfaces of the cell. For internal cells, $F_{k+1/2}$ and $F_{k-1/2}$ correspond to the fluxes between the $k^{th}$ and the $k+1^{th}$ cells and the $k-1^{th}$ and $k^{th}$ cells, respectively. For the top cell $F_{k+1/2}$ corresponds the heat flux leaving towards the surface (i.e. -G) and for the bottom cell $F_{k-1/2}$ corresponds to the flux from the ground."*

*P7 - L189*
*"The heat conduction fluxes between cells need to be estimated from the temperatures and thermal conductivities of adjacent cells. The flux $F_{k+1/2}$ between cells k and k+1 is computed as:*

*Eq. (6)*

*where $\lambda^{harm}_{k+1/2}$ is the weighted harmonic average of the thermal conductivity of the two adjacent cells. The use of an harmonic average provides better results in the case of layered media such as snow (Kadioglu et al., 2008) and ensures that no heat conduction occurs in case one of the cells is a perfect thermal insulator.*
*Note that Eq. (6) only applies to fluxes between cells and must be replaced for the two boundary cells, at the top and bottom of the domain. For the bottom cell, a flux between the domain and the ground below must be used as a bottom boundary condition. For the top cell, the heat flux coming from the surface must be used. This flux corresponds to the discretized version of the term G in the SEB, provided in Eq. (10) below."*

The authors discuss in lines 103-105 that term G depends on surface temperature and temperature within ice, which indicates that this term is indeed discretized.

This term is discretized using Eq(10), and used instead of F_{k+1/2} for the energy budget of the top-most cell. This is now clearly put in the text:

*P7 - L195*
*"This flux corresponds to the discretized version of the term G in the SEB, provided in Eq. (10) below."*

I would urge the authors to provide a more detailed and careful mathematical description of their work, as it would improve the reproducibility of their result, not only for the finite volume method community, but also researchers working with other numerical discretizations.

Following the review of Michael Lehning, we have also added an Appendix describing how to implement an equivalent model using FEM (attached at the end of this response). However, in the FEM framework appears the problem of converting element-wise temperatures into node-wise temperatures. This transformation has no straight-forward answer and requires some additional assumptions that affects the end-result of the simulations. As such, we were not able to integrate a FEM model in comparisons to the FVM ones.

This is now is explained in the new Appendix C, as well as in the main part of the revised manuscript:

*P11 - L292*
*"Finally, a translation of this numerical strategy (including the fictitious variable and the Schur-complement technique) in a FEM framework is presented in Appendix C."*

*P12 - L329*
*"Finally, note that we do not include the FEM in this comparison. As detailed in Appendix C, a specificity of FEM models is to rely on a temperature field that can be defined element-wise or node-wise. It is thus required to convert back and forth between these two*

representations. However, the relation between the two is not bijective. This prevents an unambiguous transformation from element-wise to node-wise temperatures, affecting the end-result of the simulations. Because of this problem, the FEM is not further explored in this article, as a direct comparison to the FVM models is not possible."

---

## Author Comment (AC4)

We are thankful the Micheal Lehning for its constructive review. Please find below our point by point response to the review. The comment of the referee are shown in blue and our response in black below. Proposed modifications of the manuscript are shown in green with page and line numbering corresponding to the preprint version of the article.

General:
The paper presents a review on how to numerically implement the surface energy budget into a certain class of snow and ice models. The paper is very well written and in general presents the material in a clear manner. It is overall considered to be a useful contribution to the scientific community dealing with snow and ice modelling despite its rather theoretical setting, in which conclusions on existing snow and ice models are only possible in a limited way.

In this context, it is mandatory that existing snow and ice models that have schemes that come close to the solution presented here are discussed in sufficient detail. In particular, since for example SNOWPACK uses a finite element method (FEM), for which the nodal temperature is explicitly solved at the surface, it already achieves both aspects of the paper, an explicit surface and a tight coupling with internal heat transfer merely by construction of the FEM. This is true for the original version of SNOWPACK, which is now more than 20 years old. Moreover, the statement in l.81 is not a fair representation of the current state of snow models, since also efforts have been made to implement a coupled solver in SNOWPACK that does not generate temperature overshoots. This was crucial for sea ice simulations, where an additional complexity is created by the fact that the melting point of the snow and ice is a function of salinity, and that salinity in turn is impacted by the phase changes. This means that a simple approach of allowing overshoots to occur and then setting back the temperature to fusion value is not suitable any longer. This has been presented in Wever et al. (2020) and should be discussed in the current paper. The proper acknowledgment of the state of art is necessary and as a consequence limits the novelty of the proposed approach here. It is not acceptable to say "we don't discuss FEM models" as the authors do. This neglect is even more surprising since an overlapping group of authors proposes in another paper to use the FEM method for snow modelling (Brondex et al., 2023).

It is indeed true that FEM offers the advantage of naturally having a surface node, which facilitates the tightly-coupled modeling of the SEB, as done in SNOWPACK. This is now clearly mentioned in the article. We also specified that the choice of our article to focus on FVM is motivated by the fact that the FVM is broadly employed in snowpack/glacier 1D modelling. We also now include a short analysis of the FEM case (see Appendix C and modifications listed below).

**P4 - L112**
*"Moreover, we focus on numerical schemes based on the FVM, as it is the method employed by most models (e.g. Anderson, 1976, Sauter et al., 2020, Westermann et al., 2023). We note that, contrary to the FVM, the use of the finite element method (FEM) naturally incorporates the presence of a surface temperature, which can be used for a fully-coupled treatment of the SEB, as done in SNOWPACK for instance (Bartelt and Lehning, 2002)."*

We also clarified throughout the text that the classification that we propose is applies to FVM models only, for instance in the caption of Figure 1:

**P6 - Fig 1**

*"Classification of FVM models with respect to their treatment of the SEB. Class 1: The surface energy and the internal temperatures are solved in a tightly-coupled manner but there is no explicit surface. Class 2: An explicit surface temperature (and surface melting) exists but it is solved in sequential manner with respect to the internal temperatures. Proposed scheme in this article: An explicit surface temperature is considered and is solved in a tightly-coupled manner with the internal temperatures. In the schematic, dots represent the prognostic variables of the schemes (with or without temperature at the surface) while the colors indicate which variables are solved simultaneously."*

While the article is mainly focus on FVM, we wanted to include in the revised version a brief comparison with FEM, and explain how some of the points discussed in the paper (namely fictitious variable and linear elimination) can be directly translated in a FEM framework.

Doing so we stumble upon the issue of transforming element-wise energy and temperature (description required for the bucket-scheme for instance) into temperature-wise temperature (required for the FEM solving of the heat equation). This step is non-trivial as (i) it is non-unique and (ii) it can create oscillating node-wise temperature fields. While a solution to this problem has been proposed for SNOWPACK, it could not be directly translated into the sequential treatment adopted in our paper. Our different attempts to implement this elements to nodes transformation had an impact on the simulated surface temperature. Thus, the comparison between the FVM and FEM scheme in terms of accuracy and speed of convergence towards a common solution cannot be pursued in the article.

We propose to present the implementation of the FEM equivalent to the tightly-coupled scheme already discussed in the article. This is done in the new Appendix C (attached at the end of this response) and discussed in the manuscript:

**P11 - L292**

*"Finally, a translation of this numerical strategy (including the fictitious variable and the Schur-complement technique) in a FEM framework is presented in Appendix C."*

**P12 - L329**

*"Finally, note that we do not include the FEM in this comparison. As detailed in Appendix C, a specificity of FEM models is to rely on a temperature field that can be defined element-wise or node-wise. It is thus required to convert back and forth between these two representations. However, the relation between the two is not bijective. This prevents an unambiguous transformation from element-wise to node-wise temperatures, which affects the end-result of our simulations. Because of this problem, the FEM is not further explored in this article, as a direct comparison to the FVM models is not possible."*

A second major point to address is the inconsistency and incompleteness with respect to the phase change (fusion) implementation as suggested. If I understand the set-up correctly, you explicitly implement the fusion process at the surface and keep the temperature solution at the phase change temperature with your variable switching formulation supported by the truncation method. But you don't do so below the surface, which generates an inconsistency for the sub-surface heat flux. For example, for the case of shortwave penetration into snow and ice, you would generate temperatures above the melt temperature below the surface, which would lead to an upwards heat flux towards the surface, which is at the melt temperature. But heat would flow downwards in reality. This inconsistency is not even mentioned in section 6.4 and probably has consequences for

energy conservation. While the tight coupling and explicit surface are sufficiently investigated with sensitivity cases in the paper, the same needs to be done for this fusion treatment. The effect needs to be quantified and compared to the more classical "overshoot" solution.

While doing our study, we hesitated to include phase-change directly into the internal heat budget. As pointed out by the review, this treatment is closer to the actual physics at play (with phenomena such as the blocking of heat conduction fluxes in an isothermal snowpack). We nonetheless decided not to include this effect as (i) this strategy corresponds to a large portion of current snowpack and glacier models, and (ii) we foremost focus on the treatment of the SEB and a proper study/discussion on internal phase changes would be out the scope we aim for. We note that current models that do not take into the capping of internal temperatures still do include some capping of the surface temperature, since it has a large influence on the SEB (notably through the outgoing longwave radiation).

While neglecting internal phase change when solving the heat equation might lead to a deteriorated estimation of the heat conduction fluxes within the snowpack/glacier, this does not have consequences on the energy conservation of the models. As long as these heat fluxes are consistently distributed, the models remain strictly energy conservative.

To test the influence of including phase-change while solving the internal heat equation, we have implemented versions of the three FVM models used in the article that includes phase-changes directly in the heat equation, as suggested in the referee's comment. Specifically, this was done using the enthalpy method (Meyer and Hewitt, 2017, Tubini et al., 2021). Comparison with the base versions of the models shows that this inclusion has no effect on the glacier test case (as melting occurs at the surface and not internally) and an effect of a couple of degrees on the surface temperature in the snowpack test-case. Nonetheless, the conclusions of the article on the accuracy and stability of the SEB strategies remain unchanged. This can be seen in the Figures below that compare the results of the convergence study with and without internal phase change. For each figure, the left panel corresponds to the convergence plot of the manuscript (no internal phase change in the heat equation), while the right panel corresponds to the convergence plot taking into account internal phase change in the heat equation.

[Figure]

**Fig. 1** – *Impact of internal phase-changes on the mesh converge analysis.*

[Figure]

**Fig. 2 –** *Impact of internal phase-changes on the mesh converge analysis.*

[Figure]

**Fig. 3 –** *Impact of internal phase-changes on the time step converge analysis.*

[Figure]

**Fig. 4 –** *Impact of internal phase-changes on the time step converge analysis.*

We now mention in the revised manuscript that other strategies have been proposed in the literature, and we have corrected our mistake on the strategy employed by SNOWPACK.

**P3 - L81**
*"This results in temperature overshoots that are then corrected in a second step by creating melt and setting back the temperature to the melt value (e.g. Vionnet et al., 2012, Sauter et al., 2020). In this article, we follow this simple scheme as it is commonly employed in snowpack and glacier models. That being said, other, more complex,*

*strategies have been proposed in the literature. This notably includes the use of a finite temperature-range over which melt/freezing occurs (e.g. Albert, 1983, Dutra et al., 2010), including melt/refreeze as an additional energy source term (e.g. Bartelt and Lehning, 2002, Wever et al., 2020), or the use of enthalpy as the prognostic variable (e.g. Meyer and Hewitt, 2017, Tubini et al., 2021)."*

We also now mention that we have tested the sensitivity of our results to the implementation of phase-changes and that the conclusions of the article remain unchanged.

***P12 - L329***
*"Also, as some of the current snowpack and glacier models include the effect of internal phase-change while solving the internal heat equation (e.g. Bartelt and Lehning, 2002, Meyer and Hewitt, 2017), we quantified the sensitivity of our results to this specific treatment of melt/freeze. For that, we have also implemented versions of our three models that include such internal phase-changes in the heat equation."*

***P16 - L441***
*"Finally, using the versions of the models including phase-changes in the heat equation, we quantified the sensitivity of these observations to the treatment of the melt/refreeze. While the simulated temperature sometimes differ from our basic implementations (especially in the snowpack test case where melt occurs internally), the general behavior of the models, including the potential presence of instabilities in the Class 2 models, remain unchanged."*

***P20 - L493***
*"Finally, using the versions of the models including phase-changes in the heat equation, we verified that the conclusions of this convergence analysis remain valid in the case of a different treatment of the internal phase-changes"*

Minor comments:

1) At least I am more used to the terms "melt" temperature and "heat" capacity instead of "fusion" and "thermal".
We have reformulated "fusion" and "thermal capacity" into "melt" and "heat capacity", except for *"enthalpy of fusion"* as the formulation *"enthalpy of melt(ing)"* appears less common.

2) Eq. (3) does not contain heat advection by precipitation.
We have added a rain precipitation term in the SEB throughout the article.

3) l. 108: Not true, SNOWPACK does not do a separate SEB, see above.
We now specify throughout the manuscript that the proposed classification only applies to FVM models.

4) l. 126: "result" not results.
We have corrected the typo.

5) l. 284: "equation" not equations.
We have replaced sentence with:

**P11 - L284**
*"The system of Eqs.(13) is a 2x2 non-linear system where only $A_s$ and $B_s$ need to be re-assembled at each non-linear iteration and whose solution for $U_s$ is the same as the large system of Eqs. (11)."*

6) I don't understand the argument here: "Note that the method used to downscale the data does not guarantee physical consistency of the variables. This allows us to take into account shortwave, longwave and turbulent energy fluxes at the top of our domain".
We wanted to explain that we directly used the forcing data of Potocki et al. (2022), which provides all necessary inputs for the model. However, as briefly discussed in Brun et al. (2023) there are questions about the more appropriate method to downscale ERA5 data to South Col glacier.

As the goal of our article is solely focused on numerical methods and is not meant to address the quality of the forcings, we propose to simply rewrite the sentence to:

**P13 - L341**
*"As such, our simulations are forced by the weather data provided by Potocki et al. (2022) that include all necessary information to take into account the shortwave, longwave and turbulent energy fluxes at the top of our domain."*

7) Figures 3,4: These uncertainties should be discussed in light of typical snow/ice model errors.
We now compare the difference between the modeled snow surface temperature with bias observed during the inter-comparison exercise ESM-SnowMIP.

**P15 - L432**
*"As with the glacier test case, the models exhibit surface temperature differences of about a couple of degrees. This is of the same order as the biases observed in the snow model inter-comparison exercise ESM-SnowMIP (Menard et al., 2021)."*

Unfortunately, we are not aware of such an inter-comparison model exercise for glacier temperature surfaces. We therefore propose to include a mention of Sauter et al., (2020) which includes a comparison of COSIPY with measured glacier surface temperatures.

**P15 - L421**
*"Concerning the glacier test-case, Fig. 3 shows that the class 1 model (no explicit surface) is systematically different compared to the two other models, with a slower decrease of the surface temperature at night, resulting in a surface temperature that is usually warmer of a couple of degrees for the represented period. For comparison, Sauter et al., (2020) report root mean square errors around 3K when comparing COSIPY simulations with observations of the Zhadang glacier surface temperature."*

8) l. 438: why "model 2" now, not clear?
There was indeed a typo here, it the Class 1 model that produces less melt and thus that percolates less. This is now corrected in the text:

**P16 - L 438**
*"This effect is due to the smaller melting predicted by the class 1 model."*

9) l. 450 ff. should the reference not be a hundreds (900) of seconds consistent with typical time steps used?

The reference simulation is meant to replace the analytical solutions, that we cannot derive. It is meant to provide the reference toward which the numerical schemes should converge at high spatial and temporal resolutions, and should therefore be obtained with a quite small time step (30s here).

For the range of other tested time step, we decided to go above 900s as some models use larger time steps by default (3600s for COSIPY for instance) and we think it is interesting to analyze the behavior of models at large time step, as such a choice can be motivated to reduce the numerical cost of snowpack/glacier models in large simulation systems such as Earth system models.

**P17 - L452**
*"The largest time step of 7200 s corresponds to twice the default value used for instance in COSIPY (Sauter et al., 2020) and is meant to represent the case of models used at quite large time steps for numerical cost considerations."*

We wanted to state that sometimes the Class 2 yields smaller error than the scheme we proposed, but that in these cases the Class 2 is only slightly better. This was visibly not clearly enough stated in the manuscript as Richard Essery had the same comment. We revised the sentence to:

**P17 - L458**
*"For almost all investigated time steps and in both test cases, the newly proposed scheme displays the lowest level of errors. Sometimes, the class 2 model yields the smallest error, but does so only by a small margin."*

We have also re-formulated a similar sentence later in the manuscript.

**P20 - L481**

[revised manuscript text omitted]

---

## Author Comment (AC5)

**Appendix A: Matrix expressions and numerical cost of the coupled-surface scheme**

**A1  Matrix expressions**

Combing Eqs. (5), (6), and (10), the Newton scheme of the coupled-surface model proposed in this article can be written under block matrix form

$$
\left(\begin{array}{c|c} A_{\text{diag}} & A_{\text{up}} \\ \hline A_{\text{low}} & A_{\text{s}} \end{array}\right) \left(\begin{array}{c} T_{\text{int}} \\ U_{\text{s}} \end{array}\right) = \left(\begin{array}{c} B_{\text{int}} \\ B_{\text{s}} \end{array}\right)
\tag{A1}
$$

with non-zero terms being

$$
A_{\text{diag}}(k,k) = \Delta z_{\text{k}} c_{\text{p}_{\text{k}}} + \Delta t \left( \frac{\lambda_{\text{k}+\frac{1}{2}}^{\text{harm}}}{\frac{\Delta z_{\text{k}}}{2} + \frac{\Delta z_{\text{k}+1}}{2}} + \frac{\lambda_{\text{k}-\frac{1}{2}}^{\text{harm}}}{\frac{\Delta z_{\text{k}}}{2} + \frac{\Delta z_{\text{k}-1}}{2}} \right)
\tag{A2}
$$

$$
A_{\text{diag}}(k,k-1) = -\Delta t \frac{\lambda_{\text{k}-\frac{1}{2}}^{\text{harm}}}{\frac{\Delta z_{\text{k}}}{2} + \frac{\Delta z_{\text{k}-1}}{2}}
\tag{A3}
$$

$$
A_{\text{diag}}(k,k+1) = -\Delta t \frac{\lambda_{\text{k}+\frac{1}{2}}^{\text{harm}}}{\frac{\Delta z_{\text{k}}}{2} + \frac{\Delta z_{\text{k}+1}}{2}}
\tag{A4}
$$

$$
A_{\text{up}}(N-1,1) = A_{\text{low}}(1,N-1) = -\Delta t \frac{\lambda_{\text{N}-\frac{1}{2}}^{\text{harm}}}{\frac{\Delta z_{\text{N}-1}}{2} + \frac{\Delta z_{\text{N}}}{2}}
\tag{A5}
$$

$$
A_{\text{s}}(1,1) = \Delta z_{\text{N}} c_{\text{p}_{\text{N}}} + \Delta t \left( \frac{\lambda_{\text{N}-\frac{1}{2}}^{\text{harm}}}{\frac{\Delta z_{\text{N}}}{2} + \frac{\Delta z_{\text{N}-1}}{2}} + \frac{\lambda_{\text{N}}}{\frac{\Delta z_{\text{N}}}{2}} \right)
\tag{A6}
$$

$$
A_{\text{s}}(2,2) = \Delta t \left( \frac{\lambda_{\text{N}}}{\frac{\Delta z_{\text{N}}}{2}} d_{\tau} T_{\text{surf}} + L_{\text{fus}} d_{\tau} \dot{m} - d_{\tau} H - d_{\tau} L - d_{\tau} LW_{\text{out}} - - d_{\tau} R \right)
\tag{A7}
$$

$$
A_{\text{s}}(1,2) = -\Delta t \frac{\lambda_{\text{N}}}{\frac{\Delta z_{\text{N}}}{2}} d_{\tau} T_{\text{surf}}
\tag{A8}
$$

$$
A_{\text{s}}(2,1) = -\Delta t \frac{\lambda_{\text{N}}}{\frac{\Delta z_{\text{N}}}{2}}
\tag{A9}
$$

29

$$B_{\text{int}}(k) = \Delta z_k c_{\text{p}k} T_k^{n-1} + \Delta t \text{SW}_{\text{int,k}} \tag{A10}$$

 $$B_{\text{s}}(1) = \Delta z_{\text{N}} c_{\text{pN}} T_{\text{N}}^{n-1} + \Delta t \left( \text{SW}_{\text{int,N}} - \frac{\lambda_{\text{N}}}{\frac{\Delta z_{\text{N}}}{2}} \left( d_\tau T_{\text{surf}} \tau^{\text{i}} - T_{\text{s}}(\tau^{\text{i}}) \right) \right) \tag{A11}$$

$$B_{\text{s}}(2) = \Delta t \Big( SW_{\text{net}}^{\text{surf}} + LW_{\text{in}} - \frac{\lambda_{\text{N}}}{\frac{\Delta z_{\text{N}}}{2}} \left( T_{\text{s}}(\tau^{\text{i}}) - d_\tau T_{\text{surf}} \tau^{\text{i}} \right) - L_{\text{fus}} \left( m(\tau^{\text{i}}) - d_\tau m \tau^{\text{i}} \right)$$
$$+ \left( H(\tau^{\text{i}}) - d_\tau H \tau^{\text{i}} \right) + \left( L(\tau^{\text{i}}) - d_\tau L \tau^{\text{i}} \right) + \left( R(\tau^{\text{i}}) - d_\tau R \tau^{\text{i}} \right) + \left( LW_{\text{out}}(\tau^{\text{i}}) - d_\tau LW_{\text{out}} \tau^{\text{i}} \right) \Big) \tag{A12}$$

In the above expressions, $T_k^{n-1}$ is the temperature of cell $k$ at the previous time step, $\text{SW}_{\text{int,k}}$ is the quantity of shortwave radiation absorbed in cell $k$, and $\tau^{\text{i}}$ is the value of the fictitious variable $\tau$ at the start of the current non-linear iteration. The
655   terms $T_{\text{s}}(\tau^{\text{i}})$, $H(\tau^{\text{i}})$, etc, and $d_\tau T_{\text{surf}}$, $d_\tau H$, etc, are the values of the surface temperature, sensible heat flux, etc, and their derivatives at the current $\tau^{\text{i}}$ estimation.

Among the different partial derivatives, $d_\tau H$ and $d_\tau L$ can be difficult to analytically derive. For that, we first note that the chain rule yields $d_\tau H = d_{T_{\text{s}}} H d_\tau T_{\text{s}}$, and $d_\tau L = d_{T_{\text{s}}} L d_\tau T_{\text{s}}$. Then, for the expression of $H$ given in Appendix D we have:

660   $$d_{T_{\text{s}}} H = \rho_a c_{\text{p,a}} u \left( d_{T_{\text{s}}} C_{\text{H}} (T_a - T_{\text{s}}) - C_{\text{H}} \right) \tag{A13}$$

Moreover, the chain rule yields $d_{T_{\text{s}}} C_{\text{H}} = d_{\text{Ri}_{\text{b}}} C_{\text{H}} d_{T_{\text{s}}} \text{Ri}_{\text{b}}$. In our case:

$$d_{\text{Ri}_{\text{b}}} C_{\text{H}} = \frac{\kappa^2}{\ln \left( \frac{z}{z_0} \right) \left( \frac{z}{z_{0t}} \right)} \begin{cases} 0 & \text{if } \text{Ri}_{\text{b}} < 0 \\ 50\text{Ri}_{\text{b}} - 10 & \text{if } 0 \le \text{Ri}_{\text{b}} < 0.2 \\ 0 & \text{if } 0.2 \le \text{Ri}_{\text{b}} \end{cases} \tag{A14}$$

and

$$d_{T_{\text{s}}} \text{Ri}_{\text{b}} = -\frac{g z_a}{T_a u^2} \tag{A15}$$

665   Similarly, for $L$, we have:

$$d_{T_{\text{s}}} L = \rho_a L_{\text{s}} u \left( d_{T_{\text{s}}} C_{\text{E}} (q_a - q_{\text{s}}) - C_{\text{E}} d_{T_{\text{s}}} q_{\text{s}} \right) \tag{A16}$$

[revised manuscript text omitted]

---

## Author Comment (AC6)

**635 Appendix A: Matrix expressions and numerical cost of the coupled-surface scheme**

**A1 Matrix expressions**

Combing Eqs. (5), (6), and (10), the Newton scheme of the coupled-surface model proposed in this article can be written under block matrix form

$$\begin{pmatrix} A_{\text{diag}} & A_{\text{up}} \\ \hline A_{\text{low}} & A_{\text{s}} \end{pmatrix} \begin{pmatrix} T_{\text{int}} \\ \hline U_{\text{s}} \end{pmatrix} = \begin{pmatrix} B_{\text{int}} \\ \hline B_{\text{s}} \end{pmatrix}$$
(A1)

640 with non-zero terms being

$$A_{\rm diag}(k,k) = \Delta z_{\rm k} c_{\rm p_{\rm k}} + \Delta t \left( \frac{\lambda_{\rm k+\frac{1}{2}}^{\rm harm}}{\frac{\Delta z_{\rm k}}{2} + \frac{\Delta z_{\rm k+1}}{2}} + \frac{\lambda_{\rm k-\frac{1}{2}}^{\rm harm}}{\frac{\Delta z_{\rm k}}{2} + \frac{\Delta z_{\rm k-1}}{2}} \right)$$
(A2)

$$A_{\text{diag}}(k,k-1) = -\Delta t \frac{\lambda_{k-\frac{1}{2}}^{\text{harm}}}{\frac{\Delta z_{k}}{2} + \frac{\Delta z_{k-1}}{2}}$$
(A3)

$$A_{\text{diag}}(k,k+1) = -\Delta t \frac{\lambda_{k+\frac{1}{2}}^{\text{harm}}}{\frac{\Delta z_k}{2} + \frac{\Delta z_{k+1}}{2}} \tag{A4}$$

$$A_{\rm up}(N-1,1) = A_{\rm low}(1,N-1) = -\Delta t \frac{\lambda_{\rm N-\frac{1}{2}}^{\rm harm}}{\frac{\Delta z_{\rm N-1}}{2} + \frac{\Delta z_{\rm N}}{2}}$$
(A5)

$$\quad A_{\rm s}(1,1) = \Delta z_{\rm N} c_{\rm p_N} + \Delta t \left( \frac{\lambda_{\rm N-\frac{1}{2}}^{\rm harm}}{\frac{\Delta z_{\rm N}}{2} + \frac{\Delta z_{\rm N-1}}{2}} + \frac{\lambda_{\rm N}}{\frac{\Delta z_{\rm N}}{2}} \right) \tag{A6}$$

$$A_{\rm s}(2,2) = \Delta t \left( \frac{\lambda_{\rm N}}{\frac{\Delta z_{\rm N}}{2}} \mathrm{d}_{\tau} T_{\rm surf} + L_{\rm fus} \mathrm{d}_{\tau} \dot{m} - \mathrm{d}_{\tau} H - \mathrm{d}_{\tau} L - \mathrm{d}_{\tau} L W_{\rm out} - -\mathrm{d}_{\tau} R \right) \tag{A7}$$

$$A_{\rm s}(1,2) = -\Delta t \frac{\lambda_{\rm N}}{\frac{\Delta z_{\rm N}}{2}} d_{\tau} T_{\rm surf} \tag{A8}$$

$$A_{\rm s}(2,1) = -\Delta t \frac{\lambda_{\rm N}}{\frac{\Delta z_{\rm N}}{2}} \tag{A9}$$

$$B_{\rm int}(k) = \Delta z_{\rm k} c_{\rm pk} T_{\rm k}^{n-1} + \Delta t SW_{\rm int,k}$$
(A10)

$$B_{s}(1) = \Delta z_{N} c_{PN} T_{N}^{n-1} + \Delta t \left( SW_{int,N} - \frac{\lambda_{N}}{\frac{\Delta z_{N}}{2}} \left( d_{\tau} T_{surf} \tau^{i} - T_{s}(\tau^{i}) \right) \right)$$
(A11)

$$B_{\rm s}(2) = \Delta t \left( SW_{\rm net}^{\rm surf} + LW_{\rm in} - \frac{\lambda_{\rm N}}{\frac{\Delta z_{\rm N}}{2}} \left( T_{\rm s}(\tau^{\rm i}) - {\rm d}_{\tau}T_{\rm surf}\tau^{\rm i} \right) - L_{\rm fus} \left( m(\tau^{\rm i}) - {\rm d}_{\tau}m\tau^{\rm i} \right) + \left( H(\tau^{\rm i}) - {\rm d}_{\tau}H\tau^{\rm i} \right) + \left( L(\tau^{\rm i}) - {\rm d}_{\tau}L\tau^{\rm i} \right) + \left( R(\tau^{\rm i}) - {\rm d}_{\tau}R\tau^{\rm i} \right) + \left( LW_{\rm out}(\tau^{\rm i}) - {\rm d}_{\tau}LW_{\rm out}\tau^{\rm i} \right) \right)$$
(A12)

In the above expressions,  $T_k^{n-1}$  is the temperature of cell k at the previous time step, SWint,k is the quantity of shortwave radiation absorbed in cell k, and  $\tau^i$  is the value of the fictitious variable  $\tau$  at the start of the current non-linear iteration. The terms  $T_s(\tau^i)$ ,  $H(\tau^i)$ , etc, and  $d_{\tau}T_{surf}$ ,  $d_{\tau}H$ , etc, are the values of the surface temperature, sensible heat flux, etc, and their derivatives at the current  $\tau^i$  estimation.

Among the different partial derivatives,  $d_{\tau}H$  and  $d_{\tau}L$  can be difficult to analytically derive. For that, we first note that the chain rule yields  $d_{\tau}H = d_{T_s}Hd_{\tau}T_s$ , and  $d_{\tau}L = d_{T_s}Ld_{\tau}T_s$ . Then, for the expression of H given in Appendix D we have:

660
$$d_{T_s}H = \rho_a c_{p,a} u (d_{T_s}C_H(T_a - T_s) - C_H)$$
 (A13)

Moreover, the chain rule yields  $d_{T_s}C_H = d_{Ri_b}C_H d_{T_s}Ri_b$ . In our case:

$$d_{\rm Ri_b} C_{\rm H} = \frac{\kappa^2}{\ln\left(\frac{z}{z_{0t}}\right)\left(\frac{z}{z_{0t}}\right)} \begin{cases} 0 & \text{if } {\rm Ri_b} < 0\\ 50{\rm Ri_b} - 10 & \text{if } 0 \le {\rm Ri_b} < 0.2\\ 0 & \text{if } 0.2 \le {\rm Ri_b} \end{cases}$$
(A14)

and

$$d_{T_s} Ri_b = -\frac{g z_a}{T_a u^2}$$
(A15)

665 Similarly, for *L*, we have:

$$d_{T_s}L = \rho_a L_s u \left( d_{T_s} C_E(q_a - q_s) - C_E d_{T_s} q_s \right)$$
(A16)

The derivative  $d_{T_s}C_E$  can be computed as the one of  $C_H$  through the chain rule and its dependence to  $Ri_b$ . The derivative of  $q_s$  with respect to  $T_s$  can be easily obtained using the derivative of the saturated water vapor pressure, which is given by the Clausius-Clapeyron relation.

In this paper, we focus on the FVM for spatial discretization. However, the heat budget equation could also be spatially discretized with the FEM. Indeed, the FEM naturally includes a node at the surface, and thus possesses a surface temperature, which helps to tightly couple the SEB to the interior of the snowpack/glacier. This strategy is for instance employed in the SNOWPACK model (Bartelt and Lehning, 2002; Wever et al., 2020). Specifically, in SNOWPACK, the coupled SEB is intro-

715 duced as a top Robin boundary condition.

The goal of this appendix is to briefly present how the techniques presented in the main part of the manuscript (namely the use of fictitious variable and of a Schur-complement) can be used to implement a tightly-coupled FEM model.

**C1 Expression of the heat equation in FEM**

720 We consider the mesh of the domain to be discretized into N 1D elements (the direct equivalent of the cells in FVM) and thus of N + 1 nodes (the end-points of the elements). As classically done with FEM (Pepper and Heinrich, 2005), we assume the temperature field to be a linear combination of basis functions φj, i.e. T(z,t) = ∑k=1N Tj(t)φj(z). Here, we use basic linear elements. In this framework, Tj(t) corresponds to the nodal value of the temperature field (which evolves over time) and the basis functions φj(z) are piece-wise linear functions, valued 1 at node j and 0 at all other nodes. The standard Galerkin form (Pepper and Heinrich, 2005) of the internal heat budget (Eq. (1)) is:

$$\forall i \quad \sum_{j} \mathrm{d}_{t} T_{j} \int_{\Omega} c_{\mathrm{p}} \varphi_{j} \varphi_{i} \mathrm{dL} + \sum_{j} T_{j} \int_{\Omega} \lambda \nabla \varphi_{j} \cdot \nabla \varphi_{i} \mathrm{dL} = \int_{\Omega} Q \varphi_{i} \mathrm{dL} + F_{\mathrm{s}} \varphi_{i}(\mathrm{s}) \tag{C1}$$

where Ω represents the domain of simulation, Fs is the energy fluxes entering at the top of the domain (i.e. G), and φi(s) is the basis function φi evaluated at top of the domain. We note that similarly to the FVM case, the temperature at the top of the domain presents a regime change whether the surface is melting or not. To handle this, we rely on the fictitious variable τ,
730 i.e. Ts = Ts(τ). The vector of unknowns, denoted U, is thus composed of the internal temperatures and of the surface fictitious variable. Finally, we have not included any bottom energy flux to lighten the notation, but it could be included easily. Once temporally discretized with a Backward Euler scheme and linearized, the problem can be expressed in matrix form AUn = B, with A = (M + \Delta tK + \Delta tL)J\_T and B = MTn-1 + \Delta tQ + \Delta tF (Tn-1 
[revised manuscript text omitted]

and  $B = \begin{bmatrix} -\frac{\Delta zb}{\Delta zf + 2\lambda}, 0 \end{bmatrix}^T$ . We thus have,  $U_{n+1} = QU_n + M^{-1}B$ , with (E4)

$$Q = M^{-1}N = \begin{bmatrix} 0 & \frac{2\lambda}{2\lambda + \Delta zf} \\ 0 & 1 - \Delta t \frac{2\lambda}{c_p \Delta z^2} \frac{\Delta zf}{2\lambda + \Delta zf} \end{bmatrix}$$
(E5)

By recursion, it follows that  $U_n = Q^n U_0 + M^{-n} B$ . The numerical scheme is deemed stable if  $\lim_{n \to \infty} Q^n = 0$ . This is achieved if:

$$1 - \Delta t \frac{2\lambda}{c_p \Delta z^2} \frac{\Delta z f}{2\lambda + \Delta z f} | < 1 \tag{E6}$$

which after some computation yields a criterion of the time step  $\Delta t$ :

$$\Delta t < \Delta t_{\rm crit} = \frac{c_p \Delta z}{\lambda} \frac{2\lambda + \Delta z f}{f} \tag{E7}$$

840

The (linearized) standard skin-layer is thus only conditionally stable. The stability criterion is relaxed with increasing heat capacity  $(c_p)$  and increasing cell size  $(\Delta z)$ , and is made more restrictive with increasing thermal conductivity  $(\lambda)$  or if the SEB is more sensitive to changes in the surface temperature (*f* term).

**E2 Coupled-surface formulation**

Similarly, for a one cell system, the coupled-surface equations, after linearization, write:

845
$$fT_{\rm s}^{n+1} + b + \frac{2\lambda}{\Delta z} \left( T_{\rm s}^{n+1} - T_{\rm i}^{n+1} \right) = 0$$
 (E8)

for the SEB, and

$$\Delta z c_p T_{i}^{n+1} + \Delta t \frac{2\lambda}{\Delta z} \left( T_{i}^{n+1} - T_{s}^{n+1} \right) = \Delta z c_p T_{i}^{n}$$
(E9)

for the cell's heat budget. These two equations can be cast into the matrix form  $MU_{n+1} = NU_n + B$ , with  $B = [-\frac{\Delta zb}{\Delta z f + 2\lambda}, 0]^T$ ,

$$M = \begin{bmatrix} 1 & \frac{-2\lambda}{2\lambda + \Delta zf} \\ -\frac{2\Delta t\lambda}{c_p \Delta z^2 + 2\lambda \Delta t} & 1 \end{bmatrix}$$
(E10)

850

and

$$N = \begin{bmatrix} 0 & 0\\ 0 & \frac{c_p \Delta z^2}{c_p \Delta z^2 + 2\lambda \Delta t} \end{bmatrix}$$
(E11)

We thus have  $U_n = Q^n U_0 + M^{-n} B$ , with:

$$Q = \begin{bmatrix} 0 & \frac{2\lambda}{2\lambda + \Delta zf} \frac{c_p \Delta z^2}{c_p \Delta z^2 + 2\lambda \Delta t} \\ 0 & \frac{c_p \Delta z^2}{c_p \Delta z^2 + 2\lambda \Delta t} \end{bmatrix}$$
(E12)

The numerical scheme is deemed stable if  $\lim_{n\to\infty} Q^n = 0$ . This is always achieved, as  $\frac{c_p \Delta z^2}{c_p \Delta z^2 + 2\lambda \Delta t} < 1$ . Thus, the surfacescoupled scheme is unconditionally stable.

**E3 Non-conservative skin-layer formulation**

For the non-conservative skin-layer formulation (see Section 6.4), we start with the linearized discrete SEB:

$$fT_{\rm s}^{n+1} + b + \frac{2\lambda}{\Delta z} \left( T_{\rm s}^{n+1} - T_{\rm i}^n \right) = 0$$
(E13)

Using the surface temperature  $T_s^{n+1}$  as a Dirichlet condition for the internal energy budget, we thus have

860
$$\Delta z c_p T_i^{n+1} + \Delta t \frac{2\lambda}{\Delta z} \left( T_i^{n+1} - T_s^{n+1} \right) = \Delta z c_p T_i^n$$
(E14)

These two equations can be cast into the matrix form  $MU_{n+1} = NU_n + B$ , with  $B = [-\frac{\Delta zb}{\Delta z f + 2\lambda}, 0]^T$ ,

$$M = \begin{bmatrix} 1 & 0\\ -\frac{2\Delta t\lambda}{c_p \Delta z^2 + 2\lambda \Delta t} & 1 \end{bmatrix}$$
(E15)

and

$$N = \begin{bmatrix} 0 & \frac{2\lambda}{2\lambda + \Delta zf} \\ 0 & \frac{c_p \Delta z^2}{c_p \Delta z^2 + 2\lambda \Delta t} \end{bmatrix}$$
(E16)

865 We thus have  $U_n = Q^n U_0 + M^{-n} B$ , with:

$$Q = \begin{bmatrix} 0 & \frac{2\lambda}{2\lambda + \Delta zf} \\ 0 & X \end{bmatrix}$$
(E17)

where  $X = \frac{2\lambda\Delta t \frac{2\lambda}{2\lambda+\Delta zf} + c_p \Delta z^2}{2\Delta t \lambda + c_p \Delta z^2}$ . The scheme is deemed stable if |X| < 1.

As  $\frac{2\lambda}{2\lambda+\Delta zf} < 1$ , we always have that  $2\lambda\Delta t \frac{2\lambda}{2\lambda+\Delta zf} + c_p\Delta z^2 < 2\Delta t\lambda + c_p\Delta z^2$ , and thus that the scheme is unconditionally stable. That being said, we recall that this scheme is not energy conservative and can lead to large errors.

**E4 No-surface formulation (Class 1)**

Finally, we note that the linearized No-surface formulation corresponds to a classic heat equation with a Backward Euler time integration. As demonstrated elsewhere in the literature (e.g. Butcher, 2008), it is unconditionally stable.